# RETHINKING DEGREE-CORRECTED SPECTRAL CLUSTERING: A PURE SPECTRAL ANALYSIS & EXTENSION

## ABSTRACT

Spectral clustering is a representative graph clustering technique with strong interpretability and theoretical guarantees. Recently, degree-corrected spectral clustering (DCSC) has emerged as the state-of-the-art for this technique. While prior studies have provided several theoretical results for DCSC, their analysis relies on some random graph models (e.g., stochastic block models). In this study, we explore an alternative analysis of DCSC from a pure spectral view. It gives rigorous bounds for the mis-clustered volume and conductance w.r.t. the optimal solution while involving quantities that indicate impacts of (i) high degree heterogeneity and (ii) weak clustering structures to DCSC. Inspired by recent advances in graph neural networks (GNNs) and the associated over-smoothing issue, we propose ASCENT (Adaptive Spectral ClustEring with Node-wise correcTion), a simple yet effective extension of DCSC. Different from most DCSC methods with a constant degree correction for all nodes, ASCENT follows a node-wise correction scheme. It can assign different corrections for nodes via the mean aggregation of GNNs. We further demonstrate that (i) ASCENT reduces to conventional DCSC methods when encountering over-smoothing and (ii) some early stages before over-smoothing can potentially obtain better clustering quality.

## 1 INTRODUCTION

Graph clustering (a.k.a. disjoint community detection) is a classic inference task that partitions nodes of a graph into densely connected groups (a.k.a. clusters or communities). Since the extracted clusters have been validated to correspond to some substructures of real-world systems (e.g., functional groups in protein interactions (Berahmand et al., 2021)), many network applications (e.g., protein complex detection (Qin & Gao, 2010), cellular network decomposition (Dai & Bai, 2017), and Internet traffic profiling (Qin et al., 2019)) are formulated as graph clustering.

Spectral clustering is one of the representative techniques for this task. As summarized in Table 1, a typical spectral clustering algorithm includes the (I) eigen-decomposition (ED) on graph Laplacian, (II) arrangement of spectral embedding, (III) normalization of the arranged embedding, and (IV) $K$Means clustering. In Table 1, $\mathbf{A}$ and $\mathbf{D}$ are the adjacency matrix and corresponding degree diagonal matrix of a graph; $K$ is a pre-set number of clusters; $\lambda_r$ denotes the $r$-th largest eigenvalue of graph Laplacian $\mathbf{L}$ (e.g., $\mathbf{L} := \mathbf{D}^{-1/2}\mathbf{A}\mathbf{D}^{-1/2}$ and $\mathbf{L} := \mathbf{A}$ for *NJW* (Ng et al., 2001) and *SCORE* (Jin, 2015)) with $\mathbf{u}_r \in \mathbb{R}^N$ as the corresponding eigenvector. Different spectral clustering algorithms usually differ in terms of the four steps. For instance, *NJW*, *SCORE*, and *RSC* (Qin & Rohe, 2013) only consider eigenvectors $(\mathbf{u}_1, \cdots, \mathbf{u}_K)$ w.r.t. the leading $K$ eigenvalues. Whereas, step (II) of *SCORE+* (Jin et al., 2021) and *ISC* (Qing & Wang, 2020a) involves $(\mathbf{u}_1, \cdots, \mathbf{u}_K, \mathbf{u}_{K+1})$, which are further reweighted by corresponding $(K+1)$ eigenvalues $(\lambda_1, \cdots, \lambda_K, \lambda_{K+1})$. Moreover, *NJW*, *RSC*, and *ISC* adopt the row-wise $l_2$-normalization in step (III), while *SCORE* and *SCORE+* use (reweighted) $\mathbf{u}_1$ to conduct column-wise normalization.

Recently, degree-corrected spectral clustering (DCSC), a.k.a. regularized spectral clustering in some literature (Qin & Rohe, 2013; Zhang & Rohe, 2018), has emerged as a state-of-the-art class of spectral clustering methods, due to their effectiveness in handling the high degree heterogeneity of graphs. These approaches usually incorporate an additional degree correction term $\tau$ in their graph Laplacian for ED (e.g., *RSC*, *SCORE+*, and *ISC* with different settings of $\tau$ in Table 1).

Table 1: Summary of some spectral clustering algorithms, where $\mathbf{D}_\tau := \mathbf{D} + \tau\mathbf{I}_N$; $\tau$ is the degree correction term in DCSC, with $\tau = \bar{d}$, $\delta d_{\max}$, and $\delta(d_{\min} + d_{\max})/2$ for *RSC*, *SCORE+*, and *ISC* (e.g., $\delta = 0.1$); $\bar{d}$, $d_{\min}$, and $d_{\max}$ are the average, minimum, and maximum node degrees.

| | Step (I) | Step (II) | Step (III) | Step (IV) |
|---|---|---|---|---|
| *NJW* | ED on $\mathbf{D}^{-1/2}\mathbf{A}\mathbf{D}^{-1/2}$ | $\mathbf{F} := [\mathbf{u}_1, \cdots, \mathbf{u}_K]$ | $i \in [1, N], \mathbf{F}_{i,:} \leftarrow \mathbf{F}_{i,:}/|\mathbf{F}_{i,:}|_2$ | |
| *SCORE* | ED on $\mathbf{A}$ | $\mathbf{F} := [\mathbf{u}_2, \cdots, \mathbf{u}_K]$ | $r \in [1, K-1], \mathbf{F}_{:,r} \leftarrow \mathbf{F}_{:,r}/\mathbf{u}_1$ | $K$ Means |
| *RSC* | | $\mathbf{F} := [\mathbf{u}_1, \cdots, \mathbf{u}_K]$ | $i \in [1, N], \mathbf{F}_{i,:} \leftarrow \mathbf{F}_{i,:}/|\mathbf{F}_{i,:}|_2$ | on rows |
| *SCORE+* | ED on $\mathbf{D}_\tau^{-1/2}\mathbf{A}\mathbf{D}_\tau^{-1/2}$ | $\mathbf{F} := [\lambda_2\mathbf{u}_2, \cdots, \lambda_{K+1}\mathbf{u}_{K+1}]$ | $r \in [1, K], \mathbf{F}_{:,r} \leftarrow \mathbf{F}_{:,r}/(\lambda_1\mathbf{u}_1)$ | of $\mathbf{F}$ |
| *ISC* | | $\mathbf{F} := [\lambda_1\mathbf{u}_1, \cdots, \lambda_{K+1}\mathbf{u}_{K+1}]$ | $i \in [1, N], \mathbf{F}_{i,:} \leftarrow \mathbf{F}_{i,:}/|\mathbf{F}_{i,:}|_2$ | |

Table 2: Summary of representative theoretical analysis among DCSC, where most related studies rely on the assumption of a random graph model and give bounds w.r.t. such a model.

| Analysis | Rand Model | Theoretical Bounds | Analysis | Rand Model | Theoretical Bounds |
|---|---|---|---|---|---|
| (Chaudhuri et al., 2012) | EPP model | EPP's optimal separation | (Qing & Wang, 2020a) | DCSBM | Hamming error w.r.t. DCSBM's gnd |
| (Qin & Rohe, 2013) | DCSBM | Misclustered rate w.r.t. | (Qing & Wang, 2020b) | DCSBM | |
| (Amini et al., 2013) | DCSBM | DCSBM's groundtruth | (Jin et al., 2021) | DCSBM | |
| (Zhang & Rohe, 2018) | DCSBM | Conductance | **Ours** | N/A | Misclustered vol. & conductance w.r.t. optimal solution |

**Related Theoretical Analysis on DCSC**. In the past few decades, a series of spectral clustering methods have been proposed. Ding et al. (2024) provided a comprehensive overview of related research. Table 2 summarizes some representative theoretical results regarding DCSC. We introduce more related work about recent advances in deep graph clustering in Appendix A.

As in Table 2, Chaudhuri et al. (2012) proposed a DCSC method for graphs drawn from an extended planted partition (EPP) model (Condon & Karp, 2001) and examined the performance guarantees. Qin & Rohe (2013) analyzed the potential of *RSC* to handle the high degree heterogeneity of graphs using the degree-corrected stochastic blockmodel (DCSBM) (Karrer & Newman, 2011) and provided guidance on the choice of $\tau$. (Zhang & Rohe, 2018) theoretically studied the (i) failures of spectral clustering and (ii) benefits of degree correction based on the relationship between graph conductance and spectral clustering. Amini et al. (2013) introduced a fast pseudo-likelihood method for fitting DCSBM with theoretical guarantees, where a DCSC algorithm with perturbations was used for initialization. Qing & Wang (2020a) and Jin et al. (2021) proposed *ISC* and *SCORE+*, which were further validated to be effective in handling the (i) high degree heterogeneity and (ii) weak clustering structures (a.k.a. weak signals in (Qing & Wang, 2020b;a; Jin et al., 2021)) via the theoretical analysis based on DCSBM.

In summary, most existing theoretical studies of DCSC rely on some assumptions of random graph models (e.g., EPP model and DCSBM). They usually fit the adjacency matrix or graph Laplacian using a certain random graph model (e.g., $\mathcal{A} := \boldsymbol{\Theta}\mathbf{Z}\mathbf{B}\mathbf{Z}^T\boldsymbol{\Theta}$ (Qin & Rohe, 2013) with $\{\boldsymbol{\Theta}, \mathbf{Z}, \mathbf{B}\}$ as notations defined in DCSBM) and further give theoretical bounds related to such a model (e.g., mis-clustered rate and Hamming error w.r.t. the ground-truth given by DCSBM).

**Present Analysis on DCSC & Extension**. Spectral clustering is a typical approximated algorithm for the NP-hard combinatorial optimization problem of conductance minimization (Von Luxburg, 2007). Based on this nature, some early studies (Peng et al., 2015; Mizutani, 2021) analyzed vanilla spectral clustering (e.g., *NJW*) using the spectral graph theory. Motivated by these studies, we consider an alternative analysis for DCSC from a pure spectral view, instead of using random graph models. Different from existing analysis on DCSC with bounds related to a random graph model, we provide theoretical bounds for the mis-clustered volume and conductance w.r.t. the optimal solution to conductance minimization. In contrast to early spectral-based studies on vanilla spectral clustering (Peng et al., 2015; Mizutani, 2021), our analysis involves additional quantities about (i) degree heterogeneity and (ii) weakness of clustering structures, which can help reveal impacts of (i) high degree heterogeneity and (ii) weak clustering structures to DCSC.

Inspired by recent advances in graph neural networks (GNNs) and the associated over-smoothing issue (Rusch et al., 2023), we propose ASCENT (Adaptive Spectral ClustEring with Node-wise correcTion), a simple yet effective extension of DCSC. Instead of using a constant correction term $\tau$ for all nodes (e.g., *RSC*, *SCORE+*, and *ISC* in Table 1), ASCENT follows a node-wise correction scheme, where nodes $\{v_i\}$ are allowed to be assigned with different corrections $\{\tau_i\}$. Such a scheme iteratively updates $\{\tau_i\}$ via the mean aggregation of GNNs, where nodes $\{v_i\}$ with more common high-order neighbors (e.g., in the same cluster) are more likely to have close $\{\tau_i\}$. Consistent with the over-smoothing issue of GNNs, $\{\tau_i\}$ will finally converge to a constant. In this case, ASCENT

reduces to conventional DCSC methods. Our experiments demonstrate that some early stages of this updating procedure (i.e., before over-smoothing) can potentially result in better clustering quality.

## 2 PROBLEM STATEMENTS & PRELIMINARIES

In general, an undirected and unweighted simple graph can be represented as a 2-tuple $G := (V, E)$, where $V := \{v_1, \cdots, v_N\}$ and $E := \{(v_i, v_j)|v_i, v_j \in V\}$ are the sets of nodes and edges. One can use an adjacency matrix $\mathbf{A} \in \{0, 1\}^{N \times N}$ to describe the topology of $G$, where $\mathbf{A}_{ij} = \mathbf{A}_{ji} = 1$ if $(v_i, v_j) \in E$ and $\mathbf{A}_{ij} = \mathbf{A}_{ji} = 0$ otherwise. Let $\mathbf{D} := \mathrm{diag}(d_1, d_2, \cdots, d_N)$ be the degree diagonal matrix of $G$, with $d_i := \sum_j \mathbf{A}_{ij}$ as the degree of node $v_i$.

Given a graph $G$ and a pre-set number of clusters $K$, **graph clustering** (a.k.a. **disjoint community detection**) aims to partition $V$ into $K$ disjoint subsets $(C_1, \cdots, C_K)$, which are defined as clusters or communities, with $\bigcup_r C_r = V$ and $C_r \cap C_t = \emptyset$ ($\forall r \neq t$) s.t. (i) within each cluster the edge connections between nodes are dense but (ii) between clusters the connections are relatively loose.

Note that we follow the classic problem statement of spectral clustering, where graph topology is the only available information source. Different from most deep graph clustering methods (Nazi et al., 2019; Bo et al., 2020; Bianchi et al., 2020; Tsitsulin et al., 2023; Bhowmick et al., 2024), our analysis does not consider graph attributes, due to the complicated correlations between topology and attributes validated by prior studies (Newman & Clauset, 2016; Qin et al., 2018; Wang et al., 2020; Qin & Lei, 2021). Concretely, the simple integration of attributes may bring inconsistent features or noise that lead to quality decline compared with the case only considering topology, although attributes may sometimes provide complementary information for better clustering quality.

Graph clustering is an approximated algorithm for the combinatorial optimization objective of conductance minimization (Von Luxburg, 2007). For a subset $S \subseteq V$, let $E(S, V \backslash S) := \{(v_i, v_j) \in E : v_i \in S, v_j \in V \backslash S\}$ be the set of edges across $S$ and $V \backslash S$. Let $\mu(S) := \sum_{v_i \in S} d_i$ be the **volume** of $S$. The **conductance** of $S$ is defined as $\phi(S) := |E(S, V \backslash S)|/\mu(S)$.

**Definition 1 (Conductance Minimization)** *Let $U$ be the collection of all possible $K$-way partitions of the node set $V$ in graph $G$. The **conductance minimization** objective is defined as*

$$\bar{\phi}_K(G) := \min_{(S_1, \cdots, S_K) \in U} \frac{1}{K}(\phi(S_1) + \cdots + \phi(S_K)). \tag{1}$$

*It aims to find a partition $(S_1, \cdots, S_K)$ of $V$ that can achieve the **minimal average conductance** $\bar{\phi}_K(G)$. We define that a partition $(S_1, \cdots, S_K)$ is $\bar{\phi}_K(G)$-**optimal** if its average conductance $(\phi(S_1) + \cdots + \phi(S_K))/K$ achieves $\bar{\phi}_K(G)$.*

For the ED on graph Laplacian (i.e., step (I) of Table 1), let $\lambda_r$ and $\mathbf{u}_r \in \mathbb{R}^N$ denote the $r$-th largest eigenvalue and corresponding eigenvector. When considering the normalized graph Laplacian $\mathbf{D}^{-1/2}\mathbf{A}\mathbf{D}^{-1/2}$, we have $1 = \lambda_1 \geq \cdots \geq \lambda_N \geq -1$[1] and $\mathbf{u}_r^T \mathbf{u}_t = 0$ ($\forall r \neq t$). Moreover, we have $1 > \lambda_1 \geq \lambda_2 \geq \cdots \geq \lambda_N$ for the regularized graph Laplacian $\mathbf{D}_\tau^{-1/2}\mathbf{A}\mathbf{D}_\tau^{-1/2}$. In step (II) of Table 1, we arrange the (reweighted) eigenvectors as a matrix $\mathbf{F} \in \mathbb{R}^{N \times K}$ (or $\mathbb{R}^{N \times (K+1)}$) via the column-wise concatenation. We define the $i$-th row $\mathbf{F}_{i,:}$ of $\mathbf{F}$ as the **spectral embedding** of node $v_i$. Most spectral clustering algorithms apply normalization to $\mathbf{F}$ (i.e., step (III) in Table 1). We denote the corresponding **normalized spectral embedding** as $\tilde{\mathbf{F}}$.

**Definition 2 (Clustering Cost)** *Given a set of vectors $(\mathbf{w}_1, \cdots, \mathbf{w}_K)$, we follow (Peng et al., 2015) to define the **distance** between a partition $(S_1, \cdots, S_K)$ of $V$ and $(\mathbf{w}_1, \cdots, \mathbf{w}_K)$ as*

$$g(S_1, \cdots, S_K; \mathbf{w}_1, \cdots, \mathbf{w}_K) := \sum_{r=1}^K \sum_{v_i \in S_r} d_i \left\| \tilde{\mathbf{F}}_{i,:} - \mathbf{w}_r \right\|_2^2. \tag{2}$$

*It maps each node $v_i$ to $d_i$ identical points in the embedding space. As claimed in (Peng et al., 2015), this definition allows us to bound the overlap between (i) feasible clustering results and (ii)*

---

[1] Some literature (Von Luxburg, 2007; Qin et al., 2023; Gao et al., 2023) defines the normalized graph Laplacian as $\mathbf{I}_N - \mathbf{D}^{-1/2}\mathbf{W}\mathbf{D}^{-1/2}$, which equivalently has the eigenvalues of $0 = 1 - \lambda_1 \leq \cdots \leq 1 - \lambda_N \leq 2$.

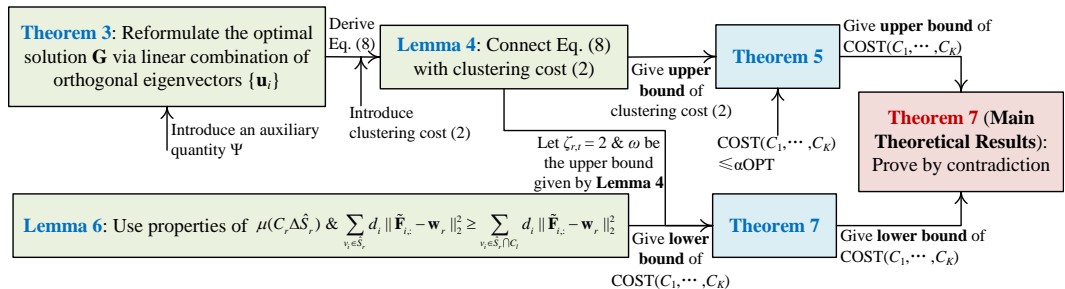

Figure 1: The high-level overview of our pure spectral analysis on DCSC.

*optimal ones, which is further used in our analysis (cf. Lemma 4 and Lemma 6). By assuming that for each node $v_i \in V$, all the $d_i$ copies of $\tilde{\mathbf{F}}_{i,:}$ are contained in one of $\{S_1, \cdots, S_K\}$, (2) reduces to the standard cost of KMeans. The **clustering cost** of a partition $(S_1, \cdots, S_K)$ is then defined as*

$$\text{COST}(S_1, \cdots, S_K) := \min_{(\mathbf{c}_1, \cdots, \mathbf{c}_K)} g(S_1, \cdots, S_K; \mathbf{c}_1, \cdots, \mathbf{c}_K), \tag{3}$$

*which finds a set of centers $(\mathbf{c}_1, \cdots, \mathbf{c}_K)$ with the minimum distance to $(S_1, \cdots, S_K)$. Based on $\text{COST}(S_1, \cdots, S_K)$, we define the **optimal clustering cost** as*

$$\text{OPT} := \min_{(S_1, \cdots, S_K) \in U} \text{COST}(S_1, \cdots, S_K). \tag{4}$$

## 3 PROPOSED ANALYSIS: A PURE SPECTRAL VIEW

Inspired by early spectral-based studies (Peng et al., 2015; Mizutani, 2021) on vanilla spectral clustering (e.g., *NJW*), we give an alternative analysis for DCSC from a pure spectral view, without using random graph models. We adopt *ISC* (see Table 1) as an example for analysis because it has a more generic format involving the reweighted $(K + 1)$ leading eigenvectors $[\lambda_1 \mathbf{u}_1, \cdots, \lambda_{K+1} \mathbf{u}_{K+1}]$. Whereas, other DCSC methods usually have simpler formats (e.g., only $[\mathbf{u}_1, \cdots, \mathbf{u}_K]$ without reweighting for *RSC*). Fig. 1 illustrates the overall sketch of our analysis. In Appendix F, we further reduce this generic analysis on *ISC* to other DCSC algorithms (e.g., *SCORE+* and *RSC*), which extends our analysis to a unified framework involving a series of spectral clustering approaches.

In contrast to early work (Peng et al., 2015; Mizutani, 2021) on vanilla spectral clustering, our analysis aims to reveal impacts of (i) degree heterogeneity and (ii) weakness of clustering structures to the clustering quality of DCSC. We first introduce a quantity measuring both aspects:

$$\Psi_{\text{ISC}} := m_K^{-1}[1 - h \cdot (1 - \bar{\phi}_K(G))] = \frac{1}{(1 - \lambda_{K+2})}[1 - \frac{d_{\min}}{d_{\max} + \tau}(1 - \bar{\phi}_K(G))], \tag{5}$$

where $m_K := 1 - \lambda_{K+2}$ and $h := d_{\min}/(d_{\max} + \tau)$. In (5), $h$ measures the degree heterogeneity, where a small $h$ (i.e., a large difference between $d_{\min}$ and $d_{\max}$) indicates high degree heterogeneity. Since $\bar{\phi}_K(G) \leq 1$, *higher degree heterogeneity (i.e., a smaller $h$) will lead to a larger $\Psi_{\text{ISC}}$.*

As validated in (Jin et al., 2021), when clustering structures of a graph (with $K$ clusters) are weak, $\tilde{m}_K := 1 - \lambda_{K+1}/\lambda_K$ is small, which is consistent with a small $|\lambda_K - \lambda_{K+1}|$ by the **eigen-gap property** of graph Laplacian (Von Luxburg, 2007). Since $1 > \lambda_K \geq \lambda_{K+1} \geq \lambda_{K+2}$, we have

$$m_K \geq \tilde{m}_K \text{ and } \Psi_{\text{ISC}} = m_K^{-1}[1 - h(1 - \bar{\phi}_K(G))] \leq \tilde{m}_K^{-1}[1 - h(1 - \bar{\phi}_K(G))]. \tag{6}$$

Therefore, *weaker clustering structures (i.e., a smaller $\tilde{m}_K$) indicates a larger upper bound of $\Psi_{\text{ISC}}$.*

**Theorem 3** *Let $(\hat{S}_1, \cdots, \hat{S}_K)$ be a $\bar{\phi}_K(G)$-**optimal** partition, with the partition membership encoded by $\mathbf{G} \in \mathbb{R}^{N \times K}$, where $\mathbf{G}_{ir} = \sqrt{d_i/\mu(\hat{S}_r)}$ if $v_i \in \hat{S}_r$ and $\mathbf{G}_{ir} = 0$ otherwise. $\mathbf{F} := [\lambda_1 \mathbf{u}_1, \cdots, \lambda_{K+1} \mathbf{u}_{K+1}] \in \mathbb{R}^{N \times (K+1)}$ is the **spectral embedding** of ISC (i.e., step (II) of Table 1). If $K\Psi_{\text{ISC}} \leq 1$, there exists an orthogonal matrix $\mathbf{O} := [\mathbf{o}_1, \cdots, \mathbf{o}_K] \in \mathbb{R}^{(K+1) \times K}$ s.t.*

$$\|\mathbf{FO} - \mathbf{G}\|_F \leq (1 + \lambda_1)\sqrt{K\Psi_{\text{ISC}}}. \tag{7}$$

As in Fig. 1, one can prove **Theorem 3** by reformulating $\mathbf{G}$ via the linear combination of orthogonal eigenvectors $\{\mathbf{u}_i\}$ (see Appendix B for the full proof). The first term in (7) can be rewritten as

$$\|\mathbf{FO} - \mathbf{G}\|_F^2 = \|\mathbf{F} - \mathbf{GO}^T\|_F^2 = \|\mathbf{F}^T - \mathbf{OG}^T\|_F^2 = \sum_{r=1}^{K} \sum_{v_i \in S_r} \left\| \mathbf{F}_{i,:} - \sqrt{\frac{d_i}{\mu(S_r)}} \mathbf{o}_r \right\|_2^2. \quad (8)$$

By using the same strategy as the proof of Lemma 2 in (Mizutani, 2021), which connects (8) with (2), we can derive the following upper bound of **clustering cost** (see Appendix C for the full proof).

**Lemma 4** *Let* $(\hat{S}_1, \cdots, \hat{S}_K)$ *be a* $\bar{\phi}_K(G)$-***optimal*** *partition and* $\tilde{\mathbf{F}}$ *be the* ***normalized spectral embedding*** *of ISC.* $\{\mathbf{o}_r\}$ *are with the same definitions as those in* ***Theorem 3***. *The followings hold:*

- $\|\mathbf{o}_r - \mathbf{o}_t\|_2^2 = 2$, $\forall r, t \in \{1, 2, \cdots, K\}$ *and* $r \neq t$;

- $g(\hat{S}_1, \cdots, \hat{S}_K; \mathbf{o}_1, \cdots, \mathbf{o}_K) \leq 4(1 + \lambda_1)^2 \mu_{\max} K \Psi_{\text{ISC}},$

*with* $\mu_{\max} := \max_{\hat{S}_r} \mu(\hat{S}_r)$ *as the maximum* ***volume***.

Obviously, we have $\text{OPT} \leq \text{COST}(C_1, \cdots, C_K) \leq g(\hat{S}_1, \cdots, \hat{S}_K; \mathbf{o}_1, \cdots, \mathbf{o}_K)$. Assume that $K$Means has an approximation ratio of $\alpha$ (i.e., $\text{COST}(C_1, \cdots, C_K) \leq \alpha \text{OPT}$), which depends on the concrete $K$Means algorithm we used (e.g., $\alpha = O(\log K)$ for $K$Means++ (Arthur & Vassilvitskii, 2007)). One can directly derive the following **Theorem 5** based on **Lemma 4**.

**Theorem 5** *Let* $(C_1, \cdots, C_K)$ *be a feasible clustering result given by ISC. When the $K$Means clustering algorithm has an approximation ratio of* $\alpha$, *we have*

$$\text{COST}(C_1, \cdots, C_K) \leq 4(1 + \lambda_1)^2 \alpha \mu_{max} K \Psi_{\text{ISC}}. \quad (9)$$

Furthermore, the lower bound of $\text{COST}(C_1, \cdots, C_K)$ can be obtained via the following **Lemma 6**.

**Lemma 6** *(Mizutani, 2021) For every* ***permutation*** $\pi : \{1, \cdots, K\} \to \{1, \cdots, K\}$, *assume that there is an index $l$ s.t.* $\mu(C_l \Delta \hat{S}_{\pi(l)}) \geq 2\epsilon \cdot \mu(\hat{S}_{\pi(l)})$, *with* $A \Delta B := (A \backslash B) \cup (B \backslash A)$ *as the* ***symmetric difference*** *between two sets and* $0 \leq \epsilon \leq 1/2$. *Let* $\zeta_{r,t}$ *and* $\omega$ *be the lower bound of* $\|\mathbf{o}_r - \mathbf{o}_t\|_2^2$ *and the upper bound of* $g(\hat{S}_1, \cdots, \hat{S}_K; \mathbf{o}_1, \cdots, \mathbf{o}_K)$ *in* ***Lemma 4***. *Then, the following inequality holds:*

$$\text{COST}(C_1, \cdots, C_K) \geq \frac{1}{8} \sum_{r \in H} \left[ \xi_r \zeta_{r,t} \min\{\mu(\hat{S}_r), \mu(\hat{S}_t)\} \right] - \omega, \quad (10)$$

*where $H$ is a subset of* $\{1, \cdots, K\}$; $t \in \{1, \cdots, K\}$; $\xi_r \geq 0$ *is a real number s.t.* $\sum_{r \in H} \xi_r \geq \epsilon$.

As highlighted in Fig. 1, the key idea to prove **Lemma 6** is to apply $\sum_{v_i \in \hat{S}_r} d_i \|\tilde{\mathbf{F}}_{i:} - \mathbf{w}_r\|_2^2 \geq \sum_{v_i \in \hat{S}_r \cap C_l} d_i \|\tilde{\mathbf{F}}_{i:} - \mathbf{w}_r\|_2^2$ to (2) and utilize properties of $\mu(C_l \Delta \hat{S}_{\pi(l)})$ (see the proof of Lemma 4 in (Mizutani, 2021)). By setting $\zeta_{r,t} = 2$ and $\omega = 4(1 + \lambda_1)^2 \alpha K \mu_{\max} \Psi_{\text{ISC}}$ according to the corresponding bounds in **Lemma 4**, we can derive the following **Theorem 7** based on **Lemma 6**.

**Theorem 7** *Suppose that the assumption of* ***Lemma 6*** *holds. Then, we have*

$$\text{COST}(C_1, \cdots, C_K) \geq \frac{1}{4} \epsilon \mu_{\min} - 4(1 + \lambda_1)^2 \alpha \mu_{\max} K \Psi_{\text{ISC}}, \quad (11)$$

*with* $\mu_{\min} := \min\{\mu(\hat{S}_r)\}$ *and* $\mu_{\max} := \max\{\mu(\hat{S}_r)\}$.

Finally, we obtain our main theoretical results based on **Theorems 5** and **7**.

**Theorem 8 (Main Theoretical Results)** *Given a graph $G$ and a pre-set number of clusters $K$, let* $(\hat{S}_1, \cdots, \hat{S}_K)$ *be a* $\bar{\phi}_K(G)$-***optimal*** *partition of* ***conductance minimization*** *and* $(C_1, \cdots, C_K)$ *be a feasible clustering result given by ISC. Assume that $K$Means has an approximation ratio of* $\alpha$. *If* $\Psi_{\text{ISC}} \leq 1/[132(1 + \lambda_1)^2 \alpha \tilde{\mu} K]$ *with* $\tilde{\mu} := \mu_{\max}/\mu_{\min}$, *after a suitable renumbering of* $(C_1, \cdots, C_K)$, *the following inequalities hold for $r \in \{1, \ldots, K\}$:*

$$\mu(C_r \Delta \hat{S}_r) \leq [66(1 + \lambda_1)^2 \alpha \tilde{\mu} K \Psi_{\text{ISC}}] \mu(\hat{S}_r), \text{ and}$$

$$\phi(C_r) \leq [1 + 132(1 + \lambda_1)^2 \alpha \tilde{\mu} K \Psi_{\text{ISC}}] \phi(\hat{S}_r) + 132(1 + \lambda_1)^2 \alpha \tilde{\mu} K \Psi_{\text{ISC}}.$$

As depicted in Fig. 1, one can prove **Theorem 8** by contradiction using the upper and lower bounds in **Theorems 5** and **7** (see Appendix D for the full proof). **Theorem 8** provides upper bounds for the **mis-clustered volume** $\mu(C_r \Delta \hat{S}_r)$ and **conductance** $\phi(C_r)$ w.r.t. the **optimal solution** $(\hat{S}_1, \cdots, \hat{S}_K)$ to **conductance minimization**. These bounds are directly proportional to $\Psi_{\mathrm{ISC}}$. *A graph with (i) higher degree heterogeneity and (ii) weaker clustering structures will cause a larger $\Psi_{\mathrm{ISC}}$ and thus lead to higher upper bounds.* Since $\mu(C_r \Delta \hat{S}_r)$ and $\phi(C_r)$ can be used to measure the clustering quality, *higher upper bounds indicate that ISC is more likely to achieve a low-quality result*. In this way, our analysis can quantitatively reveal impacts of (i) degree heterogeneity and (ii) weakness of clustering structures to the quality of DCSC.

To ensure that the condition in **Theorem 8** holds, one needs small $\bar{\phi}_K(G)/(1 - \lambda_{K+2})$ and large $d_{\min}/[(d_{\max} + \tau)(1 - \lambda_{K+2})]$. In some early studies on vanilla spectral clustering (Ng et al., 2001; Mizutani, 2021), a graph is defined to be **well-clustered**, if $\bar{\phi}_K(G)/(1 - \lambda_{K+1})$ is sufficiently small, consistent with that $\bar{\phi}_K(G)/(1 - \lambda_{K+2})$ is small ($\lambda_{K+2} \geq \lambda_{K+1}$). The well-clustered assumption adopted by early work (Ng et al., 2001; Mizutani, 2021) indicates that the optimal solution $(\hat{S}_1, \cdots, \hat{S}_K)$ describes an explicit clustering structure of $G$. Moreover, large $d_{\min}/[(d_{\max} + \tau)]$ indicates that the degree heterogeneity should not be very high. Therefore, the condition in **Theorem 8** implies an assumption that (i) $G$ is well-clustered and (ii) the degree heterogeneity is not so high. In particular, the adjustment of $\tau$ can also help resist the impacts of these two aspects. For instance, a larger $\tau$ can result in smaller eigenvalues $\lambda_1$ and $\lambda_{K+2}$, which further lead to smaller $\Psi_{\mathrm{ISC}}$ and larger $1/[132(1 + \lambda_1)^2 \alpha \tilde{\mu} K]$. The condition is more likely to satisfy.

## 4 EXTENSION OF DCSC: ASCENT

Inspired by recent advances in GNNs and the associated over-smoothing issue, we introduce AS-CENT, a simple yet effective extension of DCSC. Different from most DCSC methods with a constant correction $\tau$ for all nodes (e.g., *RSC*, *SCORE+*, and *ISC* in Table 1), ASCENT adopts a node-wise correction scheme. It can adaptively determine different corrections $\{\tau_i\}$ for nodes $\{v_i\}$ via an iterative aggregation mechanism that computes 'local' average degrees w.r.t. graph topology. Whereas, $\tau$ is usually set to be a 'global' average degree for existing DCSC algorithms (e.g., $\tau = \bar{d}$ for *RSC*). To the best of our knowledge, we are the first to explore an extension of DCSC with node-wise corrections.

Let $\boldsymbol{\tau}^{(l)} \in \mathbb{R}_+^N$ be the vector of node-wise corrections in the $l$-th iteration, with $\tau_i^{(l)}$ as the correction of node $v_i$. Suppose there are in total $L$ iterations, we obtain the node-wise corrections $\boldsymbol{\tau} \in \mathbb{R}_+^N$ of ASCENT via

$$\boldsymbol{\tau}^{(0)} = \mathbf{d}, \ \boldsymbol{\tau}^{(l)} := \hat{\mathbf{D}}^{-1} \hat{\mathbf{A}} \boldsymbol{\tau}^{(l-1)} \ (1 \leq l \leq L), \ \text{and} \ \boldsymbol{\tau} := \theta \boldsymbol{\tau}^{(L)}, \tag{12}$$

where $\hat{\mathbf{A}} := \mathbf{A} + \mathbf{I}_N$ is the adjacency matrix with self-edges; $\hat{\mathbf{D}}$ is the degree diagonal matrix w.r.t. $\hat{\mathbf{A}}$; $\theta > 0$ is a hyper-parameter. Concretely, we first let $\tau_i^{(0)} = d_i$ for initialization. Then, we iteratively update $\boldsymbol{\tau}^{(l)}$ using a typical **mean aggregation operation** of GNN (Kipf & Welling, 2016; Hamilton et al., 2017). Different from existing GNN-based graph clustering methods (Bianchi et al., 2020; Tsitsulin et al., 2023; Bhowmick et al., 2024), ASCENT does not rely on any graph attribute inputs and training procedures. Instead, it directly uses $\{\boldsymbol{\tau}^{(l)} \in \mathbb{R}_+^N\}$ as special features for aggregation. In each iteration, it computes the average correction value w.r.t. the one-hop neighbors for each node. We use $\tau_i = \theta \tau_i^{(L)}$ as the final correction value of node $v_i$. Similar to the role of $\delta$ in *RSC*, *SCORE+*, and *ISC* as summarized in Table 1, $\theta$ adjusts the scale of $\tau_i$. Furthermore, ASCENT adopts the same strategies of spectral embedding arrangement and normalization (i.e., steps (II) and (III) in Table 1) as *ISC*. Algorithm 1 summarizes the overall procedure of ASCENT.

Fig. 2 demonstrates our node-wise correction scheme on the **Karate Club** graph (Zachary, 1977) with 34 nodes and 2 clusters, where we visualize $\{\boldsymbol{\tau}^{(l)}\}$ in different iterations; each color denotes a cluster. Although different nodes have various initial values (i.e., node degrees) in $\boldsymbol{\tau}^{(0)}$, the aggregation operation in (12) *forces nodes in the same cluster (i.e., with more common high-order neighbors) to have close correction values*. For instance, in $\boldsymbol{\tau}^{(9)}$, $\boldsymbol{\tau}^{(10)}$, and $\boldsymbol{\tau}^{(11)}$, nodes in the first cluster tend to have larger corrections than those in the second cluster. It is well-known that most GNNs, especially those with a mean aggregator, suffer from the over-smoothing issue (Rusch et al., 2023), where node features converge to a constant as the number of layers increases. Similarly, the

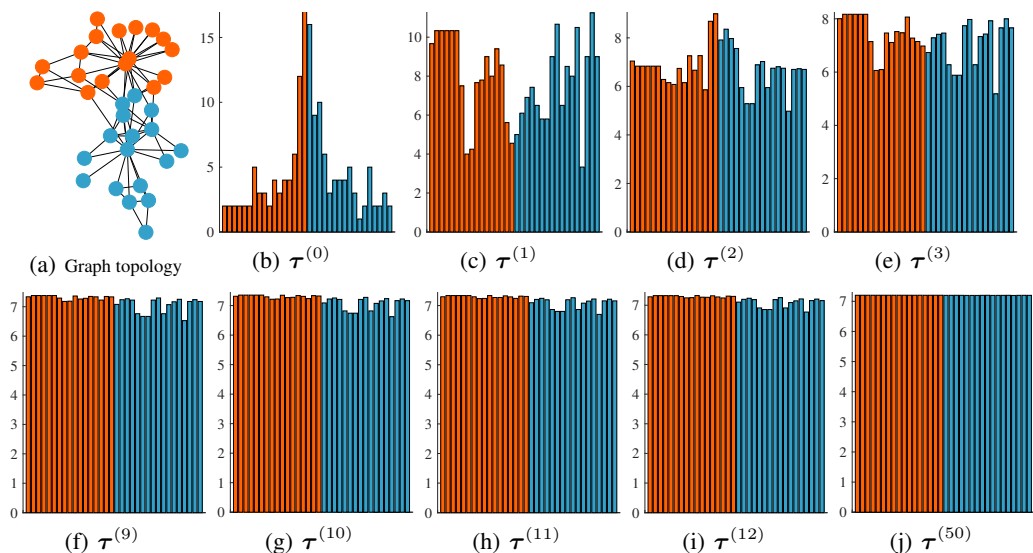

Figure 2: Case study of $\{\boldsymbol{\tau}^{(l)}\}$ on the **Karate Club** graph, where each color denotes a cluster.

node-wise corrections of ASCENT also converge to a constant for a large number of iterations $l$ (e.g., $\boldsymbol{\tau}^{(50)}$), due to the over-smoothing effect s.t. $\lim_{l \to \infty} \tau_i^{(l)} = c, \forall v_i \in V$, with $c$ as a constant. In this case, *ASCENT reduces to existing DCSC methods with a constant correction $\tau$*, corresponding to a 'global' average of node degrees. Our experiments further indicate that *ASCENT can potentially achieve better clustering quality in some early stages before over-smoothing* (e.g., with $L < 10$). It corresponds to a special 'local' average of node degrees.

We extend our analysis to the following **Proposition 9** regarding ASCENT (see Appendix E for the full proof). For each cluster $\hat{S}_r$, we introduce a cluster-wise correction $\hat{\tau}_r := \max\{\tau_i | v_i \in \hat{S}_r\}$. Since different nodes $\{v_i\}$ may have different $\{\tau_i\}$, different clusters $\{\hat{S}_r\}$ may have different $\{\hat{\tau}_r\}$, which can be used to demonstrate the advantages of node-wise corrections $\{\tau_i\}$ beyond conventional methods with a constant $\tau$. Based on **Proposition 9**, we further explore when can ASCENT potentially outperform *ISC* in Appendix F.

**Proposition 9** *Let* $\bar{\varphi}_K(G) := \sum_{r=1}^{K} \tilde{d}_r \phi(\hat{S}_r)/K$ *be a **reweighted conductance** w.r.t. the optimal partition* $(\hat{S}_1, \cdots, \hat{S}_K)$, *with* $\tilde{d}_r := d_{\min}/(d_{\max} + \hat{\tau}_r)$. *By replacing* $\Psi_{\mathrm{ISC}}$ *with* $\Psi_{\mathrm{AST}} := (1 + \lambda_{K+2})^{-1}[1 - (\hat{d} - \bar{\varphi}_K(G))]$, *where* $\hat{d} := \sum_{r=1}^{K} \tilde{d}_r/K$, ***Theorems 3** and **8** hold for **ASCENT**.*

## 5 EXPERIMENTS

### 5.1 EXPERIMENT SETUP

Table 3: Statistic details of datasets.

| Datasets | $N$ | $|E|$ | $K$ | min | max | avg $d$ |
|---|---|---|---|---|---|---|
| LFR-1 | 2,000 | 7,693-12,057 | 2-15 | 1-2 | 286-1,000 | 7-12 |
| LFR-2 | 2,000 | 8,489-32,202 | 4-14 | 2-6 | 375-1,000 | 8-32 |
| SBM-1 | 1,000 | 18,911-20,354 | 11 | 13-24 | 57-74 | 37-40 |
| SBM-2 | 1,000 | 16,368-20,202 | 11 | 12-24 | 51-72 | 32-40 |
| Caltech | 590 | 12,822 | 8 | 1 | 179 | 43.5 |
| Simmons | 1,137 | 24,257 | 4 | 1 | 293 | 42.7 |
| PolBlogs | 1,222 | 16,714 | 2 | 1 | 351 | 27.4 |
| BioGrid | 5,640 | 59,748 | 81 | 1 | 2570 | 21.2 |
| Airport | 3,158 | 18,605 | N/A | 1 | 246 | 11.8 |
| Wiki | 4,777 | 92,295 | N/A | 2 | 3,644 | 38 |
| BlogCatalog | 10,312 | 333,983 | N/A | 1 | 3,992 | 64 |
| ogbn-Protein | 132,534 | 39,561,252 | N/A | 1 | 7,750 | 597 |

Table 4: Summary of baselines.

| Baselines | Venues |
|---|---|
| NJW | NIPS 2001 |
| SCORE | Ann. Stat. 2015 |
| RSC | NIPS 2013 |
| SCORE+ | SankhyaA 2021 |
| ISC | arXiv 2020 |
| GraphEncoder (GE) | AAAI 2014 |
| GAP | arXiv 2019 |
| SDCN | WWW 2020 |
| MinCutPool (MCP) | ICML 2020 |
| DMoN | JMLR 2023 |
| DGCluster (DGC) | AAAI 2024 |

**Datasets**. We used 4 settings of synthetic benchmarks and 8 real graphs for evaluation. Table 3 summarizes statistics of these datasets, where $N$, $|E|$, and $K$ are the numbers of nodes, edges, and clusters (if available); $d$ denotes node degree.

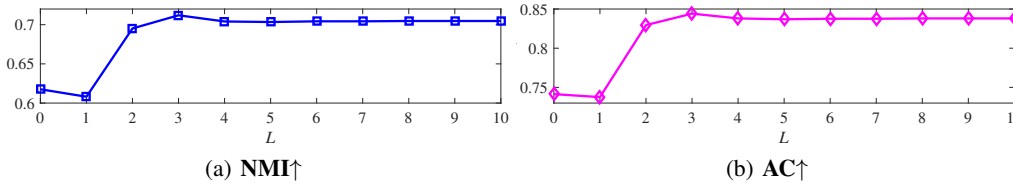

(a) **NMI↑**  (b) **AC↑**

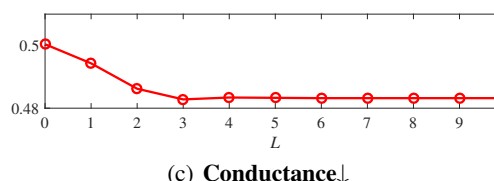

(c) **Conductance↓**

Figure 3: Parameter analysis of $L$ on **Caltech** in terms of **NMI↑**, **AC↑**, and **conductance↓**, where ASCENT achieves the best clustering quality with a small setting of $L$ (i.e., $L = 3$) before it reduces to conventional DCSC methods due to the over-smoothing issue.

Table 5: Synthetic graph analysis on LFR-net in terms of **NMI↑**, **AC↑**, and **conductance↓**.

| | LFR-1 | | | | | | | | | LFR-2 | | | | | | | | |
| | $\eta$=0.1 | | | 0.3 | | | 0.5 | | | $d$=10 | | | 20 | | | 30 | | |
| | NMI | AC | Cond | NMI | AC | Cond | NMI | AC | Cond | NMI | AC | Cond | NMI | AC | Cond | NMI | AC | Cond |
| | (↑,%) | (↑,%) | (↓,%) | (↑,%) | (↑,%) | (↓,%) | (↑,%) | (↑,%) | (↓,%) | (↑,%) | (↑,%) | (↓,%) | (↑,%) | (↑,%) | (↓,%) | (↑,%) | (↑,%) | (↓,%) |
|---|---|---|---|---|---|---|---|---|---|---|---|---|---|---|---|---|---|---|
| NJW | 87.64 | 96.90 | 10.04 | 56.90 | 77.06 | 34.89 | 24.91 | 43.93 | 57.12 | 26.27 | 44.32 | 57.10 | 66.58 | 79.15 | 53.18 | 81.47 | 90.33 | 51.37 |
| SCORE | 77.15 | 86.90 | 21.68 | 59.77 | 80.17 | 39.64 | 30.19 | 48.60 | 66.10 | 30.09 | 47.74 | 66.48 | 65.19 | 77.27 | 57.22 | 83.54 | 89.17 | 53.55 |
| RSC | 80.73 | 94.98 | 10.11 | 54.43 | 78.65 | 32.17 | 39.18 | 60.59 | 52.82 | 39.75 | 61.22 | 53.94 | 61.86 | 77.19 | 52.60 | 76.63 | 87.41 | 51.66 |
| SCORE+ | 78.99 | 91.88 | 12.75 | 60.19 | 84.03 | 32.32 | 43.66 | 65.38 | 53.94 | 44.72 | 66.30 | 53.49 | 71.81 | 85.38 | 52.22 | 85.64 | 93.61 | 50.99 |
| ISC | 88.01 | _97.23_ | _10.00_ | 65.27 | _86.59_ | 31.79 | 44.16 | _65.73_ | 52.31 | _44.77_ | _66.59_ | _52.87_ | 72.62 | 85.59 | 52.21 | 86.88 | _93.91_ | _50.97_ |
| GE | 61.70 | 81.73 | 73.49 | 38.94 | 60.69 | 79.02 | 20.60 | 34.98 | 87.18 | 21.37 | 36.21 | 87.59 | 38.89 | 54.09 | 86.13 | 54.24 | 68.11 | 84.22 |
| GAP | 2.93 | 41.30 | 66.69 | 2.02 | 35.03 | 73.87 | 0.79 | 24.36 | 84.01 | 0.23 | 23.68 | 85.02 | 0.36 | 25.27 | 83.05 | 0.35 | 25.98 | 81.85 |
| SDCN | 1.66 | 40.64 | 70.92 | 0.37 | 33.74 | 76.78 | 1.56 | 24.23 | 86.65 | 1.61 | 24.00 | 87.10 | 0.88 | 25.29 | 85.09 | 1.25 | 26.27 | 83.45 |
| MCP | 63.34 | 84.18 | 19.60 | 11.66 | 42.72 | 65.46 | 0.56 | 24.12 | 84.20 | 0.29 | 23.86 | 85.03 | 0.00 | 25.16 | 83.21 | 0.00 | 25.90 | 81.91 |
| DMoN | 80.64 | 93.41 | 11.92 | 55.36 | 76.08 | 36.38 | 30.29 | 47.46 | 57.45 | 29.70 | 46.06 | 59.16 | 49.43 | 61.52 | 60.48 | 56.68 | 67.24 | 59.51 |
| DGC | _88.47_ | 97.15 | 10.02 | _67.79_ | 85.82 | _31.82_ | _45.12_ | 64.44 | _52.66_ | 44.67 | 63.90 | 53.06 | _72.33_ | _85.56_ | _51.69_ | _85.92_ | 89.48 | 52.12 |
| **ASCENT** | **89.38** | **97.60** | **9.95** | **68.83** | **88.73** | **31.74** | **46.42** | **67.80** | **52.01** | **46.45** | **68.51** | **51.72** | **77.53** | **88.84** | **51.14** | **89.81** | **95.44** | **50.57** |
| Improv.(%) | +1.03 | +0.38 | +0.5 | +1.53 | +2.47 | +0.16 | +2.88 | +3.15 | +0.57 | +3.75 | +2.88 | +2.18 | +6.76 | +3.80 | +1.06 | +3.37 | +1.63 | +0.78 |

LFR-net (Lancichinetti et al., 2008) is a synthetic benchmark that can simulate various properties of real-world graphs. It uses $(\bar{d}, d_{\max}, c_{\min}, c_{\max}, \eta)$ to generate a graph, where $\bar{d}$ and $d_{\max}$ are the average and maximum degree; $c_{\min}$ and $c_{\max}$ are the minimum and maximum cluster size; $\eta$ is the ratio between the external degree and total degree of a node $v_i$ w.r.t. the cluster that $v_i$ belongs to. To test the ability to handle (i) weak clustering structures and (ii) high degree heterogeneity, we used LFR-net to generate two sets of graphs, denoted as **LFR-1** and **LFR-2**. For **LFR-1**, we fixed $(N, \bar{d}, d_{\max}, c_{\min}, c_{\max}) = (2000, 50, 1000, 50, 500)$ and adjusted $\eta \in \{0.1, 0.3, 0.5\}$. *With the **increase** of $\eta$, **clustering structures** are **increasingly difficult** to identify (i.e., **weaker clustering structures**)*. For **LFR-2**, we fixed $(N, d_{\max}, c_{\min}, c_{\max}, \eta) = (2000, 1000, 50, 500, 0.5)$ and set $\bar{d} \in \{10, 20, 30\}$, where ***lower $\bar{d}$ indicates **higher degree heterogeneity***.

We also used the SBM generator (Kao et al., 2017) implemented by `graph-tool`[2] to simulate the two cases about (i) weak clustering structures and (ii) high degree heterogeneity, which are denoted as **SBM-1** and **SBM-2**. The generator uses $(\gamma, \beta, \rho)$ to generate a graph, where $\gamma$ is the ratio of between the number of within- and between-cluster edges; $\beta$ controls the power-law distribution of node degrees; $\rho$ adjusts the heterogeneity of community size. For **SBM-1**, we fixed $(N, \rho, \beta) = (1000, 1, 2.5)$ and set $\gamma \in \{0.5, 0.6, 0.7\}$. *With the **increase** of $\gamma$, the **clustering structures** are **increasingly easy** to identify.* For **SBM-2**, we fixed $(N, \rho, \gamma) = (1000, 1, 0.5)$ and set $\beta \in \{2.5, 2.75, 3\}$, where ***larger $\beta$ implies **higher degree heterogeneity***.

**Caltech** (Red et al., 2011), **Simmons** (Red et al., 2011), **PolBlogs** (Adamic & Glance, 2005), and **BioGrid** (Stark et al., 2006) are real datasets with explicit ground-truth for graph clustering. In contrast, **Airport** (Chami et al., 2019), **Wiki** (Grover & Leskovec, 2016), **BlogCatalog** (Grover & Leskovec, 2016), and **obgn-Protein** (Szklarczyk et al., 2019) are real datasets that do not provide ground-truth w.r.t. our problem statements in Section 2. Due to space limit, we leave details regarding these real datasets in Appendix H.

**Baselines**. As summarized in Table 4, we compare ASCENT over 11 baselines published from 2001 to 2024, which can be divided into two categories. First, (i) *NJW* (Ng et al., 2001), (ii) *SCORE* (Jin, 2015), (iii) *RSC* (Qin & Rohe, 2013), (iv) *SCORE+* (Jin et al., 2021), and (v) *ISC* (Qing & Wang, 2020a) are representative spectral clustering methods. Second, (vi) *GraphEncoder* (Tian et al., 2014), (vii) *GAP* (Nazi et al., 2019), (viii) *SDCN* (Bo et al., 2020), (ix) *MinCutPool* (Bianchi

---

[2]https://graph-tool.skewed.de/

Table 6: Synthetic graph analysis on SBM in terms of **NMI**↑, **AC**↑, and **conductance**↓.

| | SBM-1 | | | | | | | | | SBM-2 | | | | | | | | |
| | $\gamma$=0.5 | | | 0.6 | | | 0.7 | | | $\beta$=2.5 | | | 2.75 | | | 3 | | |
| | NMI | AC | Cond | NMI | AC | Cond | NMI | AC | Cond | NMI | AC | Cond | NMI | AC | Cond | NMI | AC | Cond |
| | (↑,%) | (↑,%) | (↓,%) | (↑,%) | (↑,%) | (↓,%) | (↑,%) | (↑,%) | (↓,%) | (↑,%) | (↑,%) | (↓,%) | (↑,%) | (↑,%) | (↓,%) | (↑,%) | (↑,%) | (↓,%) |
| NJW | 79.94 | 87.74 | 70.54 | 91.81 | 95.35 | 66.21 | 96.48 | 98.16 | 62.80 | 80.20 | 87.82 | 70.59 | 76.16 | 85.23 | 70.59 | 73.34 | 83.36 | 70.56 |
| SCORE | 72.03 | 78.86 | 71.60 | 85.58 | 88.55 | 67.11 | 91.04 | 92.62 | 63.73 | 72.39 | 79.27 | 71.63 | 68.46 | 75.77 | 71.60 | 65.02 | 73.66 | 71.65 |
| RSC | 76.89 | 84.03 | 70.97 | 89.76 | 92.98 | 66.52 | 94.47 | 96.14 | 63.18 | 77.09 | 83.75 | 71.01 | 73.35 | 80.70 | 71.00 | 70.38 | 79.66 | 71.02 |
| SCORE+ | 82.08 | 89.18 | 70.33 | 92.99 | 96.10 | 66.08 | 97.02 | 98.53 | 62.72 | 82.06 | 89.10 | 70.39 | 78.29 | 86.61 | 70.42 | 75.36 | 84.54 | _70.35_ |
| ISC | _82.33_ | _89.62_ | _70.32_ | _93.18_ | _96.18_ | _66.06_ | _97.32_ | _98.66_ | _62.70_ | _82.46_ | _89.54_ | _70.38_ | _78.80_ | _87.05_ | _70.39_ | _75.70_ | _85.21_ | _70.35_ |
| GE | 66.79 | 76.28 | 74.70 | 86.12 | 90.70 | 67.80 | 93.42 | 95.41 | 63.77 | 67.68 | 77.03 | 74.61 | 62.61 | 73.19 | 75.42 | 57.20 | 68.68 | 76.39 |
| GAP | 1.27 | 15.47 | 90.72 | 1.65 | 15.60 | 90.66 | 1.76 | 15.50 | 90.66 | 1.37 | 15.11 | 90.69 | 1.18 | 15.29 | 90.71 | 1.10 | 15.07 | 90.71 |
| SDCN | 26.01 | 35.09 | 85.54 | 39.13 | 47.31 | 81.20 | 48.02 | 54.58 | 77.74 | 26.29 | 36.80 | 85.40 | 22.26 | 32.87 | 86.32 | 18.99 | 29.62 | 87.16 |
| MCP | 0.00 | 14.81 | 90.91 | 0.00 | 14.81 | 90.91 | 0.00 | 14.81 | 90.91 | 0.00 | 14.81 | 90.91 | 0.00 | 14.81 | 90.91 | 0.00 | 14.81 | 90.91 |
| DMoN | 76.25 | 79.33 | 72.14 | 84.96 | 85.07 | 68.93 | 88.84 | 87.52 | 65.99 | 75.94 | 79.17 | 72.21 | 73.42 | 77.81 | 71.86 | 69.55 | 74.98 | 71.75 |
| DGC | 68.02 | 67.92 | 77.91 | 80.19 | 76.14 | 73.44 | 87.21 | 81.50 | 69.57 | 68.23 | 69.30 | 77.59 | 66.94 | 68.42 | 77.60 | 64.90 | 67.89 | 77.45 |
| **ASCENT** | **82.52** | **90.08** | **70.27** | **93.41** | **96.51** | **66.02** | **97.38** | **98.68** | **62.68** | **82.74** | **89.87** | **70.32** | **79.00** | **87.51** | **70.33** | **76.05** | **85.79** | **70.29** |
| Improv.(%) | +0.23 | +0.51 | +0.02 | +0.24 | +0.34 | +0.06 | +0.06 | +0.02 | +0.03 | +0.34 | +0.37 | +0.09 | +0.25 | +0.53 | +0.09 | +0.46 | +0.68 | +0.09 |

Table 7: Evaluation results on datasets with ground-truth in terms of **NMI**↑, **AC**↑, & **conductance**↓.

| | Caltech | | | Simmons | | | PolBlogs | | | BioGrid | | |
| | NMI | AC | Cond | NMI | AC | Cond | NMI | AC | Cond | NMI | AC | Cond |
| | (↑,%) | (↑,%) | (↓,%) | (↑,%) | (↑,%) | (↓,%) | (↑,%) | (↑,%) | (↓,%) | (↑,%) | (↑,%) | (↓,%) |
| NJW | 62.13 | 75.39 | 50.76 | 67.96 | 73.44 | 33.87 | 0.06 | 51.88 | 26.94 | 41.83 | 12.84 | 66.20 |
| SCORE | 56.39 | 69.05 | 50.12 | 58.53 | 76.39 | 29.92 | 72.50 | 95.25 | 7.67 | 13.93 | 7.37 | 90.90 |
| RSC | 58.58 | 71.05 | 49.86 | 61.52 | 78.61 | 28.88 | 71.33 | 94.76 | _7.34_ | **43.64** | _13.52_ | _66.07_ |
| SCORE+ | 69.14 | 82.85 | 48.44 | 72.95 | 88.81 | 27.41 | _73.08_ | _95.33_ | 7.53 | 24.36 | 9.44 | 82.02 |
| ISC | _70.28_ | _83.73_ | _48.32_ | _73.57_ | _89.36_ | _27.35_ | 72.67 | 95.09 | 7.35 | 43.21 | 13.26 | _66.07_ |
| GE | 36.75 | 44.44 | 68.97 | 49.19 | 58.86 | 48.13 | 6.15 | 51.88 | 50.13 | 31.74 | 11.28 | 98.51 |
| GAP | 65.80 | 75.59 | 49.94 | 48.69 | 57.96 | 40.84 | 42.89 | 77.82 | 2.44 | 44.18 | 13.40 | 80.40 |
| SDCN | 28.50 | 34.03 | 74.79 | 38.28 | 53.91 | 51.39 | 14.96 | 62.93 | 28.48 | 20.77 | 8.44 | 95.83 |
| MCP | 50.57 | 63.32 | 58.16 | 64.66 | 82.90 | 29.80 | 58.15 | 86.56 | 16.26 | 0.00 | 4.43 | 98.77 |
| DMoN | 66.29 | 72.47 | 53.97 | 63.64 | 81.20 | 28.00 | 71.16 | 94.91 | 7.47 | 41.73 | 12.91 | 79.74 |
| DGC | 66.75 | 75.32 | 51.92 | 70.53 | 79.74 | 33.18 | 71.21 | 94.91 | 7.42 | 21.63 | 7.74 | 91.04 |
| **ASCENT** | **71.20** | **84.41** | **48.28** | **74.06** | **89.62** | **27.34** | **73.48** | **95.34** | **7.31** | _43.28_ | **13.59** | **64.88** |
| Improvement | +1.31% | +0.81% | +0.08% | +0.67% | +0.29% | +0.04% | +0.55% | +0.01% | +0.41% | – | +0.52% | +1.80% |

et al., 2020), (x) *DMoN* (Tsitsulin et al., 2023), and (xi) *DGCluster* (Bhowmick et al., 2024) are deep graph clustering approaches.

*RSC*, *SCORE+*, and *ISC* are DCSC baselines as summarized in Table 1. Note that we consider graph clustering without attributes in this study, while *GAP*, *SDCN*, *MinCutPool*, *DMoN*, and *DGCluster* are GNN-based methods originally designed for attributed graphs. We tried several widely-used strategies of (i) SVD on the adjacency matrix and (ii) one-hot encoding of node degrees to derive feature inputs for *GAP*, *MinCutPool*, *DMoN*, and *DGCluster*, with the best quality metrics reported. Moreover, we directly used the adjacency matrix as input of the auto-encoder in *SDCN*.

**Evaluation Metrics**. For datasets with ground-truth, we used **normalized mutual information** (**NMI**) and **accuracy** (**AC**) as quality metrics. We also adopted the **conductance** achieved by each method as an unsupervised metric for all the datasets. For datasets without ground-truth, we set $K \in \{2, 8, 32\}$ and recorded the corresponding conductance values. Usually, *smaller conductance as well as larger NMI and AC indicate better clustering quality*. We tuned hyper-parameters of all the methods based on the unsupervised conductance metric. Due to space limit, we detail other experiment setups (e.g., experiment environment and parameter settings) in Appendix H.

## 5.2 PARAMETER ANALYSIS

We first tested the effect of $L$ for ASCENT. Example analysis results on **Caltech** are visualized in Fig. 3, where we adjusted $L \in \{0, 1, \cdots, 10\}$. When $L = 0$, ASCENT suffers from poor clustering quality, which can be significantly improved as $L$ increases. It validates the effectiveness of the iterative aggregation in ASCENT. With the increase of $L$, the clustering quality of ASCENT gradually converges due to over-smoothing, which is consistent with our case study in Fig. 2. In particular, *ASCENT achieves the best clustering quality with a small L (i.e., $L = 3$) before it reduces to conventional DCSC methods with a constant correction $\tau$*. ASCENT also achieves the best quality with a small $L$ in the following evaluation. We leave further analysis of $\theta$ in Appendix I.

## 5.3 SYNTHETIC GRAPH ANALYSIS

For each setting of a synthetic benchmark, we independently generated 100 graphs and recorded the mean as well as standard derivation of all the quality metrics over these graphs. The average

Table 8: Evaluation results on datasets without ground-truth in terms of **conductance** (%)↓.

| | Airport | | | Wiki | | | BlogCatalog | | | ogbn-Protein | | |
|---|---|---|---|---|---|---|---|---|---|---|---|---|
| | $K$=2 | 8 | 32 | $K$=2 | 8 | 32 | $K$=2 | 8 | 32 | $K$=2 | 8 | 32 |
| NJW | **1.67** | 10.16 | 19.48 | 40.16 | 74.32 | 85.32 | 29.35 | 67.83 | 82.79 | 6.34 | 11.92 | 41.36 |
| SCORE | 10.95 | 74.98 | 85.82 | 40.29 | 82.73 | 92.08 | 29.65 | 77.84 | 93.67 | 22.38 | 44.69 | 82.42 |
| RSC | 6.09 | 23.44 | 37.11 | 37.75 | 73.06 | 85.32 | 29.24 | 66.56 | 81.74 | 12.03 | 15.87 | 36.63 |
| SCORE+ | 5.19 | 52.48 | 71.47 | 39.06 | 75.69 | 86.87 | 29.33 | 69.95 | 86.19 | 7.09 | 21.92 | 64.33 |
| ISC | 4.24 | 21.70 | 35.04 | 37.91 | 73.31 | 85.78 | 29.26 | 65.02 | 81.43 | 6.93 | 14.88 | 34.64 |
| GE | 48.14 | 84.27 | 92.58 | 52.08 | 89.02 | 97.23 | 49.59 | 87.49 | 96.78 | OOM | | |
| GAP | 32.00 | 87.50 | 96.88 | 44.66 | 87.50 | 96.88 | 41.81 | 87.50 | 96.88 | OOT | | |
| SDCN | 17.83 | 86.06 | 92.73 | 50.00 | 87.30 | 96.41 | 49.87 | 85.44 | 95.80 | OOM | | |
| MCP | 8.80 | 23.80 | 54.33 | 50.00 | 87.50 | 96.88 | 34.15 | 86.97 | 96.82 | OOM | | |
| DMoN | 6.44 | 20.40 | 51.00 | 37.09 | 77.25 | 91.59 | 42.89 | 75.95 | 90.28 | OOM | | |
| DGC | 5.75 | 13.54 | 69.80 | 37.45 | 83.53 | 95.96 | 29.89 | 77.73 | 94.34 | OOM | | |
| **ASCENT** | **1.67** | **9.97** | **18.85** | **37.43** | **72.00** | **84.77** | **29.23** | **64.22** | **80.68** | **3.47** | **11.06** | **33.20** |
| Improvement | – | +1.87% | +3.23% | +0.85% | +1.45% | +0.64% | +0.10% | +1.23% | +0.92% | +45.27% | +7.21% | +4.16% |

evaluation results on **LFR-1**, **LFR-2**, **SBM-1**, and **SBM-2** are depicted in Tables 5 and 6 (see the corresponding standard derivations in Appendix I), where a metric is in **bold** or underlined if it performs the best or second-best. In most cases, spectral clustering methods have significantly better quality than deep clustering baselines. It indicates that some GNN-based approaches, with standard strategies to extract auxiliary feature inputs from topology, may fail to handle the high degree heterogeneity and weak clustering structures, although some of them are claimed to be effective in the clustering on attributed graphs. Moreover, DCSC methods (i.e., *SCORE+*, *ISC*, and ASCENT) are always in groups with the best quality, which validates the robustness of DCSC over vanilla spectral clustering. In particular, ASCENT performs the best in most cases. In summary, ASCENT, which serves as a simple yet effective extension of DCSC, is more powerful in handling the high degree heterogeneity and weak clustering structures of graphs.

## 5.4 REAL GRAPH EVALUATION

On each real dataset, we repeated the evaluation procedure over 5 random seeds and recorded the mean as well as standard derivation of each metric. The average evaluation results on real datasets are reported in Tables 7 and 8 (see corresponding standard derivations in Appendix I), where a metric is in **bold** or underlined if it performs the best or second-best; OOM denotes the *out-of-memory* exception. We define that a method encounters the *out-of-time* (OOT) exception if it cannot derive a feasible result within $10^4$ seconds. Consistent with our synthetic graph analysis, spectral clustering methods significantly outperform deep clustering baselines in most cases. In particular, some deep clustering approaches encounter OOM or OOT exceptions on large-scale graphs (e.g., ogbn-Protein), due to the reconstruction of an $N \times N$ matrix (e.g., normalized adjacency matrices in *GraphEncoder*) or time/space-consuming training procedures. In contrast, spectral clustering methods can derive feasible clustering results on all the datasets. In most cases, ASCENT performs the best and can achieve much better quality than other DCSC baselines (i.e., *RSC*, *SCORE+*, and *ISC*). It further validates the effectiveness of ASCENT as an extension of DCSC.

## 6 CONCLUSION

In this paper, we provided an alternative analysis of DCSC from a pure spectral view. Different from most existing studies on DCSC that gave theoretical results associated with random graph models (e.g., DCSBM), our analysis gives bounds for the mis-clustered volume and conductance w.r.t. the optimal solution to conductance minimization objective without using random graph models. In contrast to early studies on vanilla spectral clustering, the presented analysis also includes quantities that indicate impacts of (i) degree heterogeneity and (ii) weakness of clustering structures to the clustering quality of DCSC. Inspired by recent advances in GNNs and the associated over-smoothing issue, we proposed ASCENT, a simple yet effective extension of DCSC. It follows a novel node-wise correction scheme that assigns nodes $\{v_i\}$ with different correction terms $\{\tau_i\}$ via the mean aggregation of GNNs. We further demonstrated that ASCENT reduces to conventional DCSC methods when encountering the over-smoothing issue. Experiments also validated that some early stages before over-smoothing can potentially obtain better clustering quality for ASCENT. Due to space limit, we discuss limitations and possible future directions of this study in Appendix J.

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

---

**Algorithm 1:** Proposed **ASCENT** Algorithm

---

**Input:** graph $G = (V, E)$, number of clusters $K$, hyper-parameters $\{\theta, L\}$
**Output:** a feasible clustering result $(C_1, \cdots, C_K)$

1 **for each** *node* $v_i \in V$ **do**
2    $\tau_i^{(0)} \leftarrow d_i$ //Initialize node-wise corrections
3 **for** $l$ **from** 1 **to** $L$ **do**
4    $\boldsymbol{\tau}^{(l)} \leftarrow \hat{\mathbf{D}}^{-1}\hat{\mathbf{A}}\boldsymbol{\tau}^{(l-1)}$ //Iteratively update node-wise corrections
5 $\boldsymbol{\tau} \leftarrow \theta\boldsymbol{\tau}^{(L)}$ //Final node-wise corrections
6 $\mathbf{L}_\tau \leftarrow (\mathbf{D} + \mathrm{diag}(\boldsymbol{\tau}))^{-1/2}\mathbf{A}(\mathbf{D} + \mathrm{diag}(\boldsymbol{\tau}))^{-1/2}$
7 Find the leading $(K+1)$ eigenvalues $(\lambda_1, \cdots, \lambda_{K+1})$ and eigenvectors $(\mathbf{u}_1, \cdots, \mathbf{u}_{K+1})$ of $\mathbf{L}_\tau$
8 $\mathbf{F} \leftarrow [\lambda_1\mathbf{u}_1, \cdots, \lambda_{K+1}\mathbf{u}_{K+1}]$
9 **for each** *node* $v_i \in V$ **do**
10    $\tilde{\mathbf{F}}_{i,:} \leftarrow \mathbf{F}_{i,:}/|\mathbf{F}_{i,:}|_2$

11 apply $K$Means to rows of $\tilde{\mathbf{F}}$ to get the clustering result $(C_1, \cdots, C_K)$

---

Yilin Zhang and Karl Rohe. Understanding regularized spectral clustering via graph conductance. *Advances in Neural Information Processing Systems*, 31, 2018.

David Zhuzhunashvili and Andrew Knyazev. Preconditioned spectral clustering for stochastic block partition streaming graph challenge (preliminary version at arxiv.). In *Proceedings of the 2017 IEEE High Performance Extreme Computing Conference (HPEC)*, pp. 1–6. IEEE, 2017.

## A    RELATED WORK OF DEEP GRAPH CLUSTERING METHODS

In recent years, several deep graph clustering methods have been proposed based on different model architectures and training objectives as reviewed in (Yue et al., 2022; Su et al., 2022).

*GraphEncoder* (Tian et al., 2014) and *DNR* (Yang et al., 2016) are early studies that learn low-dimensional community-preserving representations (a.k.a. embeddings) by reconstructing topology-related features (e.g., normalized adjacency matrices and modularity matrices) via a deep auto-encoder. A downstream clustering algorithm (e.g., $K$Means) is then applied to the learned embedding to derive a feasible clustering result. SDCN (Bo et al., 2020) further combines deep auto-encoder with GNN and uses a dual self-supervised mechanism to unify these two deep architectures. Moreover, *GAP* (Nazi et al., 2019), *ClusterNet* (Wilder et al., 2019), *MinCutPool* (Bianchi et al., 2020), and *DMoN* (Tsitsulin et al., 2023) adopt a deep end-to-end structure, which contains a GNN and an output module (e.g., a multi-layer perceptron for the derivation of clustering results), to fit some classic graph clustering objectives (e.g., normalized cut minimization (Von Luxburg, 2007) and modularity maximization (Newman, 2006)). *DGCluster* (Bhowmick et al., 2024) also uses the modularity maximization objective to optimize GNN, which outputs community-preserving embedding but derives final clustering results using BIRCH (Zhang et al., 1996).

Most of the aforementioned methods, especially those based on GNNs, were originally designed for attributed graphs. We argue that most of them do not consider the complicated correlations between graph topology and attributes as discussed in Section 2. Our empirical experiments (see Section 5) also demonstrated that when attributes are unavailable, these deep graph clustering (with standard strategies to extract auxiliary feature inputs from topology) cannot effectively handle the (i) high degree heterogeneity and (ii) weak clustering structures of graphs. Different from spectral clustering methods, most existing studies about deep graph clustering also lack interpretability and theoretical guarantees.

## B    PROOF OF THEOREM 3

Recall that we have $\mathbf{F} := [\lambda_1\mathbf{u}_1, \cdots, \lambda_{K+1}\mathbf{u}_{K+1}]$ as the rearranged spectral embedding of *ISC*; $\mathbf{G} \in \mathbb{R}^{N \times K}$ encodes membership of the optimal partition $(\hat{S}_1, \cdots, \hat{S}_K)$, where $\mathbf{G}_{ir} = \sqrt{d_i/\mu(\hat{S}_r)}$ if $v_i \in \hat{S}_r$ and $\mathbf{G}_{ir} = 0$ otherwise. For simplicity, we let $\mathbf{U} := [\mathbf{u}_1, \cdots, \mathbf{u}_{K+1}]$ be the arrangement

of eigenvectors without reweighting and $\mathbf{F} = \mathbf{U}\mathbf{\Lambda}$, where $\mathbf{\Lambda} := \mathrm{diag}(\lambda_1, \cdots, \lambda_{K+1})$ is a diagonal matrix w.r.t the leading $(K+1)$ eigenvalues.

For each $r \in \{1, \cdots, K\}$, it is obvious that $|\mathbf{G}_{:,r}|_2 = 1$. In particular, one can derive $\mathbf{G}_{:,r}$ via the linear combination of eigenvectors $\{\mathbf{u}_1, \cdots, \mathbf{u}_N\}$ w.r.t. eigenvalues $\{\lambda_1, \cdots, \lambda_N\}$ of regularized graph Laplacian $\mathbf{L}_\tau$ of *ISC*. Namely, we have $\mathbf{G}_{:,r} = \sum_{i=1}^{N} h_{ir}\lambda_i \mathbf{u}_i$ with $\{h_{ir}\}$ as corresponding weights for the linear combination. We further let $\hat{\mathbf{u}}_r := \sum_{i=1}^{K+1} h_{ir}\lambda_i \mathbf{u}_i$. Then, we have the following derivation:

$$
\begin{aligned}
\mathbf{G}_{:,r}^T(\mathbf{I}_N - \mathbf{L}_\tau)\mathbf{G}_{:,r} &= \sum_{i=1}^{N} \mathbf{G}_{ir}^2 - 2\sum_{(v_i,v_j)\in E} \frac{\mathbf{G}_{ir}\mathbf{G}_{jr}}{\sqrt{d_i+\tau}\sqrt{d_j+\tau}} \\
&= \sum_{(v_i,v_j)\in E} \left[\left(\frac{1}{\sqrt{d_i}}\mathbf{G}_{ir}\right)^2 - \frac{2\mathbf{G}_{ir}\mathbf{G}_{jr}}{\sqrt{d_i+\tau}\sqrt{d_j+\tau}} + \left(\frac{1}{\sqrt{d_j}}\mathbf{G}_{jr}\right)^2\right] \\
&= \sum_{(v_i,v_j)\in E(\hat{S}_r, V\backslash\hat{S}_r)} \frac{1}{\mu(\hat{S}_r)} + \sum_{(v_i,v_j)\in E(\hat{S}_r)} \frac{2}{\mu(\hat{S}_r)}\left(1 - \frac{\sqrt{d_id_j}}{\sqrt{d_i+\tau}\sqrt{d_j+\tau}}\right) \\
&\leq \frac{|E(\hat{S}_r, V\backslash\hat{S}_r)|}{\mu(\hat{S}_r)} + \frac{2|E(\hat{S}_r)|}{\mu(\hat{S}_r)}\left(1 - \frac{d_{\min}}{d_{\max}+\tau}\right) \\
&= \phi(\hat{S}_r) + \frac{\mu(\hat{S}_r) - |E(\hat{S}_r, V\backslash\hat{S}_r)|}{\mu(\hat{S}_r)}\left(1 - \frac{d_{\min}}{d_{\max}+\tau}\right) \\
&= 1 - (1 - \phi(\hat{S}_r))\frac{d_{\min}}{d_{\max}+\tau}.
\end{aligned}
$$

Note that we also have

$$
\begin{aligned}
\mathbf{G}_{:,r}^T(\mathbf{I}_N - \mathbf{L}_\tau)\mathbf{G}_{:,r} &= \left(\sum_{i=1}^{N} h_{ir}\lambda_i \mathbf{u}_i\right)^T(\mathbf{I}_N - \mathbf{L}_\tau)\left(\sum_{i=1}^{N} h_{ir}\lambda_i \mathbf{u}_i\right) \\
&= \sum_{i=1}^{N} h_{ir}^2\lambda_i^2 - \sum_{i=1}^{N} h_{ir}^2\lambda_i^3 \\
&= \sum_{i=1}^{N} h_{ir}^2\lambda_i^2(1 - \lambda_i) \\
&\geq \sum_{i=K+2}^{N} h_{ir}^2\lambda_i^2(1 - \lambda_i) \\
&\geq (1 - \lambda_{K+2})\sum_{i=K+2}^{N} h_{ir}^2\lambda_i^2.
\end{aligned}
$$

By combining the aforementioned two inequalities, one can have

$$
\|\hat{\mathbf{u}}_r - \mathbf{G}_{:,r}\|_2^2 = \sum_{i=K+2}^{N} h_{ir}^2\lambda_i^2 \leq \frac{1}{1-\lambda_{K+2}}\left[1 - (1-\phi(\hat{S}_r))\frac{d_{\min}}{d_{\max}+\tau}\right],
$$

for each $\hat{S}_r$ ($r \in \{1, \cdots, K\}$). Let $\hat{\mathbf{F}} := [\hat{\mathbf{u}}_1, \cdots, \hat{\mathbf{u}}_K]$. For the whole graph $G$, we have

$$
\|\hat{\mathbf{F}} - \mathbf{G}\|_F^2 = \sum_{r=1}^{K} \|\hat{\mathbf{u}}_r - \mathbf{G}_{:,r}\|_2^2 \leq \frac{K}{1-\lambda_{K+2}}\left[1 - (1-\bar{\phi}_K(G))\frac{d_{\min}}{d_{\max}+\tau}\right] = K\Psi_{\mathrm{ISC}}. \quad (13)
$$

This inequality can be rewritten as

$$
\|\hat{\mathbf{F}} - \mathbf{G}\|_F^2 = \|\mathbf{U}\mathbf{\Lambda}\mathbf{H} - \mathbf{G}\|_F^2 \leq K\Psi_{\mathrm{ISC}}, \quad (14)
$$

where $\mathbf{H} \in \mathbb{R}^{(K+1)\times K}$ is a matrix rearranging weights $\{h_{ir}\}$ in the linear combination.

**Claim.** Let $\mathbf{Z} := \hat{\mathbf{F}} - \mathbf{G} = [\mathbf{z}_1, \cdots, \mathbf{z}_N]$. Then, $\mathbf{Z}^T\mathbf{Z} = \mathbf{I}_K - \mathbf{H}^T\mathbf{\Lambda}^T\mathbf{\Lambda}\mathbf{H}$.

**Proof.** First, we have

$$\mathbf{z}_r^T\mathbf{z}_s = (\hat{\mathbf{u}}_r - \mathbf{G}_{:,r})^T(\hat{\mathbf{u}}_s - \mathbf{G}_{:,s}) = \hat{\mathbf{u}}_r^T\hat{\mathbf{u}}_s - \hat{\mathbf{u}}_r^T\mathbf{G}_{:,s} - \mathbf{G}_{:,r}^T\hat{\mathbf{u}}_s + \mathbf{G}_{:,r}^T\mathbf{G}_{:,s}. \quad (15)$$

Consider each term in (15). We further have

$$\hat{\mathbf{u}}_r^T\hat{\mathbf{u}}_s = (\sum_{t=1}^{K+1} h_{rt}\lambda_t\mathbf{u}_t)^T(\sum_{t=1}^{K+1} h_{st}\lambda_t\mathbf{u}_t) = \sum_{t=1}^{K+1} h_{rt}h_{st}\lambda_t^2;$$

$$\hat{\mathbf{u}}_r^T\mathbf{G}_{:,s} = (\sum_{t=1}^{K+1} h_{rt}\lambda_t\mathbf{u}_t)^T(\sum_{t=1}^{N} h_{st}\lambda_t\mathbf{u}_t)^T = \sum_{t=1}^{K+1} h_{rt}h_{st}\lambda_t^2;$$

$$\mathbf{G}_{:,r}^T\hat{\mathbf{u}}_s = (\sum_{t=1}^{N} h_{rt}\lambda_t\mathbf{u}_t)^T(\sum_{t=1}^{K+1} h_{st}\lambda_t\mathbf{u}_t)^T = \sum_{t=1}^{K+1} h_{rt}h_{st}\lambda_t^2;$$

$$\mathbf{G}_{:,r}^T\mathbf{G}_{:,s} = \left\{ \begin{array}{l} 0, r = s \\ 1, r \neq s \end{array} \right. .$$

Therefore, one can rewrite (15) as

$$\mathbf{z}_r^T\mathbf{z}_s = \left\{ \begin{array}{l} 1 - \sum_{t=1}^{K+1} h_{rt}h_{st}\lambda_t^2, \ r = s \\ -\sum_{t=1}^{K+1} h_{rt}h_{st}\lambda_t^2, \quad r \neq s \end{array} \right. ,$$

which corresponds to the following matrix form:

$$\mathbf{Z}^T\mathbf{Z} = \mathbf{I}_K - \mathbf{H}^T\mathbf{\Lambda}^T\mathbf{\Lambda}\mathbf{H}.$$

This completes the proof of **Claim**.

Consider the singular value decomposition of $\mathbf{H} \in \mathbb{R}^{(K+1)\times K}$ denoted as $\mathbf{H} = \mathbf{X}\mathbf{\Sigma}\mathbf{Y}^T$, where $\mathbf{X} \in \mathbb{R}^{(K+1)\times(K+1)}$ and $\mathbf{Y} \in \mathbb{R}^{K\times K}$ are orthogonal matrices; $\mathbf{\Sigma}$ is a $(K+1) \times K$ matrix, with only the $(i,i)$-th entries as non-zero values $\{\sigma_i\}$. One can rewrite (14) as

$$K\Psi_{\text{ISC}} \geq ||\hat{\mathbf{F}} - \mathbf{G}||_F^2 = ||\mathbf{Z}||_F^2 \geq ||\mathbf{Z}^T\mathbf{Z}||_F$$
$$= ||\mathbf{I}_K - \mathbf{H}^T\mathbf{\Lambda}^2\mathbf{H}||_F$$
$$= ||\mathbf{I}_K - \mathbf{Y}\mathbf{\Sigma}\mathbf{X}^T\mathbf{\Lambda}^2\mathbf{X}\mathbf{\Sigma}\mathbf{Y}^T||_F$$
$$= ||\mathbf{Y}(\mathbf{I}_K - \mathbf{\Sigma}\mathbf{X}^T\mathbf{\Lambda}^2\mathbf{X}\mathbf{\Sigma})\mathbf{Y}^T||_F$$
$$= ||\mathbf{I}_K - \mathbf{\Sigma}\mathbf{X}^T\mathbf{\Lambda}^2\mathbf{X}\mathbf{\Sigma}||_F.$$

Note that $(\mathbf{\Lambda}\mathbf{X}\mathbf{\Sigma})_{ij} = \lambda_i\sigma_j\mathbf{X}_{ij}$, which can lead to

$$(\mathbf{\Sigma}\mathbf{X}^T\mathbf{\Lambda}^2\mathbf{X}\mathbf{\Sigma})_{ij} = \left\{ \begin{array}{l} \lambda_i^2\sigma_i^2, \ i = j \\ 0, \quad i \neq j \end{array} \right. .$$

We further have

$$K\Psi_{\text{ISC}} \geq ||\mathbf{I}_K - \mathbf{\Sigma}\mathbf{X}^T\mathbf{\Lambda}^2\mathbf{X}\mathbf{\Sigma}||_F = [\sum_{r=1}^{K}(1 - \lambda_r^2\sigma_r^2)^2]^{1/2}$$

$$\geq [\sum_{r=1}^{K}(1 - \lambda_1^2\sigma_r^2)^2]^{1/2} \geq [\sum_{r=1}^{K}(1 - \sigma_r^2)^2]^{1/2}$$

$$= ||\mathbf{I}_{(K+1)\times K} - \mathbf{\Sigma}||_F,$$

where $\mathbf{I}_{(K+1)\times K} := [\mathbf{I}_K, \mathbf{0}_{K\times 1}]^T$. Let $\mathbf{O} := \mathbf{X}\mathbf{I}_{(K+1)\times K}\mathbf{Y}^T$, which is an orthogonal matrix. One can derive the following inequalities:

$$||\mathbf{F}\mathbf{O} - \mathbf{G}||_F = ||\mathbf{U}\mathbf{\Lambda}\mathbf{O} - \mathbf{G}||_F$$
$$= ||\mathbf{U}\mathbf{\Lambda}\mathbf{O} - \hat{\mathbf{F}} + \hat{\mathbf{F}} - \mathbf{G}||_F$$
$$\leq ||\mathbf{U}\mathbf{\Lambda}\mathbf{O} - \hat{\mathbf{F}}||_F + ||\hat{\mathbf{F}} - \mathbf{G}||_F$$
$$= ||\mathbf{U}(\mathbf{\Lambda}\mathbf{O} - \mathbf{\Lambda}\mathbf{H})||_F + ||\hat{\mathbf{F}} - \mathbf{G}||_F$$
$$\leq ||\mathbf{\Lambda}\mathbf{O} - \mathbf{\Lambda}\mathbf{H}||_F + ||\hat{\mathbf{F}} - \mathbf{G}||_F$$

For the first term, one can derive the following inequality:

$$
\begin{aligned}
||\mathbf{\Lambda O} - \mathbf{\Lambda H}||_F &= ||\mathbf{\Lambda X}(\mathbf{I}_{(K+1)\times K} - \mathbf{\Sigma})\mathbf{Y}^T||_F \\
&= ||\mathbf{\Lambda}(\mathbf{I}_{(K+1)\times K} - \mathbf{\Sigma})||_F \\
&= \sqrt{\lambda_1^2(1-\sigma_1)^2 + \cdots + \lambda_K^2(1-\sigma_K)^2} \\
&\leq \sqrt{\lambda_1^2[(1-\sigma_1)^2 + \cdots + (1-\sigma_K)^2]} \\
&\leq \lambda_1\sqrt{(1-\sigma_1)^2(1+\sigma_1)^2 + \cdots + (1-\sigma_K)^2(1+\sigma_K)^2} \\
&= \lambda_1\sqrt{(1-\sigma_1^2)^2 + \cdots + (1-\sigma_K^2)^2} \\
&= \lambda_1||\mathbf{I} - \mathbf{\Sigma}^2||_F = \lambda_1||\mathbf{I} - \mathbf{H}^T\mathbf{H}||_F \\
&\leq \lambda_1||\mathbf{I} - \mathbf{H}^T\mathbf{\Lambda}^2\mathbf{H}||_F = \lambda_1||\mathbf{Z}^T\mathbf{Z}||_F \\
&= \lambda_1||(\hat{\mathbf{F}} - \mathbf{G})^T(\hat{\mathbf{F}} - \mathbf{G})||_F \leq \lambda_1 K\Psi_{\mathrm{ISC}}.
\end{aligned}
$$

By combining (13), we finally have

$$
||\mathbf{FO} - \mathbf{G}||_F \leq ||\mathbf{\Lambda O} - \mathbf{\Lambda H}||_F + ||\hat{\mathbf{F}} - \mathbf{G}||_F \leq \lambda_1 K\Psi + \sqrt{K\Psi_{\mathrm{ISC}}}.
$$

If $K\Psi_{\mathrm{ISC}} < 1$, then

$$
||\mathbf{FO} - \mathbf{G}||_F \leq (1 + \lambda_1)\sqrt{K\Psi_{\mathrm{ISC}}}.
$$

This completes the proof of **Theorem 3**.

## C  PROOF OF LEMMA 4

Since $\mathbf{o}_r$ and $\mathbf{o}_t$ ($\forall r \neq t$) are orthogonal, we have

$$
|\mathbf{o}_r - \mathbf{o}_t|_2^2 = (\mathbf{o}_r - \mathbf{o}_t)^T(\mathbf{o}_r - \mathbf{o}_t) = |\mathbf{o}_r|_2^2 + |\mathbf{o}_r|_2^2 - \mathbf{o}_t^T\mathbf{o}_r - \mathbf{o}_r^T\mathbf{o}_t = 2.
$$

According to Lemma 1 in (Mizutani, 2021), one can have

$$
\left|\frac{\mathbf{a}}{|\mathbf{a}|_2} - \mathbf{b}\right|_2 \leq 2|\mathbf{a} - \mathbf{b}|_2,
$$

for vectors $\mathbf{a}, \mathbf{b} \in \mathbb{R}^{K+1}$. By (2), (8), and **Theorem 3**, we have the following derivation:

$$
\begin{aligned}
g(\hat{S}_1, \cdots, \hat{S}_K; o_1, \cdots, o_K) &= \sum_{r=1}^{K}\sum_{v_i\in\hat{S}_r} d_i \left\|\tilde{\mathbf{F}}_{i,:} - \mathbf{o}_r\right\|_2^2 \\
&= \sum_{r=1}^{K}\sum_{v_i\in\hat{S}_r} d_i \left\|\frac{\mathbf{F}_{i,:}}{|\mathbf{F}_{i,:}|} - \mathbf{o}_r\right\|_2^2 \\
&= \sum_{r=1}^{K}\sum_{v_i\in\hat{S}_r} d_i \left\|\frac{\mathbf{F}_{i,:}\sqrt{\mu(\hat{S}_r)/d_i}}{|\mathbf{F}_{i,:}\sqrt{\mu(\hat{S}_r)/d_i}|} - \mathbf{o}_r\right\|_2^2 \\
&\leq 4\sum_{r=1}^{K}\sum_{v_i\in\hat{S}_r} d_i \left\|\mathbf{F}_{i,:}\sqrt{\frac{\mu(\hat{S}_r)}{d_i}} - \mathbf{o}_r\right\|_2^2 \\
&= 4\sum_{r=1}^{K}\sum_{v_i\in\hat{S}_r} \mu(\hat{S}_r) \left\|\mathbf{F}_{i,:} - \sqrt{\frac{d_i}{\mu(\hat{S}_r)}}\mathbf{o}_r\right\|_2^2 \\
&\leq 4\sum_{r=1}^{K}\sum_{v_i\in\hat{S}_r} \mu_{\max} \left\|\mathbf{F}_{i,:} - \sqrt{\frac{d_i}{\mu(\hat{S}_r)}}\mathbf{o}_r\right\|_2^2 \\
&\leq 4(1+\lambda_1)^2\mu_{\max}K\Psi_{\mathrm{ISC}}.
\end{aligned}
$$

This completes the proof of **Lemma 4**.

# D  PROOF OF THEOREM 8

One can prove **Theorem 8** by contradiction. We first choose a real number

$$\epsilon = 33(1 + \lambda_1)^2 \alpha \tilde{\mu} K \Psi_{\text{ISC}} < 1/4,$$

where $\tilde{\mu} := \mu_{\max}/\mu_{\min}$. For every permutation $\pi : \{1, \cdots, K\} \to \{1, \cdots, K\}$, assume that there is an index $l$ s.t. $\mu(C_l \Delta \hat{S}_{\pi(l)}) \geq 2\epsilon \cdot \mu(S_{\pi(l)})$ for a real number $\epsilon$ (i.e., the assumption of **Lemma 6** and **Theorem 7**). For the *lower bound* given by **Theorem 7**, we have

$$\begin{aligned}
\text{COST}(C_1, \cdots, C_K) &\geq \frac{1}{4}\epsilon \cdot \mu_{\min} - 4(1 + \lambda_1)^2 \alpha K \mu_{\max} \Psi_{\text{ISC}} \\
&\geq \frac{33}{4}(1 + \lambda_1)^2 \alpha K \mu_{\max} \Psi_{\text{ISC}} - 4(1 + \lambda_1)^2 \alpha K \mu_{\max} \Psi_{\text{ISC}}, \\
&> 4(1 + \lambda_1)^2 \alpha K \mu_{\max} \Psi_{\text{ISC}}
\end{aligned}$$

which contracts to the *upper bound* given by **Theorem 5**. It indicates that the assumption $\mu(C_l \Delta \hat{S}_{\pi(l)}) \geq 2\epsilon \cdot \mu(\hat{S}_{\pi(l)})$ is false. Namely, after a suitable renumbering of $(C_1, \cdots, C_K)$, we can derive

$$\mu(C_r \Delta \hat{S}_r) \leq 2\epsilon \cdot \mu(\hat{S}_r) = [66(1 + \lambda_1)^2 \alpha \tilde{\mu} K \Psi_{\text{ISC}}]\mu(\hat{S}_r),$$

for $r \in \{1, \cdots, K\}$. Note that for any sets $A, B \subseteq V$, we have

$$|E(A, V \backslash A)| \leq |E(B, V \backslash B)| + \mu(A \Delta B).$$

For $\mu(C_r)$, we further have

$$\mu(C_r) \geq \mu(C_r \cap \hat{S}_r) = \mu(\hat{S}_r) - \mu(\hat{S}_r \backslash C_r) \geq \mu(\hat{S}_r) - \mu(C_r \Delta \hat{S}_r) \geq (1 - 2\epsilon)\mu(\hat{S}_r).$$

For $\psi(C_r)$, we have

$$\begin{aligned}
\phi(C_r) = \frac{|E(C_r, V \backslash C_r)|}{\mu(C_r)} &\leq \frac{|E(C_r, V \backslash C_r)|}{(1 - 2\epsilon)\mu(\hat{S}_r)} \\
&\leq \frac{|E(\hat{S}_r, V \backslash \hat{C}_r)| + 2\epsilon\mu(\hat{S}_r)}{(1 - 2\epsilon)\mu(\hat{S}_r)} = \frac{1}{1 - 2\epsilon}\phi(\hat{S}_r) + \frac{2\epsilon}{1 - 2\epsilon} \\
&\leq (1 + 4\epsilon)\phi(\hat{S}_r) + 4\epsilon = [1 + 132(1 + \lambda_1)^2 \alpha \tilde{\mu} K \Psi_{\text{ISC}}]\phi(\hat{S}_r) + 132(1 + \lambda_1)^2 \alpha \tilde{\mu} K \Psi_{\text{ISC}}.
\end{aligned}$$

This completes the proof of **Theorem 8**.

# E  PROOF OF PROPOSITION 9

Similar to the derivation of the first inequality in Appendix B (i.e., the proof of **Theorem 3**), we have the following derivations for ASCENT:

$$\begin{aligned}
\mathbf{G}_{:,r}^T(\mathbf{I}_N - \mathbf{L}_\tau)\mathbf{G}_{:,r} &= \sum_{i=1}^{N} \mathbf{G}_{ir}^2 - 2 \sum_{(v_i, v_j) \in E} \frac{\mathbf{G}_{ir}\mathbf{G}_{jr}}{\sqrt{d_i + \tau_i}\sqrt{d_j + \tau_j}}, \\
&= \sum_{(v_i, v_j) \in E} [(\frac{1}{\sqrt{d_i}}\mathbf{G}_{ir})^2 - \frac{2\mathbf{G}_{ir}\mathbf{G}_{jr}}{\sqrt{d_i + \tau_i}\sqrt{d_i + \tau_i}} + [(\frac{1}{\sqrt{d_j}}\mathbf{G}_{jr})^2] \\
&= \sum_{(v_i, v_j) \in E(S_r, V \backslash \hat{S}_r)} \frac{1}{\mu(\hat{S}_r)} + \sum_{(v_i, v_j) \in E(\hat{S}_r)} \frac{2}{\mu(\hat{S}_r)}(1 - \frac{\sqrt{d_i d_j}}{\sqrt{d_i + \tau_i}\sqrt{d_j + \tau_j}}).
\end{aligned}$$

Let $\hat{\tau}_r := \max\{\tau_i | v_i \in \hat{S}_r\}$, i.e., the maximum corrections among nodes in cluster $\hat{S}_r$. Then, we have the following derivations:

$$\mathbf{G}_{:,r}^T(\mathbf{I}_N - \mathbf{L}_\tau)\mathbf{G}_{:,r} \leq \sum_{(v_i,v_j)\in E(S_r,V\backslash\hat{S}_r)} \frac{1}{\mu(\hat{S}_r)} + \sum_{(v_i,v_j)\in E(\hat{S}_r)} \frac{2}{\mu(\hat{S}_r)}(1 - \frac{\sqrt{d_i d_j}}{\sqrt{d_i + \hat{\tau}_r}\sqrt{d_j + \hat{\tau}_r}})$$

$$\leq \frac{|E(\hat{S}_r, V\backslash\hat{S}_r)|}{\mu(\hat{S}_r)} + \frac{2|E(\hat{S}_r)|}{\mu(\hat{S}_r)}(1 - \frac{d_{\min}}{d_{\max} + \hat{\tau}_r})$$

$$= \phi(\hat{S}_r) + \frac{\mu(\hat{S}_r) - |E(\hat{S}_r, V\backslash\hat{S}_r)|}{\mu(\hat{S}_r)}(1 - \frac{d_{\min}}{d_{\max} + \hat{\tau}_r})$$

$$= \phi(\hat{S}_r) + (1 - \phi(\hat{S}_r))(1 - \frac{d_{\min}}{d_{\max} + \hat{\tau}_r})$$

$$= 1 - (1 - \phi(\hat{S}_r))\frac{d_{\min}}{d_{\max} + \hat{\tau}_r}.$$

Following the same definitions of $\{h_{ir}\}$, $\{\hat{\mathbf{u}}_r\}$, and $\hat{\mathbf{F}}$ in Appendix B, we further have

$$||\hat{\mathbf{u}}_r - \mathbf{G}_{:,r}||_2^2 = \sum_{i=K+2}^N h_{ir}^2 \lambda_i^2 \leq \frac{1}{1 - \lambda_{K+2}}[1 - (1 - \phi(\hat{S}_r))\frac{d_{\min}}{d_{\max} + \hat{\tau}_r}],$$

for each cluster $\hat{S}_r$. For the whole graph, we have

$$||\hat{\mathbf{F}} - \mathbf{G}||_F^2 = \sum_{r=1}^K ||\hat{\mathbf{u}}_r - \mathbf{G}_{:,r}||_2^2$$

$$\leq \frac{K}{1 - \lambda_{K+2}} - \frac{1}{1 - \lambda_{K+2}}\sum_{r=1}^K (\frac{d_{\min}}{d_{\max} + \hat{\tau}_r} - \frac{d_{\min}}{d_{\max} + \hat{\tau}_r}\phi(\hat{S}_r))$$

$$= K\left[\frac{1}{1 - \lambda_{K+2}} - \frac{1}{K(1 - \lambda_{K+2})}\sum_{r=1}^K \frac{d_{\min}(1 - \phi(\hat{S}_r))}{d_{\max} + \hat{\tau}_r}\right]$$

$$= K\Psi_{\text{AST}}.$$

By following the same strategy of the proof of **Theorems 3**, **5**, **7**, and **8**, we can complete the proof of **Proposition 9**.

# F  FURTHER ANALYSIS OF DCSC

Our analysis of *ISC* (cf. Section 3) can be easily reduced to other DCSC algorithms. As a demonstration, we summarize the corresponding theoretical results for *NJW*, *RSC*, *SCORE+*, *ISC*, and *ASCENT* in Table 9. For simplicity, we use the subscripts (or superscripts) of 'NJW', 'RSC', 'SC+', 'ISC', and 'AST' to denote corresponding variables of *NJW*, *RSC*, *SCORE+*, *ISC*, and *ASCENT*, respectively. Based on Table 9, we try to answer the following questions.

- **Q1**: Why can *ISC* potentially outperform *RSC*?
- **Q2**: When can a DCSC algorithm (e.g., *RSC*) potentially outperform vanilla spectral clustering (i.e., *NJW*)?
- **Q3**: When can ASCENT potentially outperform *ISC*?

## F.1  **Q1**: WHY CAN *ISC* POTENTIALLY OUTPERFORM *RSC*?

Suppose *RSC* and *ISC* have almost the same approximation ratio $\alpha$ of $K$Means. Usually, $\alpha$ may be related to the dimensionality $m$ of input data (i.e., $m = K$ and $m = (K + 1)$ for *RSC* and *ISC* as summarized in Table 1), depending on the concrete $K$Means algorithm (e.g., $\alpha = O(\log K)$ for $K$Means++ (Arthur & Vassilvitskii, 2007)). Moreover, suppose *RSC* and *ISC* use the same correction term $\tau$. Then, we have

$$\lambda_{K+1}^{\text{RSC}} = \lambda_{K+1}^{\text{ISC}} \geq \lambda_{K+2}^{\text{ISC}} \Rightarrow \Psi_{\text{RSC}} \geq \Psi_{\text{ISC}}.$$

Table 9: Further theoretical results of DCSC.

| | | |
|---|---|---|
| $\Psi$ | *NJW* | $\Psi_{\mathrm{NJW}} := \left(1 - \lambda_{K+1}^{\mathrm{NJW}}\right)^{-1} \bar{\phi}_K(G)$ |
| | *RSC* | $\Psi_{\mathrm{RSC}} := \left(1 - \lambda_{K+1}^{\mathrm{RSC}}\right)^{-1} \left[1 - \tilde{d}(1 - \bar{\phi}_K(G))\right]$ |
| | *SCORE+* | $\Psi_{\mathrm{SC+}} = \left(1 - \lambda_{K+2}^{\mathrm{SC+}}\right)^{-1} \left[1 - \tilde{d}(1 - \bar{\phi}_K(G))\right]$ |
| | *ISC* | $\Psi_{\mathrm{ISC}} := \left(1 - \lambda_{K+2}^{\mathrm{ISC}}\right)^{-1} \left[1 - \tilde{d}(1 - \bar{\phi}_K(G))\right]$ |
| | *ASCENT* | $\Psi_{\mathrm{AST}} := \left(1 - \lambda_{K+2}^{\mathrm{AST}}\right)^{-1} \left[1 - (\hat{d} - \bar{\varphi}_K(G))\right]$ |
| **Assumptions** | *NJW* | $\Psi_{\mathrm{NJW}} \le 1/(600\alpha\tilde{\mu}K)$ |
| | *RSC* | $\Psi_{\mathrm{RSC}} \le 1/(528\alpha\tilde{\mu}K)$ |
| | *SCORE+* | $\Psi_{\mathrm{SC+}} \le 1/[132(1 + \lambda_2^{\mathrm{SC+}})\alpha\tilde{\mu}K]$ |
| | *ISC* | $\Psi_{\mathrm{ISC}} \le 1/[132(1 + \lambda_1^{\mathrm{ISC}})^2\alpha\tilde{\mu}K]$ |
| | *ASCENT* | $\Psi_{\mathrm{AST}} \le 1/[132(1 + \lambda_1^{\mathrm{AST}})^2\alpha\tilde{\mu}K]$ |
| $\mu(C_r \Delta \hat{S}_r) \le$ | *NJW* | $[300\alpha\tilde{\mu}K\Psi_{\mathrm{NJW}}]\mu(\hat{S}_r)$ |
| | *RSC* | $[264\alpha\tilde{\mu}K\Psi_{\mathrm{RSC}}]\mu(\hat{S}_r)$ |
| | *SCORE+* | $[66(1 + \lambda_2^{\mathrm{SC+}})^2\alpha\tilde{\mu}K\Psi_{\mathrm{SC+}}]\mu(\hat{S}_r)$ |
| | *ISC* | $[66(1 + \lambda_1^{\mathrm{ISC}})^2\alpha\tilde{\mu}K\Psi_{\mathrm{ISC}}]\mu(\hat{S}_r)$ |
| | *ASCENT* | $[66(1 + \lambda_1^{\mathrm{AST}})^2\alpha\tilde{\mu}K\Psi_{\mathrm{AST}}]\mu(\hat{S}_r)$ |
| $\phi(C_r) \le$ | *NJW* | $[1 + 600\alpha\tilde{\mu}K\Psi_{\mathrm{NJW}}]\phi(\hat{S}_r) + 300\alpha\tilde{\mu}K\Psi_{\mathrm{NJW}}$ |
| | *RSC* | $[1 + 528\alpha\tilde{\mu}K\Psi_{\mathrm{RSC}}]\phi(\hat{S}_r) + 528\alpha\tilde{\mu}K\Psi_{\mathrm{RSC}}$ |
| | *SCORE+* | $[1 + 132(1 + \lambda_2^{\mathrm{SC+}})\alpha\tilde{\mu}K\Psi_{\mathrm{SC+}}]\phi(\hat{S}_r) + 132(1 + \lambda_2^{\mathrm{SC+}})\alpha\tilde{\mu}K\Psi_{\mathrm{SC+}}$ |
| | *ISC* | $[1 + 132(1 + \lambda_1^{\mathrm{ISC}})^2\alpha\tilde{\mu}K\Psi_{\mathrm{ISC}}]\phi(\hat{S}_r) + 132(1 + \lambda_1^{\mathrm{ISC}})\alpha\tilde{\mu}K\Psi_{\mathrm{ISC}}$ |
| | *ASCENT* | $[1 + 132(1 + \lambda_1^{\mathrm{AST}})^2\alpha\tilde{\mu}K\Psi_{\mathrm{AST}}]\phi(\hat{S}_r) + 132(1 + \lambda_1^{\mathrm{AST}})\alpha\tilde{\mu}K\Psi_{\mathrm{AST}}$ |

Note that $\lambda_1^{\mathrm{ISC}} < 1$. For the upper bond of **mis-clustered volume** $\mu(C_r \Delta \hat{S}_r)$, we further have

$$66(1 + \lambda_1^{\mathrm{ISC}})^2\alpha\tilde{\mu}K\Psi_{\mathrm{ISC}}\mu(\hat{S}_r) < 264\alpha\tilde{\mu}K\Psi_{\mathrm{RSC}}\mu(\hat{S}_r),$$

which indicates that *ISC* has a tighter upper bound for $\mu(C_r \Delta \hat{S}_r)$ than that of *RSC*. One can also reach the same conclusion for **conductance** $\phi(C_r)$. Therefore, *ISC* can potentially achieve better clustering quality (measured by $\mu(C_r \Delta \hat{S}_r)$ or $\phi(C_r)$) than *RSC*.

## F.2 **Q2**: WHEN CAN A DCSC ALGORITHM POTENTIALLY OUTPERFORM VANILLA SPECTRAL CLUSTERING?

We first compare the upper bound of $\mu(C_r \Delta \hat{S}_r)$ for *NJW* and *RSC*, which is equivalent to comparing values of $300\Psi_{\mathrm{NJW}}$ and $264\Psi_{\mathrm{RSC}}$. When *RSC* outperforms *NJW*, *RSC* is more likely to have a tighter upper bound of $\mu(C_r \Delta \hat{S}_r)$, which indicates that $300\Psi_{\mathrm{NJW}} \ge 264\Psi_{\mathrm{RSC}} \Rightarrow (300\Psi_{\mathrm{NJW}} - 264\Psi_{\mathrm{RSC}}) \ge 0$. For simplicity, let $\tilde{d} := d_{\min}/(d_{\max} + \tau)$. We further have

$$300\Psi_{\mathrm{NJW}} - 264\Psi_{\mathrm{RSC}} = \frac{300}{1 - \lambda_{K+1}^{\mathrm{NJW}}}\bar{\phi}_K(G) - \frac{264}{1 - \lambda_{K+1}^{\mathrm{RSC}}} + \frac{264\tilde{d}}{1 - \lambda_{K+1}^{\mathrm{RSC}}} - \frac{264\tilde{d}}{1 - \lambda_{K+1}^{\mathrm{RSC}}}\bar{\phi}_K(G) \ge 0,$$

$$\Rightarrow \left(\frac{300}{1 - \lambda_{K+1}^{\mathrm{NJW}}} - \frac{264\tilde{d}}{1 - \lambda_{K+1}^{\mathrm{RSC}}}\right)\bar{\phi}_K(G) \ge \frac{264}{1 - \lambda_{K+1}^{\mathrm{RSC}}}(1 - \tilde{d}),$$

$$\Rightarrow \left[\frac{300}{264}\frac{1 - \lambda_{K+1}^{\mathrm{RSC}}}{1 - \lambda_{K+1}^{\mathrm{NJW}}} - \tilde{d}\right]\bar{\phi}_K(G) \ge (1 - \tilde{d}),$$

$$\Rightarrow \frac{75}{66} \cdot \frac{1 - \lambda_{K+1}^{\mathrm{RSC}}}{1 - \lambda_{K+1}^{\mathrm{NJW}}} \cdot \frac{d_{\max} + \tau}{d_{\max} - d_{\min} + \tau} - \frac{d_{\min}}{d_{\max} - d_{\min} + \tau} \ge \bar{\phi}_K^{-1}(G).$$

Let $q := (1 - \lambda_{K+1}^{\mathrm{RSC}})/(1 - \lambda_{K+1}^{\mathrm{NJW}})$. Usually, we have $\lambda_{K+1}^{\mathrm{RSC}} \le \lambda_{K+1}^{\mathrm{NJW}}$ and thus $q \ge 1$. Assume *RSC* adopts its default setting of $\tau$ (i.e., $\tau = \bar{d}$). One can rewrite the aforementioned inequality as

$$\frac{1.14q(d_{\max} + \tau) - d_{\min}}{d_{\max} - d_{\min} + \tau} = \frac{1.14qd_{\max} - d_{\min} + 1.14q\bar{d}}{d_{\max} - d_{\min} + \bar{d}} \ge \bar{\phi}_K^{-1}(G).$$

To ensure that the aforementioned inequality holds, one may first ensure that the right part $\bar{\phi}_K^{-1}(G)$ is small enough (i.e., $\phi_K(G)$ is large). It implies that the graph $G$ is not so well-clustered, in contrast to the well-clustered condition (Ng et al., 2001; Mizutani, 2021) interpreted in Section 3. Moreover, one may also ensure that the left part is large enough. With the increase of degree heterogeneity, the value of numerator increases faster than that of denominator. Therefore, higher degree heterogeneity results in a larger value of the left part. In summary, a graph $G$ (i) has a high degree heterogeneity and (ii) is not so well-clustered, *RSC* has a tighter upper bound of **mis-clustered volume** $\mu(C_r \Delta \hat{S}_r)$ than that of *NJW*, indicating that *RSC* may potentially outperform *NJW*. One can also reach a similar conclusion by comparing the upper bounds of **conductance** $\phi(C_r)$, because the upper bound of $\phi(C_r)$ is derived based on that of $\mu(C_r \Delta \hat{S}_r)$.

### F.3 **Q3**: WHEN CAN ASCENT POTENTIALLY OUTPERFORM *ISC*?

When $\lambda_{K+2}^{\text{ISC}} \geq \lambda_{K+2}^{\text{AST}}$, we have $\Psi_{\text{ISC}} = (1 - \lambda_{K+2}^{\text{ISC}})^{-1}[1 - (\tilde{d} - \tilde{d}\bar{\phi}_K(G))] \geq (1 - \lambda_{K+2}^{\text{AST}})^{-1}[1 - (\tilde{d} - \tilde{d}\bar{\phi}_K(G))]$, with $\tilde{d} := d_{\min}/(d_{\max} + \tau)$. We further have

$$
\Psi_{\text{ISC}} - \Psi_{\text{AST}} \geq \frac{1}{1 - \lambda_{K+2}^{\text{AST}}}[(\hat{d} - \tilde{d}) + (\tilde{d}\bar{\phi}_K(G) - \bar{\varphi}_K(G))]
$$

$$
= \frac{1}{1 - \lambda_{K+2}^{\text{AST}}} \frac{1}{K} \sum_{r=1}^{K} \left[ (\frac{d_{\min}}{d_{\max} + \hat{\tau}_r} - \frac{d_{\min}}{d_{\max} + \tau})(1 - \phi(\hat{S}_r)) \right]
$$

$$
= \frac{1}{1 - \lambda_{K+2}^{\text{AST}}} \frac{1}{K} \sum_{r=1}^{K} \left[ \frac{d_{\min}(\tau - \hat{\tau}_r)}{(d_{\max} + \tau)(d_{\max} + \hat{\tau}_r)}(1 - \phi(\hat{S}_r)) \right]
$$

$$
= \frac{1}{1 - \lambda_{K+2}^{\text{AST}}} \frac{\tilde{d}}{K} \sum_{r=1}^{K} \left[ \frac{\tau - \hat{\tau}_r}{d_{\max} + \hat{\tau}_r}(1 - \phi(\hat{S}_r)) \right].
$$

To ensure $\Psi_{\text{ISC}} \geq \Psi_{\text{AST}} \Rightarrow \Psi_{\text{ISC}} - \Psi_{\text{AST}} \geq 0$, which indicates that ASCENT can potentially outperform *ISC*, one needs to ensure

$$
\sum_{r=1}^{K} \frac{\tau - \hat{\tau}_r}{d_{\max} + \hat{\tau}_r}(1 - \phi(\hat{S}_r)) \geq 0.
$$

Therefore, it is possible for ASCENT to satisfy the aforementioned conditions.

## G COMPLEXITY ANALYSIS

Given a large-scale graph, we usually have $K, L \ll N < |E|$. Assume that the graph to be partitioned is sparse. For ASCENT, the time complexity of deriving node-wise corrections $\{\tau_i\}$ (i.e., lines 1-5 in Algorithm 1) is no more than $O(|E|L) = O(|E|)$ by fully utilizing the sparsity of a graph and the sparse-dense matrix multiplication operation. ASCENT follows the same steps of (i) ED, (ii) spectral embedding arrangement, (iii) embedding normalization, and (iv) $K$Means clustering with *ISC*, which have complexities of (i) $O((N + |E|)K) = O(|E|)$ (using the efficient Lanczos algorithm (Lehoucq et al., 1998) for ED), (ii) $O(NK) = O(N)$, (iii) $O(NK) = O(N)$, and (iv) $O(NK^2t) = O(N)$ (with $t \ll N$ as the number of iterations in $K$Means), respectively. In summary, the overall time complexity of ASCENT is about $O(|E|)$. It has the same complexity with most existing DCSC algorithms. Therefore, the additional step of deriving node-wise corrections $\{\tau_i\}$ will not increase the complexity of ASCENT.

## H DETAILED EXPERIMENT SETUP

**Datasets**. **Caltech** (Red et al., 2011) and **Simmons** (Red et al., 2011) are two graphs regarding friendships of two online social networks. **PolBlogs** (Adamic & Glance, 2005) is a graph constructed based on the links between blogs with different political leaning. **Airport** (Chami et al., 2019) is a graph describing the real-world airline routes as from OpenFlights.org. **Wiki** (Grover & Leskovec, 2016) is a cooccurrence graph of words that appear in the first million bytes of the Wikipedia dump.

Table 10: Recommended parameter settings of ASCENT.

|   | LFR-1 | LFR-2 | SBM-1 | SBM-2 | Caltech | Simmons | PolBlogs | BioGrid | Airport | Wiki | BlogCatalog | ogbn-Protein |
|---|---|---|---|---|---|---|---|---|---|---|---|---|
| $\theta$ | 1.0 | 1.0 | 0.01 | 0.01 | 0.1 | 0.2 | 0.05 | 0.1 | 0.01 | 0.1 | 0.1 | 0.1 |
| $L$ | 4 | 2 | 1 | 1 | 3 | 3 | 1 | 5 | 1 | 4 | 7 | 2 |

Table 11: Detailed evaluation results on **LFR-1** in terms of **NMI**↑.

|   | $\eta$=0.1 | 0.3 | 0.5 |
|---|---|---|---|
| NJW | 0.8764 (0.0230) | 0.5690 (0.1018) | 0.2491 (0.1397) |
| SCORE | 0.7715 (0.1217) | 0.5977 (0.1322) | 0.3019 (0.0835) |
| RSC | 0.8073 (0.0419) | 0.5443 (0.0906) | 0.3918 (0.0901) |
| SCORE+ | 0.7899 (0.0598) | 0.6019 (0.0802) | 0.4366 (0.1048) |
| ISC | 0.8801 (0.0278) | 0.6527 (0.0827) | 0.4416 (0.0947) |
| GE | 0.6170 (0.1227) | 0.3894 (0.0998) | 0.2060 (0.0548) |
| GAP | 0.0293 (0.1445) | 0.0202 (0.1063) | 0.0079 (0.0377) |
| SDCN | 0.0166 (0.0628) | 0.0037 (0.0067) | 0.0156 (0.0207) |
| MCP | 0.6334 (0.1967) | 0.1166 (0.1849) | 0.0056 (0.0200) |
| DMoN | 0.8064 (0.0532) | 0.5536 (0.1224) | 0.3029 (0.1115) |
| DGC | 0.8847 (0.0275) | 0.6779 (0.0775) | 0.4512 (0.0844) |
| **ASCENT** | **0.8938** (0.0270) | **0.6883** (0.0735) | **0.4642** (0.0954) |
| Improvement | +1.03% | +1.53% | +2.88% |

**BlogCatalog** (Grover & Leskovec, 2016) is extracted from social relationships provided by blogger authors. **BioGrid** (Stark et al., 2006) and **ogbn-Protein** (Szklarczyk et al., 2019) are two protein-protein interaction graphs

During preprocessing, we followed (Qin & Gao, 2010) to extract clustering ground-truth of **BioGrid**, where the complex set CYC2008 (Pu et al., 2009) with 231 protein complexes was used as the reference set. For the rest datasets, we directly used their original formats in our experiments.

Note that we could not use the ground-truth of **Wiki** and **BlogCatalog**, which describes overlapping community structures, since we focus on disjoint graph clustering in this study. As highlighted in Section 2, we assume that graph attributes are unavailable. The ground-truth of **Airport** describes the structural role that each node plays in the graph topology (a.k.a. node identity), which may preserve information contradicting with clustering structures (Qin & Yeung, 2024; Yan et al., 2024), so we could also not use its ground-truth for the evaluation of graph clustering. **ogbn-Protein** is an attributed graph for the evaluation of node classification. Our evaluation only utilized the graph topology of this dataset. In particular, we did not use the ground-truth of **ogbn-Protein**, because it is unclear that such ground-truth is dominated by graph topology or attributes.

**Experiment Environment**. All the experiments were conducted on a server with one Intel Xeon CPU (4214R @2.40GHz), one 24GB memory GPU, 512GB main memory, and Ubuntu Linux OS. We implemented each spectral clustering method (i.e., *NJW*, *SCORE*, *RSC*, *SCORE+*, *ISC*, and ASCENT) using `Python`, including the sparse ED supported by `SciPy`. Moreover, we adopted the official open-source implementations of all the deep learning baselines (i.e., *GraphEncoder*, *GAP*, *SDCN*, *MinCutPool*, *DMoN*, and *DGCluster*), which are based on `PyTorch` or `TensorFlow` and thus were ran on the GPU.

**Parameter Settings**. The details parameter settings of $\{\theta, L\}$ in ASCENT on all the datasets are depicted in Table 10.

Table 12: Detailed evaluation results on **LFR-1** in terms of **AC**↑.

|   | $\eta$=0.1 | 0.3 | 0.5 |
|---|---|---|---|
| NJW | 0.9690 (0.0051) | 0.7706 (0.0854) | 0.2491 (0.1397) |
| SCORE | 0.8690 (0.1403) | 0.8017 (0.1122) | 0.3019 (0.0835) |
| RSC | 0.9498 (0.0109) | 0.7865 (0.0700) | 0.3918 (0.0901) |
| SCORE+ | 0.9188 (0.0861) | 0.8403 (0.0407) | 0.4366 (0.1048) |
| ISC | 0.9723 (0.0086) | 0.8659 (0.0489) | 0.4416 (0.0947) |
| GE | 0.8173 (0.0895) | 0.6069 (0.1156) | 0.2060 (0.0548) |
| GAP | 0.4130 (0.1182) | 0.3503 (0.1088) | 0.0079 (0.0377) |
| SDCN | 0.4064 (0.1046) | 0.3374 (0.0712) | 0.0156 (0.0207) |
| MCP | 0.8418 (0.1359) | 0.4272 (0.1694) | 0.0056 (0.0200) |
| DMoN | 0.9341 (0.0370) | 0.7608 (0.0839) | 0.3029 (0.1115) |
| DGC | 0.9715 (0.0078) | 0.8582 (0.0814) | 0.4512 (0.0844) |
| **ASCENT** | **0.9760** (0.0074) | **0.8873** (0.0357) | **0.4642** (0.0954) |
| Improvement | +0.38% | +2.47% | +2.88% |

Table 13: Detailed evaluation results on **LFR-1** in terms of **conductance**↓.

|  | $\eta$=0.1 | 0.3 | 0.5 |
|---|---|---|---|
| NJW | 0.1004 (0.0068) | 0.3489 (0.0465) | 0.5712 (0.0356) |
| SCORE | 0.2168 (0.1307) | 0.3964 (0.0982) | 0.6610 (0.0953) |
| RSC | 0.1011 (0.0076) | 0.3217 (0.0192) | 0.5282 (0.0160) |
| SCORE+ | 0.1275 (0.0620) | 0.3232 (0.0369) | 0.5394 (0.0344) |
| ISC | 0.1000 (0.0098) | 0.3179 (0.0145) | 0.5231 (0.0144) |
| GE | 0.7349 (0.0919) | 0.7902 (0.0663) | 0.8718 (0.0473) |
| GAP | 0.6669 (0.1271) | 0.7387 (0.1069) | 0.8401 (0.0681) |
| SDCN | 0.7092 (0.0822) | 0.7678 (0.0722) | 0.8665 (0.0503) |
| MCP | 0.1960 (0.1491) | 0.6546 (0.1746) | 0.8420 (0.0626) |
| DMoN | 0.1192 (0.0491) | 0.3638 (0.0583) | 0.5745 (0.0392) |
| DGC | 0.1002 (0.0087) | 0.3182 (0.0127) | 0.5266 (0.0260) |
| **ASCENT** | **0.0995 (0.0082)** | **0.3174 (0.0161)** | **0.5201 (0.0212)** |
| Improvement | +0.5% | +0.16% | +0.57% |

Table 14: Detailed evaluation results on **LFR-2** in terms of **NMI**↑.

|  | $d$=10 | 20 | 30 |
|---|---|---|---|
| NJW | 0.2627 (0.1517) | 0.6658 (0.0955) | 0.8147 (0.1124) |
| SCORE | 0.3009 (0.0917) | 0.6519 (0.1125) | 0.8354 (0.0906) |
| RSC | 0.3975 (0.0933) | 0.6186 (0.0981) | 0.7663 (0.0867) |
| SCORE+ | 0.4472 (0.1021) | 0.7181 (0.1177) | 0.8564 (0.0690) |
| ISC | 0.4477 (0.0942) | 0.7262 (0.0913) | 0.8688 (0.0679) |
| GE | 0.2137 (0.0588) | 0.3889 (0.0887) | 0.5424 (0.0856) |
| GAP | 0.0023 (0.0115) | 0.0036 (0.0291) | 0.0035 (0.0228) |
| SDCN | 0.0161 (0.0194) | 0.0088 (0.0157) | 0.0125 (0.0233) |
| MCP | 0.0029 (0.0173) | 0.0000 (0.0002) | 0.0000 (0.0000) |
| DMoN | 0.2970 (0.1166) | 0.4943 (0.1230) | 0.5668 (0.1371) |
| DGC | 0.4467 (0.0918) | 0.7233 (0.0886) | 0.8592 (0.1173) |
| **ASCENT** | **0.4645 (0.0943)** | **0.7753 (0.0845)** | **0.8981 (0.0593)** |
| Improvement | +3.75% | +6.76% | +3.37% |

Table 15: Detailed evaluation results on **LFR-2** in terms of **AC**↑.

|  | $d$=10 | 20 | 30 |
|---|---|---|---|
| NJW | 0.4432 (0.1154) | 0.7915 (0.0800) | 0.9033 (0.0779) |
| SCORE | 0.4774 (0.1070) | 0.7727 (0.1119) | 0.8917 (0.0881) |
| RSC | 0.6122 (0.0474) | 0.7719 (0.0751) | 0.8741 (0.0669) |
| SCORE+ | 0.6630 (0.0573) | 0.8538 (0.0738) | 0.9361 (0.0429) |
| ISC | 0.6659 (0.0519) | 0.8559 (0.0672) | 0.9391 (0.0463) |
| GE | 0.3621 (0.0635) | 0.5409 (0.1017) | 0.6811 (0.0947) |
| GAP | 0.2368 (0.0396) | 0.2527 (0.0329) | 0.2598 (0.0341) |
| SDCN | 0.2400 (0.0384) | 0.2529 (0.0302) | 0.2627 (0.0347) |
| MCP | 0.2386 (0.0403) | 0.2516 (0.0316) | 0.2590 (0.0342) |
| DMoN | 0.4606 (0.0565) | 0.6152 (0.0738) | 0.6724 (0.1022) |
| DGC | 0.6390 (0.0680) | 0.8556 (0.0887) | 0.8948 (0.1092) |
| **ASCENT** | **0.6851 (0.0540)** | **0.8884 (0.0569)** | **0.9544 (0.0389)** |
| Improvement | +2.88% | +3.80% | +1.63% |

Table 16: Detailed evaluation results on **LFR-2** in terms of **conductance**↓.

|  | $d$=10 | 20 | 30 |
|---|---|---|---|
| NJW | 0.5710 (0.0368) | 0.5318 (0.0214) | 0.5137 (0.0119) |
| SCORE | 0.6648 (0.1052) | 0.5722 (0.0578) | 0.5355 (0.0392) |
| RSC | 0.5394 (0.0154) | 0.5260 (0.0148) | 0.5166 (0.0133) |
| SCORE+ | 0.5349 (0.0283) | 0.5222 (0.0089) | 0.5099 (0.0052) |
| ISC | 0.5287 (0.0128) | 0.5221 (0.0070) | 0.5097 (0.0050) |
| GE | 0.8759 (0.0476) | 0.8613 (0.0416) | 0.8422 (0.0339) |
| GAP | 0.8502 (0.0592) | 0.8305 (0.0502) | 0.8185 (0.0450) |
| SDCN | 0.8710 (0.0466) | 0.8509 (0.0456) | 0.8345 (0.0466) |
| MCP | 0.8503 (0.0600) | 0.8321 (0.0493) | 0.8191 (0.0458) |
| DMoN | 0.5916 (0.0374) | 0.6048 (0.0379) | 0.5951 (0.0450) |
| DGC | 0.5306 (0.0356) | 0.5169 (0.0263) | 0.5212 (0.0297) |
| **ASCENT** | **0.5172 (0.0147)** | **0.5114 (0.0089)** | **0.5057 (0.0064)** |
| Improvement | +2.18% | +1.06% | +0.78% |

Table 17: Detailed evaluation results on **SBM-1** in terms of **NMI**↑.

|  | $\gamma$=0.5 | 0.6 | 0.7 |
|---|---|---|---|
| NJW | 0.7994 (0.0374) | 0.9181 (0.0329) | 0.9648 (0.0172) |
| SCORE | 0.7203 (0.0567) | 0.8558 (0.0571) | 0.9104 (0.0514) |
| RSC | 0.7689 (0.0455) | 0.8976 (0.0417) | 0.9447 (0.0328) |
| SCORE+ | 0.8208 (0.0331) | 0.9299 (0.0246) | 0.9702 (0.0127) |
| ISC | 0.8233 (0.0336) | 0.9318 (0.0251) | 0.9732 (0.0111) |
| GE | 0.6679 (0.0672) | 0.8612 (0.0431) | 0.9342 (0.0311) |
| GAP | 0.0127 (0.0342) | 0.0165 (0.0410) | 0.0176 (0.0455) |
| SDCN | 0.2601 (0.0491) | 0.3913 (0.0739) | 0.4802 (0.0869) |
| MCP | 0.0000 (0.0000) | 0.1481 (0.0197) | 0.9091 (0.0000) |
| DMoN | 0.7625 (0.0420) | 0.8496 (0.0310) | 0.8884 (0.0344) |
| DGC | 0.6802 (0.0536) | 0.8019 (0.0546) | 0.8721 (0.0490) |
| **ASCENT** | **0.8252 (0.0318)** | **0.9341 (0.0228)** | **0.9738 (0.0109)** |
| Improvement | +0.23% | +0.24% | +0.06% |

Table 18: Detailed evaluation results on **SBM-1** in terms of **AC↑**.

| | γ=0.5 | 0.6 | 0.7 |
|---|---|---|---|
| NJW | 0.8774 (0.0360) | 0.9535 (0.0312) | 0.9816 (0.0143) |
| SCORE | 0.7886 (0.0668) | 0.8855 (0.0716) | 0.9262 (0.0625) |
| RSC | 0.8403 (0.0516) | 0.9298 (0.0517) | 0.9614 (0.0372) |
| SCORE+ | 0.8918 (0.0315) | 0.9610 (0.0229) | 0.9853 (0.0076) |
| ISC | 0.8962 (0.0318) | 0.9618 (0.0255) | 0.9866 (0.0084) |
| GE | 0.7628 (0.0742) | 0.9070 (0.0516) | 0.9541 (0.0387) |
| GAP | 0.1547 (0.0264) | 0.1560 (0.0277) | 0.1550 (0.0307) |
| SDCN | 0.3509 (0.0632) | 0.4731 (0.0912) | 0.5458 (0.0985) |
| MCP | 0.0000 (0.0000) | 0.1481 (0.0197) | 0.9091 (0.0000) |
| DMoN | 0.7933 (0.0596) | 0.8507 (0.0457) | 0.8752 (0.0591) |
| DGC | 0.6792 (0.0674) | 0.7614 (0.0715) | 0.8150 (0.0670) |
| **ASCENT** | **0.9008** (0.0258) | **0.9651** (0.0200) | **0.9868** (0.0089) |
| Improvement | +0.51% | +0.34% | +0.02% |

Table 19: Detailed evaluation results on **SBM-1** in terms of **conductance↓**.

| | γ=0.5 | 0.6 | 0.7 |
|---|---|---|---|
| NJW | 0.7054 (0.0063) | 0.6621 (0.0075) | 0.6280 (0.0055) |
| SCORE | 0.7160 (0.0086) | 0.6711 (0.0108) | 0.6373 (0.0118) |
| RSC | 0.7097 (0.0073) | 0.6652 (0.0086) | 0.6318 (0.0085) |
| SCORE+ | 0.7033 (0.0060) | 0.6608 (0.0068) | 0.6272 (0.0050) |
| ISC | 0.7032 (0.0061) | 0.6606 (0.0068) | 0.6270 (0.0048) |
| GE | 0.7470 (0.0209) | 0.6780 (0.0161) | 0.6377 (0.0145) |
| GAP | 0.9072 (0.0049) | 0.9066 (0.0061) | 0.9066 (0.0065) |
| SDCN | 0.8554 (0.0133) | 0.8120 (0.0227) | 0.7774 (0.0292) |
| MCP | 0.0000 (0.0000) | 0.1481 (0.0197) | 0.9091 (0.0000) |
| DMoN | 0.7214 (0.0087) | 0.6893 (0.0115) | 0.6599 (0.0160) |
| DGC | 0.7791 (0.0198) | 0.7344 (0.0244) | 0.6957 (0.0271) |
| **ASCENT** | **0.7027** (0.0058) | **0.6602** (0.0065) | **0.6268** (0.0047) |
| Improvement | +0.02% | +0.06% | +0.03% |

Table 20: Detailed evaluation results on **SBM-2** in terms of **NMI↑**.

| | β=2.5 | 2.75 | 3 |
|---|---|---|---|
| NJW | 0.8020 (0.0384) | 0.7616 (0.0361) | 0.7334 (0.0370) |
| SCORE | 0.7239 (0.0609) | 0.6846 (0.0540) | 0.6502 (0.0521) |
| RSC | 0.7709 (0.0485) | 0.7335 (0.0425) | 0.7038 (0.0404) |
| SCORE+ | 0.8206 (0.0347) | 0.7829 (0.0308) | 0.7536 (0.0338) |
| ISC | 0.8246 (0.0336) | 0.7880 (0.0324) | 0.7570 (0.0347) |
| GE | 0.6768 (0.0587) | 0.6261 (0.0523) | 0.5720 (0.0632) |
| GAP | 0.0137 (0.0380) | 0.0118 (0.0311) | 0.0110 (0.0303) |
| SDCN | 0.2629 (0.0529) | 0.2226 (0.0571) | 0.1899 (0.0502) |
| MCP | 0.0000 (0.0000) | 0.0000 (0.0000) | 0.0000 (0.0000) |
| DMoN | 0.7594 (0.0375) | 0.7342 (0.0471) | 0.6955 (0.0509) |
| DGC | 0.6823 (0.0562) | 0.6694 (0.0499) | 0.6490 (0.0578) |
| **ASCENT** | **0.8274** (0.0320) | **0.7900** (0.0309) | **0.7605** (0.0319) |
| Improvement | +0.34% | +0.25% | +0.34% |

Table 21: Detailed evaluation results on **SBM-2** in terms of **AC↑**.

| | β = 2.5 | 2.75 | 3 |
|---|---|---|---|
| NJW | 0.8782 (0.0354) | 0.8523 (0.0351) | 0.8336 (0.0376) |
| SCORE | 0.7927 (0.0704) | 0.7577 (0.0637) | 0.7366 (0.0597) |
| RSC | 0.8375 (0.0568) | 0.8070 (0.0560) | 0.7966 (0.0475) |
| SCORE+ | 0.8910 (0.0326) | 0.8661 (0.0279) | 0.8454 (0.0340) |
| ISC | 0.8954 (0.0315) | 0.8705 (0.0319) | 0.8521 (0.0334) |
| GE | 0.7703 (0.0634) | 0.7319 (0.0530) | 0.6868 (0.0674) |
| GAP | 0.1511 (0.0307) | 0.1529 (0.0271) | 0.1507 (0.0261) |
| SDCN | 0.3680 (0.0660) | 0.3287 (0.0630) | 0.2962 (0.0588) |
| MCP | 0.1481 (0.0197) | 0.1481 (0.0197) | 0.1481 (0.0197) |
| DMoN | 0.7917 (0.0511) | 0.7781 (0.0631) | 0.7498 (0.0658) |
| DGC | 0.6930 (0.0666) | 0.6842 (0.0630) | 0.6789 (0.0676) |
| **ASCENT** | **0.8987** (0.0295) | **0.8751** (0.0279) | **0.8579** (0.0269) |
| Improvement | +0.37% | +0.53% | +0.68% |

Table 22: Detailed evaluation results on **SBM-2** in terms of **conductance↓**.

| | β = 2.5 | 2.75 | 3 |
|---|---|---|---|
| NJW | 0.7059 (0.0069) | 0.7059 (0.0065) | 0.7056 (0.0060) |
| SCORE | 0.7163 (0.0092) | 0.7160 (0.0085) | 0.7165 (0.0072) |
| RSC | 0.7101 (0.0078) | 0.7100 (0.0070) | 0.7102 (0.0062) |
| SCORE+ | 0.7039 (0.0065) | 0.7042 (0.0064) | 0.7035 (0.0059) |
| ISC | 0.7038 (0.0066) | 0.7039 (0.0065) | 0.7035 (0.0060) |
| GE | 0.7461 (0.0193) | 0.7542 (0.0181) | 0.7639 (0.0196) |
| GAP | 0.9069 (0.0058) | 0.9071 (0.0050) | 0.9071 (0.0050) |
| SDCN | 0.8540 (0.0153) | 0.8632 (0.0149) | 0.8716 (0.0147) |
| MCP | 0.9091 (0.0000) | 0.9091 (0.0000) | 0.9091 (0.0000) |
| DMoN | 0.7221 (0.0078) | 0.7186 (0.0086) | 0.7175 (0.0072) |
| DGC | 0.7759 (0.0207) | 0.7761 (0.0174) | 0.7745 (0.0195) |
| **ASCENT** | **0.7032** (0.0064) | **0.7033** (0.0064) | **0.7029** (0.0056) |
| Improvement | +0.09% | +0.09% | +0.09% |

Table 23: Detailed evaluation results on **Caltech** and **Simmons**.

| | Caltech | | | Simmons | | |
|---|---|---|---|---|---|---|
| | NMI↑ | AC↑ | Cond↓ | NMI↑ | AC↑ | Cond↓ |
| NJW | 0.6213 (0.0032) | 0.7539 (0.0043) | 0.5076 (0.0007) | 0.6796 (0.0000) | 0.7344 (0.0000) | 0.3387 (0.0000) |
| SCORE | 0.5639 (0.0035) | 0.6905 (0.0028) | 0.5012 (0.0004) | 0.5853 (0.0002) | 0.7639 (0.0004) | 0.2992 (0.0001) |
| RSC | 0.5858 (0.0011) | 0.7105 (0.0007) | 0.4986 (0.0002) | 0.6152 (0.0011) | 0.7861 (0.0009) | 0.2888 (0.0001) |
| SCORE+ | 0.6914 (0.0063) | 0.8285 (0.0047) | 0.4844 (0.0017) | 0.7295 (0.0000) | 0.8881 (0.0004) | 0.2741 (0.0002) |
| ISC | 0.7028 (0.0021) | 0.8373 (0.0019) | 0.4832 (0.0002) | 0.7357 (0.0000) | 0.8936 (0.0000) | 0.2735 (0.0000) |
| GE | 0.3675 (0.0811) | 0.4444 (0.0803) | 0.6897 (0.0417) | 0.4919 (0.0337) | 0.5886 (0.0876) | 0.4813 (0.0570) |
| GAP | 0.6580 (0.0366) | 0.7559 (0.0699) | 0.4994 (0.0124) | 0.4869 (0.2484) | 0.5796 (0.1529) | 0.4084 (0.1778) |
| SDCN | 0.2850 (0.0746) | 0.3403 (0.0610) | 0.7479 (0.0403) | 0.3828 (0.0856) | 0.5391 (0.0767) | 0.5139 (0.0870) |
| MCP | 0.5057 (0.2537) | 0.6332 (0.2387) | 0.5816 (0.1468) | 0.6466 (0.0231) | 0.8290 (0.0100) | 0.2980 (0.0035) |
| DMoN | 0.6629 (0.0011) | 0.7247 (0.0014) | 0.5397 (0.0063) | 0.6364 (0.0035) | 0.8120 (0.0064) | 0.2800 (0.0003) |
| DGC | 0.6675 (0.0159) | 0.7532 (0.0298) | 0.5192 (0.0263) | 0.7053 (0.0200) | 0.7974 (0.0437) | 0.3318 (0.0432) |
| **ASCENT** | **0.7120** (0.0105) | **0.8441** (0.0056) | **0.4828** (0.0002) | **0.7406** (0.0000) | **0.8962** (0.0000) | **0.2734** (0.0000) |
| Improvement | +1.31% | +0.81% | +0.08% | +0.67% | +0.29% | +0.04% |

Table 24: Detailed evaluation results on **PolBlogs** and **BioGrid**.

| | PolBlogs | | | BioGrid | | |
|---|---|---|---|---|---|---|
| | NMI↑ | AC↑ | Cond↓ | NMI↑ | AC↑ | Cond↓ |
| NJW | 0.0006 (0.0000) | 0.5188 (0.0000) | 0.2694 (0.0000) | 0.4183 (0.0034) | 0.1284 (0.0020) | 0.6620 (0.0048) |
| SCORE | 0.7250 (0.0000) | 0.9525 (0.0000) | 0.0767 (0.0000) | 0.1393 (0.0072) | 0.0737 (0.0034) | 0.9090 (0.0083) |
| RSC | 0.7133 (0.0000) | 0.9476 (0.0000) | 0.0734 (0.0000) | **0.4364** (0.0012) | 0.1352 (0.0010) | 0.6607 (0.0038) |
| SCORE+ | 0.7308 (0.0000) | 0.9533 (0.0000) | 0.0753 (0.0000) | 0.2436 (0.0099) | 0.0944 (0.0034) | 0.8202 (0.0053) |
| ISC | 0.7267 (0.0000) | 0.9509 (0.0000) | 0.0735 (0.0000) | 0.4321 (0.0016) | 0.1326 (0.0021) | 0.6607 (0.0016) |
| GE | 0.0615 (0.0063) | 0.5188 (0.0046) | 0.5013 (0.0003) | 0.3174 (0.0027) | 0.1128 (0.0041) | 0.9851 (0.0014) |
| GAP | 0.4289 (0.3502) | 0.7782 (0.2105) | 0.2440 (0.2090) | 0.4418 (0.0058) | 0.1340 (0.0029) | 0.8040 (0.0169) |
| SDCN | 0.1496 (0.0767) | 0.6293 (0.0560) | 0.2848 (0.1107) | 0.2077 (0.0225) | 0.0844 (0.0052) | 0.9583 (0.0044) |
| MCP | 0.5815 (0.2908) | 0.8656 (0.1726) | 0.1626 (0.1687) | 0.0000 (0.0000) | 0.0443 (0.0000) | 0.9877 (0.0000) |
| DMoN | 0.7116 (0.0053) | 0.9491 (0.0013) | 0.0747 (0.0001) | 0.4173 (0.0048) | 0.1291 (0.0038) | 0.7974 (0.0162) |
| DGC | 0.7121 (0.0098) | 0.9491 (0.0014) | 0.0742 (0.0006) | 0.2163 (0.0229) | 0.0774 (0.0066) | 0.9104 (0.0023) |
| **ASCENT** | **0.7348** (0.0000) | **0.9534** (0.0000) | **0.0731** (0.0000) | 0.4328 (0.0027) | **0.1359** (0.0020) | **0.6488** (0.0041) |
| Improvement | +0.55% | +0.01% | +0.41% | – | +0.52% | +1.80% |

Table 25: Detailed evaluation results on **Airport** and **Wiki** in terms of **conductance**↓.

| | Airport | | | Wiki | | |
|---|---|---|---|---|---|---|
| | K=2 | 8 | 32 | K=2 | 8 | 32 |
| NJW | **0.0167** (0.0000) | 0.1016 (0.0057) | 0.1948 (0.0103) | 0.4016 (0.0000) | 0.7432 (0.0007) | 0.8532 (0.0013) |
| SCORE | 0.1095 (0.0000) | 0.7498 (0.0233) | 0.8582 (0.0056) | 0.4029 (0.0001) | 0.8273 (0.0036) | 0.9208 (0.0032) |
| RSC | 0.0609 (0.0002) | 0.2344 (0.0002) | 0.3711 (0.0040) | 0.3775 (0.0000) | 0.7306 (0.0002) | 0.8532 (0.0005) |
| SCORE+ | 0.0519 (0.0008) | 0.5248 (0.0173) | 0.7147 (0.0179) | 0.3906 (0.0001) | 0.7569 (0.0002) | 0.8687 (0.0023) |
| ISC | 0.0424 (0.0000) | 0.2170 (0.0049) | 0.3504 (0.0047) | 0.3791 (0.0001) | 0.7331 (0.0002) | 0.8578 (0.0011) |
| GE | 0.4814 (0.0272) | 0.8427 (0.0179) | 0.9258 (0.0169) | 0.5208 (0.0024) | 0.8902 (0.0010) | 0.9723 (0.0005) |
| GAP | 0.3200 (0.2205) | 0.8750 (0.0000) | 0.9688 (0.0000) | 0.4466 (0.0655) | 0.8750 (0.0000) | 0.9688 (0.0000) |
| SDCN | 0.1783 (0.0335) | 0.8606 (0.0087) | 0.9273 (0.0141) | 0.5000 (0.0000) | 0.8730 (0.0050) | 0.9641 (0.0012) |
| MCP | 0.0880 (0.0365) | 0.2380 (0.0179) | 0.5433 (0.1725) | 0.5000 (0.0000) | 0.8750 (0.0000) | 0.9688 (0.0000) |
| DMoN | 0.0644 (0.0105) | 0.2040 (0.0123) | 0.5100 (0.0200) | 0.3709 (0.0021) | 0.7725 (0.0091) | 0.9159 (0.0036) |
| DGC | 0.0575 (0.0124) | 0.1354 (0.0155) | 0.6980 (0.0196) | 0.3745 (0.0045) | 0.8353 (0.0099) | 0.9596 (0.0023) |
| **ASCENT** | **0.0167** (0.0000) | **0.0997** (0.0046) | **0.1885** (0.0068) | **0.3743** (0.0001) | **0.7200** (0.0002) | **0.8477** (0.0014) |
| Improvement | – | +1.87% | +3.23% | +0.85% | +1.45% | +0.64% |

Table 26: Detailed evaluation results on **BlogCatalog** and **ogbn-Protein** in terms of **conductance**↓.

| | BlogCatalog | | | ogbn-Protein | | |
|---|---|---|---|---|---|---|
| | K=2 | 8 | 32 | K=2 | 8 | 32 |
| NJW | 0.2935 (0.0000) | 0.6783 (0.0003) | 0.8279 (0.0011) | 0.0634 (0.0000) | 0.1192 (0.0009) | 0.4136 (0.0032) |
| SCORE | 0.2965 (0.0000) | 0.7784 (0.0027) | 0.9367 (0.0064) | 0.2238 (0.0001) | 0.4469 (0.0224) | 0.8242 (0.0049) |
| RSC | 0.2924 (0.0000) | 0.6656 (0.0028) | 0.8174 (0.0018) | 0.1203 (0.0006) | 0.1587 (0.0038) | 0.3663 (0.0054) |
| SCORE+ | 0.2933 (0.0000) | 0.6995 (0.0003) | 0.8619 (0.0025) | 0.0709 (0.0000) | 0.2192 (0.0113) | 0.6433 (0.0143) |
| ISC | 0.2926 (0.0000) | 0.6502 (0.0001) | 0.8143 (0.0017) | 0.0693 (0.0000) | 0.1488 (0.0001) | 0.3464 (0.0071) |
| GE | 0.4959 (0.0000) | 0.8749 (0.0016) | 0.9678 (0.0003) | | OOM | |
| GAP | 0.4181 (0.1003) | 0.8750 (0.0000) | 0.9688 (0.0000) | | OOT | |
| SDCN | 0.4987 (0.0404) | 0.8544 (0.0185) | 0.9580 (0.0063) | | OOM | |
| MCP | 0.3415 (0.0793) | 0.8697 (0.0063) | 0.9682 (0.0011) | | OOM | |
| DMoN | 0.4289 (0.0188) | 0.7595 (0.0147) | 0.9028 (0.0045) | | OOM | |
| DGC | 0.2989 (0.0022) | 0.7773 (0.0244) | 0.9434 (0.0041) | | OOM | |
| **ASCENT** | **0.2923** (0.0000) | **0.6422** (0.0034) | **0.8068** (0.0010) | **0.0347** (0.0000) | **0.1106** (0.0050) | **0.3320** (0.0043) |
| Improvement | +0.10% | +1.23% | +0.92% | +45.27% | +7.21% | +4.16% |

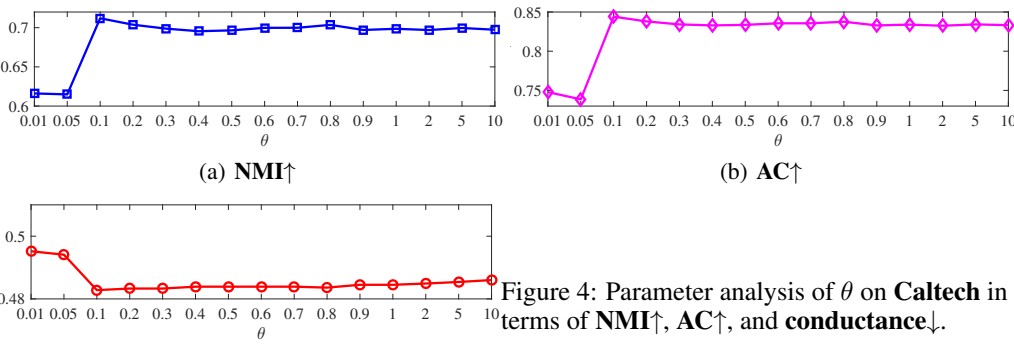

(a) **NMI↑**

(b) **AC↑**

(c) **Conductance↓**

Figure 4: Parameter analysis of $\theta$ on **Caltech** in terms of **NMI↑**, **AC↑**, and **conductance↓**.

Table 27: Evaluation of runtime↓ (sec) on datasets with ground-truth.

| | Caltech | | | | Simmons | | | | PolBlogs | | | | BioGrid | | | |
|---|---|---|---|---|---|---|---|---|---|---|---|---|---|---|---|---|
| | Total | $\tau$ | ED | $K$M | Total | $\tau$ | ED | $K$M | Total | $\tau$ | ED | $K$M | Total | $\tau$ | ED | $K$M |
| NJW | 0.13 | N/A | 0.10 | 0.03 | 0.23 | N/A | 0.21 | 0.03 | 0.17 | N/A | 0.14 | 0.03 | 1.00 | N/A | 0.84 | 0.16 |
| SCORE | 0.10 | N/A | 0.07 | 0.03 | 0.20 | N/A | 0.17 | 0.03 | 0.15 | N/A | 0.13 | 0.02 | 0.80 | N/A | 0.63 | 0.17 |
| RSC | 0.13 | N/A | 0.10 | 0.03 | 0.19 | N/A | 0.16 | 0.03 | 0.15 | N/A | 0.13 | 0.02 | 0.86 | N/A | 0.76 | 0.10 |
| SCORE+ | 0.13 | N/A | 0.11 | 0.02 | 0.31 | N/A | 0.28 | 0.03 | 0.16 | N/A | 0.13 | 0.02 | 0.74 | N/A | 0.64 | 0.10 |
| ISC | 0.17 | N/A | 0.14 | 0.03 | 0.23 | N/A | 0.20 | 0.03 | 0.21 | N/A | 0.19 | 0.02 | 0.93 | N/A | 0.82 | 0.11 |
| GE | 4.36 | N/A | N/A | N/A | 4.85 | N/A | N/A | N/A | 4.12 | N/A | N/A | N/A | 122.01 | N/A | N/A | N/A |
| GAP | 21.92 | N/A | N/A | N/A | 32.85 | N/A | N/A | N/A | 4.29 | N/A | N/A | N/A | 648.30 | N/A | N/A | N/A |
| SDCN | 0.97 | N/A | N/A | N/A | 0.99 | N/A | N/A | N/A | 3.23 | N/A | N/A | N/A | 218.64 | N/A | N/A | N/A |
| MCP | 91.31 | N/A | N/A | N/A | 163.51 | N/A | N/A | N/A | 54.15 | N/A | N/A | N/A | 350.96 | N/A | N/A | N/A |
| DMoN | 101.79 | N/A | N/A | N/A | 174.15 | N/A | N/A | N/A | 257.95 | N/A | N/A | N/A | 579.64 | N/A | N/A | N/A |
| DGC | 81.26 | N/A | N/A | N/A | 131.59 | N/A | N/A | N/A | 164.26 | N/A | N/A | N/A | 559.25 | N/A | N/A | N/A |
| **ASCENT** | 0.15 | 0.01 | 0.11 | 0.03 | 0.24 | 0.03 | 0.18 | 0.03 | 0.17 | 0.02 | 0.13 | 0.02 | 1.02 | 0.07 | 0.81 | 0.14 |

## I  DETAILED EXPERIMENT RESULTS

**Quantitative Evaluation Results**. On each dataset, we recorded the mean $m$ and standard derivation $s$ of each quality metric. Detailed evaluation results in the format of '$m$ ($s$)' are depicted in Tables 11, 12, 13, 14, 15, 16, 17, 18, 19, 20, 21, 22, 23, 24, 25, and 26, where each quality metric is in **bold** or underlined if it performs the best or second-best.

**Further Parameter Analysis**. Example analysis results of $\theta$ on **Caltech** are visualized in Fig. 4, where we adjusted $\theta \in \{0.01, 0.05, 0.1, 0.2, \cdots, 1, 2, 5, 10\}$. In summary, we recommend adjusting $L \in \{1, 2, \cdots, 10\}$ and $\theta \in \{0.01, 0.05, 0.1, 0.5, 1.0\}$ for ASCENT.

**Efficiency Analysis**. In addition to the clustering quality, we further evaluated the efficiency of each method in terms of its overall runtime (sec) to get a feasible clustering result. In particular, we also recorded the runtime of different steps for each spectral clustering method. Results of the efficiency analysis on all the real datasets are depicted in Tables 27, 28, 29, and 30, where (i) $\tau$, (ii) ED, and (iii) $K$M denote the runtime of (i) deriving node-wise corrections $\{\tau_i\}$ (only for ASCENT), (ii) eigen-decomposition of the corresponding graph Laplacian, and (iii) $K$Means clustering (including the arrangement and normalization of corresponding spectral embeddings), respectively.

Compared with deep graph clustering approaches (e.g., *GE* and *GAP*) which involve a time-consuming learning procedure (e.g., gradient descent to iteratively update model parameters), all the spectral clustering methods can achieve significantly better efficiency. Moreover, ED is the major bottleneck for all the spectral clustering algorithms. For ASCENT, the derivation of node-wise corrections $\{\tau_i\}$ would not significantly increase the overall runtime compared with other spectral clustering baselines. In summary, ASCENT can still achieve high inference efficiency close to that of other conventional spectral clustering methods.

## J  LIMITATIONS AND FUTURE DIRECTIONS

**Clustering on Attributed Graphs**. As described in Section 2, we followed the conventional problem statement of graph clustering where topology is the only available information source (without any attributes), due to the complicated corrections between graph topology and attributes. In our future work, we will analyze DCSC on attributed graphs with the consideration of the possible in-

Table 28: Evaluation of runtime↓ (sec) with $K = 2$ on datasets without ground-truth.

| | Airport | | | | Wiki | | | | BlogCatalog | | | | ogbn-Protein | | | |
|---|---|---|---|---|---|---|---|---|---|---|---|---|---|---|---|---|
| | Total | $\tau$ | ED | $KM$ | Total | $\tau$ | ED | $KM$ | Total | $\tau$ | ED | $KM$ | Total | $\tau$ | ED | $KM$ |
| NJW | 0.23 | N/A | 0.20 | 0.03 | 0.88 | N/A | 0.84 | 0.04 | 1.63 | N/A | 1.61 | 0.03 | 201.94 | N/A | 201.77 | 0.17 |
| SCORE | 0.18 | N/A | 0.16 | 0.02 | 0.35 | N/A | 0.32 | 0.03 | 0.88 | N/A | 0.85 | 0.03 | 80.42 | N/A | 80.16 | 0.27 |
| RSC | 0.22 | N/A | 0.18 | 0.03 | 0.76 | N/A | 0.73 | 0.03 | 1.89 | N/A | 1.86 | 0.03 | 189.30 | N/A | 189.07 | 0.23 |
| SCORE+ | 0.21 | N/A | 0.18 | 0.03 | 0.78 | N/A | 0.74 | 0.03 | 1.92 | N/A | 1.89 | 0.03 | 182.00 | N/A | 181.79 | 0.21 |
| ISC | 0.21 | N/A | 0.19 | 0.02 | 0.60 | N/A | 0.57 | 0.03 | 1.86 | N/A | 1.82 | 0.04 | 172.51 | N/A | 172.27 | 0.24 |
| GE | 8.40 | N/A | N/A | N/A | 16.80 | N/A | N/A | N/A | 128.60 | N/A | N/A | N/A | OOM | N/A | N/A | N/A |
| GAP | 38.74 | N/A | N/A | N/A | 116.62 | N/A | N/A | N/A | 241.89 | N/A | N/A | N/A | OOT | N/A | N/A | N/A |
| SDCN | 16.33 | N/A | N/A | N/A | 321.51 | N/A | N/A | N/A | 133.26 | N/A | N/A | N/A | OOM | N/A | N/A | N/A |
| MCP | 125.90 | N/A | N/A | N/A | 116.98 | N/A | N/A | N/A | 113.26 | N/A | N/A | N/A | OOM | N/A | N/A | N/A |
| DMoN | 105.39 | N/A | N/A | N/A | 790.42 | N/A | N/A | N/A | 682.02 | N/A | N/A | N/A | OOM | N/A | N/A | N/A |
| DGC | 60.47 | N/A | N/A | N/A | 634.26 | N/A | N/A | N/A | 642.09 | N/A | N/A | N/A | OOM | N/A | N/A | N/A |
| **ASCENT** | 0.26 | 0.02 | 0.21 | 0.02 | 0.76 | 0.10 | 0.62 | 0.04 | 2.36 | 0.36 | 1.93 | 0.07 | 242.29 | 41.41 | 198.24 | 2.64 |

Table 29: Evaluation of runtime↓ (sec) with $K = 8$ on datasets without ground-truth.

| | Airport | | | | Wiki | | | | BlogCatalog | | | | ogbn-Protein | | | |
|---|---|---|---|---|---|---|---|---|---|---|---|---|---|---|---|---|
| | Total | $\tau$ | ED | $KM$ | Total | $\tau$ | ED | $KM$ | Total | $\tau$ | ED | $KM$ | Total | $\tau$ | ED | $KM$ |
| NJW | 0.29 | N/A | 0.27 | 0.02 | 0.82 | N/A | 0.78 | 0.03 | 1.85 | N/A | 1.81 | 0.04 | 200.58 | N/A | 200.50 | 0.08 |
| SCORE | 0.25 | N/A | 0.24 | 0.01 | 0.37 | N/A | 0.33 | 0.04 | 0.97 | N/A | 0.96 | 0.01 | 86.48 | N/A | 86.40 | 0.07 |
| RSC | 0.26 | N/A | 0.25 | 0.01 | 0.76 | N/A | 0.74 | 0.02 | 2.02 | N/A | 2.00 | 0.02 | 194.82 | N/A | 194.74 | 0.08 |
| SCORE+ | 0.26 | N/A | 0.25 | 0.01 | 0.77 | N/A | 0.75 | 0.02 | 1.82 | N/A | 1.80 | 0.02 | 188.81 | N/A | 188.74 | 0.07 |
| ISC | 0.28 | N/A | 0.27 | 0.01 | 0.61 | N/A | 0.59 | 0.02 | 1.88 | N/A | 1.86 | 0.02 | 202.37 | N/A | 202.25 | 0.12 |
| GE | 9.25 | N/A | N/A | N/A | 20.98 | N/A | N/A | N/A | 133.09 | N/A | N/A | N/A | OOM | N/A | N/A | N/A |
| GAP | 58.85 | N/A | N/A | N/A | 134.81 | N/A | N/A | N/A | 301.02 | N/A | N/A | N/A | OOT | N/A | N/A | N/A |
| SDCN | 13.46 | N/A | N/A | N/A | 333.90 | N/A | N/A | N/A | 139.26 | N/A | N/A | N/A | OOM | N/A | N/A | N/A |
| MCP | 125.84 | N/A | N/A | N/A | 112.09 | N/A | N/A | N/A | 115.71 | N/A | N/A | N/A | OOM | N/A | N/A | N/A |
| DMoN | 105.27 | N/A | N/A | N/A | 792.56 | N/A | N/A | N/A | 625.64 | N/A | N/A | N/A | OOM | N/A | N/A | N/A |
| DGC | 59.54 | N/A | N/A | N/A | 634.21 | N/A | N/A | N/A | 634.89 | N/A | N/A | N/A | OOM | N/A | N/A | N/A |
| **ASCENT** | 0.30 | 0.02 | 0.27 | 0.01 | 0.93 | 0.16 | 0.73 | 0.04 | 2.49 | 0.43 | 2.04 | 0.02 | 240.83 | 45.56 | 195.18 | 0.09 |

consistency between the two sources (Newman & Clauset, 2016; Qin et al., 2018; Wang et al., 2020; Qin & Lei, 2021).

**Learnable Node-wise Corrections** $\{\tau_i\}$. In ASCENT, we still manually set the node-wise corrections $\{\tau_i\}$ by adjusting hyper-parameters $\{\theta, L\}$. We plan to extend it to a more advanced setting with learnable node-wise corrections $\{\tau_i\}$ and provide theoretical analysis combined with recent advances in GNNs

**Better Efficiency and Scalability**. As demonstrated in our efficiency analysis (cf. Appendix I), ED is the major bottleneck of ASCENT. We intend to further improve the efficiency and scalability of this bottleneck using the advanced Locally Optimal Block Preconditioned Conjugate Gradient (LOBPCG) solver (Knyazev, 2001; Zhuzhunashvili & Knyazev, 2017) and consider its parallel implementations (Yamada et al., 2022).

**Other Graph Clustering Objectives**. In this study, we only considered the conductance minimization objective (or equivalently normalized cut minimization) as defined in **Definition 1**. Spectral clustering can be considered as an approximated algorithm for a relaxed version of this objective. Some graph clustering algorithms may consider other objectives (e.g., ratio-cut minimization (Von Luxburg, 2007) and modularity maximization (Newman, 2006; Yu & Ding, 2010; Qin et al., 2024)) that have relations close to conductance minimization. We also plan to further extend our analysis to these objectives.

**Improved Analysis with Looser Conditions**. As discussed in Section 3, the condition in **Theorem 8** implies an assumption that (i) $G$ is well-clustered and (ii) the degree heterogeneity is not so high. It is possible for a given graph $G$ that this condition may not hold. In our future work, we intend to further improve this condition by extending some new theoretical results on the combinatorial optimization problem of graph-cut minimization (e.g., conductance minimization in this paper) to DCSC.

Table 30: Evaluation of runtime↓ (sec) with $K = 32$ on datasets without ground-truth.

| | Airport | | | | Wiki | | | | BlogCatalog | | | | ogbn-Protein | | | |
|---|---|---|---|---|---|---|---|---|---|---|---|---|---|---|---|---|
| | **Total** | $\tau$ | ED | $K$M | **Total** | $\tau$ | ED | $K$M | **Total** | $\tau$ | ED | $K$M | **Total** | $\tau$ | ED | $K$M |
| NJW | 0.38 | N/A | 0.37 | 0.01 | 1.00 | N/A | 0.96 | 0.04 | 2.10 | N/A | 2.04 | 0.07 | 201.61 | N/A | 201.21 | 0.40 |
| SCORE | 0.31 | N/A | 0.30 | 0.01 | 0.42 | N/A | 0.39 | 0.03 | 1.18 | N/A | 1.15 | 0.03 | 98.87 | N/A | 98.61 | 0.26 |
| RSC | 0.34 | N/A | 0.33 | 0.01 | 0.82 | N/A | 0.79 | 0.03 | 2.31 | N/A | 2.20 | 0.11 | 189.30 | N/A | 189.07 | 0.23 |
| SCORE+ | 0.32 | N/A | 0.31 | 0.01 | 0.98 | N/A | 0.91 | 0.07 | 2.17 | N/A | 2.11 | 0.07 | 204.08 | N/A | 203.70 | 0.38 |
| ISC | 0.31 | N/A | 0.30 | 0.01 | 0.73 | N/A | 0.68 | 0.05 | 2.07 | N/A | 2.02 | 0.05 | 257.38 | N/A | 256.94 | 0.44 |
| GE | 12.92 | N/A | N/A | N/A | 20.19 | N/A | N/A | N/A | 135.09 | N/A | N/A | N/A | OOM | N/A | N/A | N/A |
| GAP | 74.08 | N/A | N/A | N/A | 178.94 | N/A | N/A | N/A | 450.36 | N/A | N/A | N/A | OOT | N/A | N/A | N/A |
| SDCN | 18.92 | N/A | N/A | N/A | 343.42 | N/A | N/A | N/A | 161.03 | N/A | N/A | N/A | OOM | N/A | N/A | N/A |
| MCP | 126.05 | N/A | N/A | N/A | 109.82 | N/A | N/A | N/A | 112.79 | N/A | N/A | N/A | OOM | N/A | N/A | N/A |
| DMoN | 105.31 | N/A | N/A | N/A | 798.33 | N/A | N/A | N/A | 763.28 | N/A | N/A | N/A | OOM | N/A | N/A | N/A |
| DGC | 60.29 | N/A | N/A | N/A | 633.89 | N/A | N/A | N/A | 638.75 | N/A | N/A | N/A | OOM | N/A | N/A | N/A |
| **ASCENT** | 0.38 | 0.02 | 0.35 | 0.01 | 1.10 | 0.16 | 0.90 | 0.04 | 2.78 | 0.40 | 2.34 | 0.04 | 246.39 | 44.02 | 202.04 | 0.33 |

