# OpenReview forum: "Rethinking Degree-Corrected Spectral Clustering: a Pure Spectral Analysis & Extension"
_ICLR.cc/2025/Conference — Submitted to ICLR 2025_

### Official Review · Reviewer_hGxZ · 2024-10-31

**Soundness:** 2
**Presentation:** 1
**Contribution:** 2
**Rating:** 5
**Confidence:** 3

**Summary:**

This paper presents an alternative analysis of degree-corrected spectral clustering (DCSC) from a spectral perspective. The authors derive theoretical bounds on the mis-clustered volume relative to an optimal solution for conductance minimization. In addition, they propose ASCENT, an extension of DCSC that uses a node-wise correction scheme, which aims to improve clustering quality by adaptively correcting node-specific degrees. The authors validate ASCENT against established spectral clustering and DCSC baselines, reporting improved results on synthetic and real-world datasets.

**Strengths:**

1. The pure spectral approach is interesting and sidesteps the usual model assumptions, which makes the results more broadly applicable.
2. I do appreciate the ASCENT's node-wise correction scheme, which shows the potential for handling challenging regimes, especially with severe degree heterogeneity.
3. The authors test the proposed algorithm on a range of datasets, demonstrating that it can outperform many traditional methods, which strengthens the practical utility.

**Weaknesses:**

1. The paper's notation is dense and at times difficult to follow. For example, definitions for $E(S, V \setminus S)$ and $\mu(S)$ should be provided as separate statements above Definition 1 to improve clarity;  $m_K$ is used in (5) before it is defined in (6), which breaks the flow; In Proposition 9, $\Phi$ is redefined with a minor variation ($\tau_{max}$), which could be better handled by using a different symbol or subscript to make the distinction clearer.
2. Some parts of the derivations, especially in Section 3, appear redundant. For example, the sentence following Lemma 6 that explains the proof idea seems unnecessary since the lemma is cited from prior work. Overall, the buildup before Theorem 8 (the main result) feels too lengthy; some of these derivations could be relocated to the appendix. The technical depth in these derivations does not appear to add significant new insights unless I’ve overlooked a non-trivial component.
3. As previously mentioned, the ASCENT algorithm and its node-wise correction scheme should be the highlight of this paper. However, the theoretical results in Section 4 don’t convincingly showcase the advantages of ASCENT. Proposition 9, for example, does not demonstrate the benefit of the iterative correction step, and Theorem 10 feels somewhat disconnected, reading more like a heuristic explanation of Figure 1. I recommend replacing Theorem 10 with a result that ties more directly to the ASCENT algorithm, or alternatively, presenting a simpler explanation in text form.
4. Selecting suitable values for parameters $(\theta, L, K)$ could be challenging in practice. While $L=3$ seems effective in these experiments, a more adaptive approach or further guidance on parameter selection for real-world datasets would be helpful.
5. The paper contains several typographical errors, and a thorough proofreading is recommended. Examples include: (1) Line 100: (iii) (iv) should be (i) and (ii); (2) Line 154: $1 > \lambda_1 \le \lambda_2 \cdots $ (3) Line 1163: The sentence ending in "..in Fig." is incomplete (4) "Eigenvalue decomposition" could be replaced with "eigen-decomposition," which is the more common term.

**Questions:**

See weakness part. Also
1. What is the rationale for selecting $K \in \{2, 8, 32\}$ in the experiments?
2. The range of $\theta$ values tested in the experiments seems broad. Could you discuss whether the algorithm is sensitive to the choice of $\theta$?

---

> ### Author Response · Authors · 2024-11-24
> **Response to Comments of Reviewer hGxZ (1)**
>
> Dear Reviewer hGxZ,
>
> We sincerely thank you for summarizing the strengths of our paper. In addition, we have also tried our best to make a major revision according to your comments and suggestions on the weaknesses.
>
> ***
> >**W1**. The paper's notation is dense and at times difficult to follow ...
>
> **Response to W1**: We are sorry for the presentation issue regarding notations. During the revision, we have adopted your suggestion to give separate definitions for $E(S, V \backslash S)$ and $\mu (S)$ before **Definition 1**. Moreover, we have given the definition of $m_K$ just after it is first used in (5). In the analysis of the main paper, we have used $\Psi_{\rm{ISC}}$ and $\Psi_{\rm{AST}}$ to denote the $\Psi$ quantities of ISC and ASCENT, respectively. Moreover, we have further extended our analysis to a series of spectral clustering algorithms in **Appendix F**, where we used the subscripts (or superscripts) of 'NJW', 'RSC', 'SC+', 'ISC', and 'AST' to denote corresponding variables of NJW, RSC, SCORE+, ISC, and ASCENT, respectively.
>
> We hope that our revisions can help you better follow the analysis.
>
> ***
> >**W2**. Some parts of the derivations, especially in Section 3, appear redundant ...
>
> **Response to W2**: We are sorry for the redundant presentation. However, **Reviewer g4ng** also suggested to add some more explanations in **Section 3** (e.g., the high-level sketch in **Fig. 1** of the revised paper), in addition to the redundant presentation you mentioned. In this sense, some redundant derivations and explanations may be necessary for those who lack related backgrounds to follow the logical flows of our theoretical analysis.
>
> We really hope that you can understand our difficulties in reaching a presentation appropriate for readers with different backgrounds.
>
> ***
> >**W3**. As previously mentioned, the ASCENT algorithm and its node-wise correction scheme should be the highlight of this paper. However, the theoretical results in Section 4 don’t convincingly showcase the advantages of ASCENT ... I recommend replacing Theorem 10 with a result that ties more directly to the ASCENT algorithm, or alternatively, presenting a simpler explanation in text form.
>
> **Response to W3**: Thank you for your comments regarding our analysis of ASCENT. During the revision, we have adopted your suggestion to replace Proposition 9 and Theorem 10 with an improved analysis (cf. **Proposition 9** in the revised paper) and given the corresponding proof in **Appendix E**. Instead of simply replacing the constant $\tau$ with the maximum correction $\tau_{\max}$ of ASCENT (i.e., our previous analysis), **the upper bounds in our improved analysis are related to the maximum correction $\hat\tau_r$ among nodes in each cluster $\hat S_r$**. Since different nodes $\{ v_i \}$ may have different corrections $\{\tau_i\}$, **different clusters $\{ \hat S_r\}$ may have different $\{\hat \tau_r\}$** and thus **the upper bounds of $\mu (C_r \Delta \hat S_r)$ and $\phi (C_r)$ are determined by cluster-wise corrections $(\hat \tau_1, \cdots, \hat \tau_K)$**. As demonstrated in **Fig. 2** (of the revised paper), **ASCENT can force nodes $\{ v_i\}$ in the same cluster to have close corrections $\{ \tau_i \}$**. Based on this property, we believe that the improved analysis with cluster-wise corrections $(\hat \tau_1, \cdots, \hat \tau_K)$ can better demonstrate the advantages of ASCENT, compared with the previous version with $\tau_{\max}$.
>
> Moreover, based on our extended analysis in **Appendix F**, we have further tried to answer the question: **When can ASCENT potentially outperform ISC**? It further verifies that under certain conditions, **it is possible for ASCENT to outperform some conventional DCSC algorithms** (e.g., ISC).
>
> We really hope that our revisions and explanations can help you better understand the advantages of our method and address your concerns.

---

> ### Author Response · Authors · 2024-11-24
> **Response to Comments of Reviewer hGxZ (2)**
>
> >**W4**. Selecting suitable values for parameters $(\theta, L, K)$ could be challenging in practice ... a more adaptive approach or further guidance on parameter selection for real-world datasets would be helpful.
>
> >**Q2**. The range of $\theta$ values tested in the experiments seems broad. Could you discuss whether the algorithm is sensitive to the choice of $\theta$?
>
> **Response to W4 and Q2**: Please note that for most graph clustering methods (including ASCENT and other baselines in our experiments), $K$ is assumed to be given, as described in the **2nd paragraph of Section 2**. Hence, we usually do not need to select $K$ in practice.
>
> From the perspective of machine learning, $(\theta, L)$ are hyper-parameters for ASCENT, just like the number of GNN layers in some deep graph clustering baselines (e.g., GAP, SDCN, MinCutPool, DMoN, and DGCluster in our experiments). Moreover, the constant correction $\tau$ in RSC, SCORE+, and ISC can also be considered as a tunable hyper-parameter. Since graph clustering is a typical unsupervised task, one can tune hyper-parameters of a method using the unsupervised conductance metrics. Concretely, we can compute conductance metrics w.r.t. clustering results given by different hyper-parameter settings and then select the one with minimum conductance.
>
> As described in **Appendix I** of the revised paper, we recommend to adjust $\theta \in \{ 0.01, 0.05, 0.1, 0.5, 1\}$ for ASCENT. According to the parameter analysis results in **Fig. 4** of the revised paper, the clustering quality of ASCENT is sensitive to $\theta$.
>
> We believe that ASCENT is a good beginning to consider node-wise corrections $\{ \tau_i \}$ for DCSC, even though we still manually set $\{ \tau_i \}$ by adjusting $\{ \theta, L\}$. As summarized in **Appendix J** of the revised paper, we intend to consider a more advanced setting with learnable $\{ \tau_i \}$ instead of manually adjusting $\{ \theta, L\}$ in our future work.
>
> ***
> >**W5**. The paper contains several typographical errors, and a thorough proofreading is recommended ...
>
> **Response to W5**: We are sorry for the errors you mentioned. During the revision, we have corrected them with the corresponding revisions highlighted in **blue text**. For instance, we have (1) changed '(iii) degree heterogeneity and (iv) weakness of clustering structures' to '**(i)** degree heterogeneity and **(ii)** weakness of clustering structures', (2) changed '$1 > \lambda_1 \le \lambda_2 \cdots$' to '$1 > \lambda_1 \ge \lambda_2 \cdots \lambda_N$', (3) completed the sentence to 'example analysis results of $\theta$ on Caltech are visualized in **Fig. 3**', and (4) replaced 'eigenvalue decomposition' with '**eigen-decomposition**'. Besides, we have also tried our best to find some other potential errors and correct them.
>
> ***
> >**Q1**. What is the rationale for selecting $K \in \{ 2, 8, 32\}$ in the experiments?
>
> **Response to Q1**: Please note that in our previous version, we have already explained why we set $K \in \{ 2, 8, 32\}$ for Airport, Wiki, BlogCatalog, and ogbn-Protein in the last paragraph of **Section 5.1** and **Appendix H**.
>
> In our experiments, CalTech, Simmons, PolBlogs, and BioGrid are widely-used datasets that provide explicit clustering ground-truth w.r.t. our problem statement but with relatively small scales. Therefore, we could directly set $K$ according to their ground-truth (e.g., $K = 8$ for CalTech as summarized in **Table 3**) while using NMI and AC to measure the correspondence between (i) the clustering result given by a method and (ii) the ground-truth.
>
> To further test the efficiency and scalability of ASCENT, we also used additional datasets (i.e., Airport, Wiki, BlogCatalog, and ogbn-Protein) with relatively larger scales. However, as described in **Appendix H** of the revised paper, **these four datasets do not provide ground-truth w.r.t. our problem statement** (i.e., **disjoint graph clustering without attributes**). Concretely, the original ground-truth of Wiki and BlogCatalog is overlapping but not disjoint. Airport and ogbn-Protein are attributed graphs. It is unclear that their ground-truth is dominated by graph topology or attributes due to the complicated correlations between the two sources described in **Section 2**. In this sense, **we could not directly use their ground-truth for evaluation but had to manually set $K \in \{ 2, 8, 32\}$ for them and compared their corresponding conductance values**. It is also a widely-used strategy for graph clustering (community detection) when ground-truth is unavailable [1-2].
>
> [1] Dhillon et al. Weighted Graph Cuts without Eigenvectors: A Multilevel Approach.
>
> [2] Chan et al. A Convex Formulation of Modularity Maximization for Community Detection.

---

> > ### Comment · Reviewer_hGxZ · 2024-11-26
> >
> > Thank you for your responses and revisions. While the updates, such as clarifying key notations and improving the theoretical analysis, are valuable, the writing remains a key issue. The notation is still dense, and the overall presentation is overly complex, making it difficult for readers unfamiliar with the topic to follow. Additionally, the organization could be more streamlined to improve accessibility and readability for a broader audience. For these reasons, I will maintain my score. Thank you for your efforts.

---

> ### Author Response · Authors · 2024-11-27
> **Response to Further Comments of Reviewer hGxZ**
>
> Dear Reviewer hGxZ,
>
> We sincerely appreciate your positive feedbacks regarding our responses and revisions.
>
> We are sorry for the dense presentation of our paper. As we can check, **Reviewer VrzY** gave positive feedbacks regarding the presentation of previous version. **Reviewer g4ng** further suggested to added some more explanations regarding theoretical analysis. It indicates that **people with different backgrounds may have different opinions about the presentation of a paper with rigorous theoretical analysis**. We have tired our best to make a compromise among all the reviewers' comments.
>
> We really hope that you can understanding our difficulties in reaching a presentation appropriate for readers with different backgrounds while following the page limit of main paper.

---

### Official Review · Reviewer_2rdr · 2024-11-01

**Soundness:** 3
**Presentation:** 2
**Contribution:** 3
**Rating:** 8
**Confidence:** 3

**Summary:**

The paper explores advancements in graph clustering, specifically focusing on the Degree-Corrected Spectral Clustering (DCSC) method. The authors critique traditional analyses of DCSC, which often rely on random graph models like SBM. They propose an alternative analysis framework based on spectral theory alone, presenting new metrics for assessing mis-clustered volumes and analyzing the algorithm's ability to handle diverse node degrees and weak clustering structures.

A key contribution is also the introduction of a novel clustering algorithm, ASCENT (Adaptive Spectral Clustering with Node-wise Correction), which differs from DCSC by applying individualized correction values to each node rather than a single global correction factor. This node-wise adjustment, inspired by techniques in Graph Neural Networks (GNNs), helps ASCENT overcome issues related to graph over-smoothing. In tests, ASCENT demonstrated superior performance over both conventional spectral clustering and DCSC approaches in various scenarios involving high degree heterogeneity and weak clustering structures.

**Strengths:**

Traditional analyses of DCSC often depend on assumptions from random graph models, like the Stochastic Block Model (SBM), to derive theoretical guarantees. The authors propose a new approach by analyzing DCSC solely through spectral graph theory. This analysis provides an upper bound on the mis-clustered volume relative to an optimal solution, without relying on random graph models.

The ASCENT algorithm seems promising as a simple improvement over DCSC.

**Weaknesses:**

1) Over-Smoothing Control: Although ASCENT attempts to mitigate over-smoothing, the choice of the parameter L (number of correction iterations) is critical. Incorrectly setting L could lead to suboptimal clustering, and a more dynamic method for determining this parameter might enhance robustness.

2) Complexity and Scalability: Although ASCENT aims to address high degree heterogeneity and weak clustering structures, node-wise correction might increase computational complexity, especially on large-scale graphs with millions of nodes. The over-smoothing issue addressed by ASCENT could still emerge in massive networks where iterative corrections might become infeasible.

3) Initialisation in ASCNET: Since initial corrections are based on node degrees, ASCENT’s performance could vary depending on the graph’s degree distribution. In graphs with degree heterogeneity, initial corrections may differ widely, and ASCENT’s iterative process may need more iterations to bring node corrections to a stable, locally consistent state, or possibly oscillate ?

4) The presentation is a bit dense and it could useful to try to re-organize a bit the paper to make it more more reader-firendly.

**Questions:**

The questions are related to the identified weaknesses:
1) Can you effectively  do cross validation to find L?
2) How would you consider scaling the algorithm to big graphs?
3) It would be useful to define over smoothing more rigorously, especially in the context of ASCENT which is pretty different from GNN?
4) The paper also notes that, after enough iterations, ASCENT effectively reduces to a conventional DCSC method with a global, constant correction term, indicating that the algorithm reaches a state where further updates do not change the corrections meaningfully. Is there  a convergence guarantee in the strict mathematical sense? Are there possible guarantees  against oscillations?
5) Maybe give more intuition on crucial parameters like psi

---

> ### Author Response · Authors · 2024-11-24
> **Response to Comments of Reviewer 2rdr (1)**
>
> Dear Reviewer 2rdr,
>
> We sincerely thank you for summarizing the strengths of our paper. Besides, we have also tried our best to make a major revision according to your concerns and suggestions on the weaknesses.
>
> ***
> >**W1**. Over-Smoothing Control: Although ASCENT attempts to mitigate over-smoothing, the choice of the parameter L (number of correction iterations) is critical ...
>
> >**Q1**. Can you effectively do cross validation to find L?
>
> **Response to W1 & Q1**: Please note that $L$ is a hyper-parameter of ASCENT, like the number of GNN layers in some deep graph clustering approaches (e.g., GAP, SDCN, MinCutPool, DMoN, and DGCluster in our experiments). Since graph clustering is a typical unsupervised task, one cannot use the cross-validation designed for supervised tasks to tune $L$. Instead, **we can use the unsupervised conductance metric to tune $L$**, as spectral clustering is an approximated algorithm for conductance minimization. Concretely, we can **compute conductance metrics w.r.t. clustering results given by different settings of $L$** and **select the $L$ that has the minimum conductance**. Such a strategy is also adopted by other baselines in our experiments to tune their hyper-parameters.
>
> We believe that ASCENT is a good beginning to consider node-wise corrections for DCSC. As summarized in **Appendix J** of the revised paper, we intend to **extend ASCENT to a setting with learnable node-wise corrections $\{ \tau_i \}$** in our future work, instead of manually setting $\{ \tau_i \}$ by adjusting $\{ \theta, L\}$.
>
> ***
> >**W2**. Complexity and Scalability: Although ASCENT aims to address high degree heterogeneity and weak clustering structures, node-wise correction might increase computational complexity, especially on large-scale graphs with millions of nodes ...
>
> >**Q2**. How would you consider scaling the algorithm to big graphs?
>
> **Response to W2 & Q2**: Thank you for your comment and question regarding the complexity and scalability of ASCENT. During the revision, we have provided **complexity analysis** in **Appendix G**, where ***the derivation of node-wise corrections $\{ \tau_i\}$ will not increase the overall complexity of ASCENT***. Namely, ***ASCENT has the same complexity with most existing DCSC algorithms***.
>
> We have also conducted additional **efficiency analysis** for all the methods in **Appendix I**. Our results (cf. **Tables 27-30**) demonstrate that ***spectral clustering methods (including ASCENT) can achieve significantly better efficiency and scalability than deep graph clustering baselines***. Moreover, ***eigen-decomposition is the major bottleneck for all the spectral clustering algorithms*** and ***the derivation of $\{ \tau_i\}$ would not significantly increase the overall runtime of ASCENT***.
>
> To further improve the scalability of ASCENT, we need to first improve its bottleneck (i.e., eigen-decomposition). One can speed up eigen-decomposition using the efficient LOBPCG solver, which also has parallel implementations. As summarized in **Appendix J**, we plan to consider this improvement in our future work.
>
> ***
> >**W3**. Initialisation in ASCNET: ... In graphs with degree heterogeneity, initial corrections may differ widely, and ASCENT’s iterative process may need more iterations to bring node corrections to a stable, locally consistent state, or possibly oscillate?
>
> **Response to W3**: One of our contributions is that **ASCENT can derive node-wise corrections $\{ \tau_i \}$ (i.e., different nodes may have different corrections) before it reduces to the case with a constant correction $\tau$ due to over-smoothing**. Our experiments also demonstrate that **ASCENT can potentially obtain better clustering quality** than conventional DCSC methods with a constant $\tau$ **in some early steps of iteration before over-smoothing**.
>
> In this sense, **it does not matter that ASCENT may need more iterations to bring node corrections to a stable state** as you mentioned. We really hope that our explanations can help you better understand our contributions of ASCENT.
>
> ***
> >**W4**. The presentation is a bit dense and it could useful to try to re-organize a bit the paper to make it more more reader-friendly.
>
> **Response to W4**: We are sorry for the dense presentation of main paper, due to some redundant explanations about theoretical analysis. However, **Reviewer g4ng** still suggested to added some more interpretations regarding the analysis in Section 3 (e.g., the high-level sketch in **Fig. 1** of the revised paper), in addition to the previous presentation. In this sense, some redundant derivations and explanations may be necessary for those who lack related backgrounds to follow the logical flows of our theoretical analysis.
>
> We really hope that you can understand our difficulties in reaching a presentation appropriate for readers with different backgrounds while following the page length of the conference (i.e., at most 10 pages for the main paper).

---

> ### Author Response · Authors · 2024-11-24
> **Response to Comments of Reviewer 2rdr (2)**
>
> >**Q3**. It would be useful to define over smoothing more rigorously, especially in the context of ASCENT which is pretty different from GNN?
>
> **Response to Q3**: Thank you for your suggestion regarding the definition of over-smoothing. Despite the strict page limit of main paper, during the revision, we have still tried our best to give a brief but formal definition of the over-smoothing effect of ASCENT in **Section 4**. Concretely, we have $\mathop {\lim }\limits_{l \to \infty } \tau _i^{(l)} = c, \forall {v_i} \in V$, with $c$ as a constant.
>
> ***
> >**Q4**. The paper also notes that, after enough iterations, ASCENT effectively reduces to a conventional DCSC method with a global, constant correction term ... Is there a convergence guarantee in the strict mathematical sense? Are there possible guarantees against oscillations?
>
> **Response to Q4**: Please note that ASCENT reduces to a conventional DCSC algorithm with a constant correction $\tau$, due to the over-smoothing issue of the GNN mean aggregator. Since ASCENT adopts the typical mean aggregator (e.g., same as that of GraphSAGE-mean), **any convergence guarantees regarding the over-smoothing of this aggregator also hold for ASCENT**. To the best of our knowledge, **there is some rigorous theoretical analysis regarding the convergence of over-smoothing (cf. Theorem 1 and Corollary 1 in [1])**. Our toy example in **Fig. 2** of the revised paper also justified this effect.
>
> [1] Keriven et al. Not Too Little, Not Too Much: a Theoretical Analysis of
> Graph (Over)smoothing.
>
> ***
> >**Q5**. Maybe give more intuition on crucial parameters like $\Psi$.
>
> **Response to Q5**: As highlighted before **Theorem 3** of the revised paper, $\Psi$ is **an auxiliary quantity that can measure impacts of (i) degree heterogeneity and (ii) weakness of clustering structures**. Concretely, $\Psi$ includes two variables (i.e., $h$ and $m_K$), where (i) a small $h$ indicates high degree heterogeneity and (ii) a small lower bound of $m_K$ indicates weak clustering structures.
>
> Since **$\Psi$ is inversely proportional to $h$ and $m_K$**, high degree heterogeneity and weak clustering structures may result in a large value of $\Psi$.
>
> Furthermore, **the upper bounds of mis-clustered volume $\mu (C_r \Delta \hat S_r)$ and conductance $\phi (C_r)$ given by Theorem 8 are directly proportional to $\Psi$**. Therefore, (i) higher degree heterogeneity and (ii) weak clustering structures result in higher upper bounds of $\mu (C_r \Delta \hat S_r)$ and $\phi (C_r)$, indicating lower clustering quality.
>
> In this setting, $\Psi$ enables our main theoretical results in **Theorem 8** to **quantitatively reveal the impacts of (i) degree heterogeneity and (ii) weakness of clustering structures to the clustering quality measured by $\mu (C_r \Delta \hat S_r)$ and $\phi (C_r)$**.
>
> We really hope that our explanations can help you better understand the motivation and intuition of introducing $\Psi$.

---

### Official Review · Reviewer_VrzY · 2024-11-02

**Soundness:** 2
**Presentation:** 3
**Contribution:** 2
**Rating:** 3
**Confidence:** 3

**Summary:**

The paper proposes an extension of the degree-corrected spectral clustering algorithm, ASCENT, from a spectral perspective to address networks with high degree heterogeneity or weak clustering structures.  The method assigns different corrections for nodes via
the mean aggregation of GNNs, instead of constant degree correction.  Theoretically, the paper gives a rigorous bound for the mis-clustered volume.

**Strengths:**

1) Without relying on specific random graph models,  the paper provides a rigorous bound on the mis-clustered volume.
2) Being able to assign different corrections for different nodes.
3) The experiment identifies some interesting phenomena relates to 'early stages'.

**Weaknesses:**

The theoretical results on the mis-clustered volume (i.e. Theorem 3 and Theorem 8) are nearly identical to Theorem 1 and Theorem 5 in the paper 'Fangmeng Liu, Wei Li, and Yiwen Zhong. A further study on the degree-corrected
spectral clustering under spectral graph theory. Symmetry, 14(11):2428, 2022.' . Additionally, the structure of the current paragraph closely mirrors that of the previous paper. However, the prior work is not cited, which raises concerns regarding proper attribution. As a result, the novelty of this paper is limited, and it would benefit from further clarification on how it advances the existing literature.

The paper provides insufficient detail regarding the tuning of the parameter $\theta$, lacking a clear explanation of how it was chosen during the data analysis process. Furthermore, the experimental results in Section 5.1 show only a slight improvement over the baseline methods. This raises concerns that if an inappropriate value for $\theta$ is selected, the proposed approach may offer no noticeable advantage over existing spectral clustering methods.

**Questions:**

In Theorem 8, the assumption of an upper bound on $\Psi$ is made without providing any justification. It would be beneficial to offer an intuitive explanation for this assumption and validate the upper bound using both simple network structures (e.g. the Erdos-Renyi model) and real-world networks to demonstrate its practical relevance. Additionally, theoretical analysis or sufficient conditions that highlight the universality of this assumption would further strengthen the argument.

 The theoretical bound for the mis-clustered volume is derived based on the solution of minimal conductance, which is an unconventional approach in network clustering analysis. It would be helpful to explain whether this mis-cluster rate is valid in practical applications. Ideally, the mis-cluster rate should be evaluated with respect to the ground truth, as this is a more commonly accepted and meaningful metric.

The limitation on the number of iteration steps in the ASCENT model is introduced to address over-smoothing issues. However, the impact of neglecting the over-smoothing problem on clustering results remains unclear. The original degree-corrected spectral clustering (DCSC) method already performs well, and the proposed approach shows only slight improvements by adding two extra parameters, $L$ and $\theta$. If the tuning parameters does not selected optimally, the performance may be inferior than that of the original DCSC method. Also it is unclear whether the performance of the method is sensitive to the tuning parameters.

 The authors should provide an analysis of the computational complexity of the proposed method ASCENT. Additionally, a comparison of ASCENT with other existing approaches in terms of runtime would be beneficial. Given that the proposed method involves iterative steps for the correction of node-wise parameters $\{\tau_i\}_{i=1}^n$, this analysis is necessary for assessing the efficiency and practicality of the approach.

The authors should justify the use of the mean aggregation operation in the computation of the node-wise parameters $\{\tau_i\}_{i=1}^n$. Additionally, please explain the rationale behind employing a GNN-based graph clustering method within the context of spectral clustering analysis.

---

> ### Author Response · Authors · 2024-11-24
> **Response to Comments of Reviewer VrzY (1)**
>
> Dear Reviewer VrzY,
>
> We sincerely thank you for summarizing the strengths of our paper. In addition, we have also tried our best to make a major revision of our paper according to your concerns and suggestions on the weaknesses.
>
> ***
> >**W1**. The theoretical results on the mis-clustered volume (i.e. Theorem 3 and Theorem 8) are nearly identical to Theorem 1 and Theorem 5 in the paper Fangmeng Liu et al. ... it would benefit from further clarification on how it advances the existing literature.
>
> **Response to W1**: Thank you for providing a related work on *RSC*. After checking the paper you mentioned, **we conclude that our analysis is more generic than this work**. As summarized at the beginning of **Section 3**, we adopt *ISC* as an example for analysis, which has a more generic formate than *RSC*. During the revision, **we have further provided a new upper bound for conductance** $\phi (C_r)$ in **Theorem 8**, besides that of mis-clustered volume $\mu (C_r \Delta {\hat S}_r)$. In contrast, **the paper you mentioned only does not include such analysis**.
>
> To demonstrate the generality of our analysis, we have also provided further analysis on DCSC in **Appendix F**, where we reduced our main theoretical results to NJW, RSC, and SCORE+. In this sense, **our analysis serves as a unified framework involving a series of spectral clustering algorithms**. Based on this framework, we further try to answer three questions. (i) *Why can ISC potentially outperform RSC*? (ii) *When can a DCSC algorithm (e.g., RSC) potentially outperform vanilla spectral clustering (i.e., NJW)*? (iii) *When can ASCENT potentially outperform ISC*?
>
> To the best of our knowledge, **most existing studies (including the paper you mentioned) do not include the aforementioned generic analysis**. During the revision, we have adopted your suggestion to cite the paper you mentioned in **Section 1** and highlighted the aforementioned contributions beyond it.
>
> Please also note that in addition to our analysis, we also proposed ASCENT, an extension of DCSC with node-wise corrections $\{ \tau_i \}$ and explored its potential to achieve better clustering quality. To the best of our knowledge, **we are the first to consider DCSC with node-wise corrections $\{ \tau_i \}$ determined by the mean aggregation mechanism of GNNs**. In particular, ASCENT reduces to existing DCSC methods (e.g., ISC) when encountering the over-smoothing of mean aggregation.
>
> We really hope that you can re-evaluate our paper by comprehensively considering our contributions, including the **unified analysis** and the **extension of DCSC** (i.e., ASCENT).

---

> ### Author Response · Authors · 2024-11-24
> **Response to Comments of Reviewer VrzY (2)**
>
> >**W2**. The paper provides insufficient detail regarding the tuning of the parameter $\theta$, lacking a clear explanation of how it was chosen during the data analysis process ... the experimental results in Section 5.1 show only a slight improvement over the baseline methods.
>
> >**Q3**. The limitation on the number of iteration steps in the ASCENT model is introduced to address over-smoothing issues ... The original degree-corrected spectral clustering (DCSC) method already performs well, and the proposed approach shows only slight improvements by adding two extra parameters $L$ and $\theta$.
>
> **Response to W2 & Q3**: Thank you for your comments regarding the parameter setting and quality improvement of ASCENT.
>
> From the perspective of machine learning, $(\theta, L)$ are hyper-parameters for ASCENT, just like the number of GNN layers in some deep graph clustering baselines (e.g., GAP, SDCN, MinCutPool, DMoN, and DGCluster in our experiments). Moreover, the constant correction $\tau$ in RSC, SCORE+, and ISC can also be considered as a tunable hyper-parameter. Since graph clustering is a typical unsupervised task, **one can tune hyper-parameters of a method (e.g., ASCENT and other graph clustering baselines) using the unsupervised conductance metrics**. Concretely, we computed the conductance metrics w.r.t. the clustering results given by different hyper-parameter settings and then selected the setting with minimum conductance. During the revision, we have added the aforementioned details in the last paragraph of **Section 5.1**. We have further provided results of parameter analysis in **Fig. 3** and **Fig. 4**.
>
> Please also note that **our ASCENT method can achieve significant improvement in some cases**, e.g., improvement of more than ~3% on **LFR-2** and ~5% on **ogbn-Protein** as highlighted in **Table 5** and **Table 8**. Our main purpose of the extension of node-wise corrections (i.e., ASCENT) is to **explore whether it can potentially achieve better clustering quality**, which has not been considered by most existing methods. To some extent, **our experiments validate such a possibility and provide some interesting phenomena** as you mentioned in **S3**.
>
> Therefore, we believe that ASCENT is a good beginning to consider node-wise corrections $\{ \tau_i \}$ for DCSC. As summarized in **Appendix J**, we plan to consider an advanced extension with learnable $\{ \tau_i \}$, instead of manually setting them by adjusting $\{ \theta, L\}$ in our future work.

---

> ### Author Response · Authors · 2024-11-24
> **Response to Comments of Reviewer VrzY (3)**
>
> >**Q1**. In Theorem 8, the assumption of an upper bound on $\Psi$ is made without providing any justification. It would be beneficial to offer an intuitive explanation for this assumption ...
>
> **Response to Q1**: We are sorry for the insufficient explanations regarding the condition of $\Psi$ in Theorem 8. During the revision, we have adopted your suggestion to give additional explanations for this condition at the end of **Section 3**.
>
> Concretely, to ensure that the condition $\Psi \le 1/[132(1+\lambda_1)^2\alpha {\tilde \mu} K]$ holds, we need small $\bar \phi_K (G) / (1 - \lambda_{K+2})$ and large $d_{\min} / [(d_{\max + \tau}) (1 - \lambda_{K+2})]$. In some early studies on vanilla spectral clustering (e.g., Peng et al. and Mizutani et al. in our paper), a graph is defined to be **well-clustered** if $\bar \phi_K(G) / (1 - \lambda_{K+1})$ is sufficiently small. It is consistent with that $\bar \phi_K (G) / (1 - \lambda_{K+2})$ is small. The well-clustered assumption adopted by Peng et al. and Mizutani et al. indicates that the optimal solution $(\hat S_1, \cdots, \hat S_K)$ (i.e., clustering ground-truth in real applications) describes an explicit clustering structure of $G$. Our analysis can be considered as **a unified extension of these early studies**. Moreover, large $d_{\min} / [(d_{\max + \tau}) (1 - \lambda_{K+2})]$ indicates that the degree heterogeneity should not be very high. In summary, the condition in **Theorem 8** implies an implicit assumption that (i) $G$ is well-clustered and (ii) the degree heterogeneity is not so high.
>
> During the revision, we have also considered your suggestion to verify this condition on both synthetic and real graphs. However, we encountered the following difficulties in exactly computing $\Psi$ and its upper bound. Please note that this condition includes variables (e.g., $\bar \phi_K (G)$, $\mu_{\min}$, and $\mu_{\max}$) associated with the optimal solution to conductance minimization. It is well-known that conductance minimization is an NP-hard combinatorial optimization problem, so we could not expect a polynomial-time algorithm to obtain its optimal solution. It is intractable to get the optimal solution using the brute-force algorithm with a complexity of $O(N^K)$. Although we can still 'approximately' treat the ground-truth of a dataset as the optimal solution, we also found that the upper bound includes the approximation ratio $\alpha$ of KMeans, which depends on the concrete algorithm we use and is usually not an exact number (e.g., $\alpha = O(log K)$ for KMeans++). In summary, we could not exactly compute $\Psi$ and its upper bound to verify the condition. We really hope that you can understand the difficulties we encountered.
>
> Also as summarized in **Appendix J** of the revised paper, we plan to further improve this condition by extending some new theoretical results on the combinatorial optimization problem of graph-cut minimization (e.g., conductance minimization in this paper) to DCSC.

---

> ### Author Response · Authors · 2024-11-24
> **Response to Comments of Reviewer VrzY (4)**
>
> >**Q2**. The theoretical bound for the mis-clustered volume is derived based on the solution of minimal conductance, which is an unconventional approach in network clustering analysis. It would be helpful to explain whether this mis-cluster rate is valid in practical applications.
>
> **Response to Q2**: It is well known that spectral clustering adopts a relaxed version of the NP-hard combinatorial optimization objective of conductance minimization. From the perspective of combinatorial optimization, **the optimal solution $(\hat S_1, \cdots, \hat S_K)$ to conductance minimization corresponds to the ground-truth you mentioned, in an ideal case**.
>
> Please also note that **the mis-clustered volume $\mu (C_r \Delta {\hat S}_r)$ in Theorem 8 is consistent with the accuracy metric of graph clustering**. Following the same renumbering of $(C_1, \cdots, C_K)$ in **Theorem 8**, the clustering accuracy is defined as $\sum\nolimits_{r = 1}^K {|{C_r} \cap {{\hat S}_r}|} /N$. In particular, we have ${C_r} \cap {\hat S_r} = {C_r} \cup {\hat S_r} - {C_r}\Delta {\hat S_r} \le V - {C_r}\Delta {\hat S_r}$ and thus lower mis-clustered volume $\mu (C_r \Delta {\hat S}_r)$ indicates higher clustering accuracy.
>
> To the best of our knowledge, **to provide bonds regarding the feasible solution given by an algorithm (e.g., $(C_1, \cdots, C_K)$ of ISC) w.r.t. the optimal solution (e.g., $(\hat S_1, \cdots, \hat S_K)$ of conductance minimization) is a widely-used and conventional strategy in the analysis of many combinatorial optimization problems on graphs**. For instance, we have $\rm{OBJ} \le \gamma {\rm{OPT}}$ (i.e., $\gamma$-approximation) for some well-known algorithms of traveling salesman problem (TSP) based on minimum spanning tree [1,2], with OBJ and OPT as objective values of (i) feasible solution given by the algorithm and (ii) optimal solution. Since **spectral clustering is an approximated algorithm for the NP-hard conductance minimization problem**, it is straightforward to adopt this strategy for our analysis of DCSC. Some early studies on vanilla spectral clustering (e.g., Peng et al. and Mizutani et al. in our paper) also follow this strategy. **Our analysis can be considered as a unified extension of these early studies**.
>
> We really hope that the aforementioned explanations can help you better understand that **our analysis follows a conventional strategy in the analysis of combinatorial optimization** and **is valid in practical applications**.
>
> [1] Pop et al. A Comprehensive Survey on the Generalized Traveling Salesman Problem.
>
> [2] Arora et al. Approximation Schemes for NP-hard Geometric Optimization Problems: a Survey
>
> ***
> >**Q4**. The authors should provide an analysis of the computational complexity of the proposed method ASCENT. Additionally, a comparison of ASCENT with other existing approaches in terms of runtime would be beneficial.
>
> **Response to Q4**: During the revision, we have adopted your suggestion to add **complexity analysis** for ASCENT in **Appendix G** and conduct further **efficiency analysis** (in terms of runtime) for all the methods in **Appendix I**, with corresponding results reported in **Tables 27-30**. According to our analysis, we can reach the following conclusions.
>
> - ***The derivation of node-wise corrections $\{ \tau_i \}$ will not increase the overall complexity of ASCENT***. Therefore, ***ASCENT has the same complexity with most conventional DCSC algorithms***.
> - For all the spectral clustering algorithms (including ASCENT), ***eigen-decomposition is the major bottleneck in their runtime***, while ***the derivations of $\{ \tau_i \}$ would not significantly increase the overall runtime of ASCENT***.
> - All the spectral clustering methods (including ASCENT) have significantly better efficiency than other deep graph clustering baselines.

---

> ### Author Response · Authors · 2024-11-24
> **Response to Comments of Reviewer VrzY (5)**
>
> >**Q5**. he authors should justify the use of the mean aggregation operation in the computation of the node-wise parameters ...
>
> **Response to Q5**: Please note that most existing DCSC methods use the 'average/mean' of degrees as their constant correction $\tau$ (e.g., $\tau := \bar d$ and $\delta (d_{\min} + d_{\max})/2$ for RSC and ISC as summarized in **Table 1**). In this sense, **the constant $\tau$ can be considered as a *global* average of degrees**. Based on this interpretation of $\tau$, it is interesting to consider ***local* average of degrees w.r.t. the local topology of each node, which can be computed by the GNN mean aggregation defined in (12)**.
>
> Also as justified by the toy example in **Fig. 2** of the revised paper, (i) $\tau_i$ of each node $v_i$ is initialized by $d_i$ and (ii) **the mean aggregation forces nodes $\{ v_i \}$ with dense linkage (e.g., in the same cluster) to have similar corrections $\{ \tau_i \}$** (i.e., the *local* average degrees). Finally, for a large number of iterations (e.g., $L=50$ in **Fig. 2**), corrections $\{ \tau_i \}$ of all the nodes converge to a constant, which can be considered as a special *global* average degree, and thus reduces to a conventional DCSC method (e.g., ISC) with a constant $\tau$.
>
> In summary, the basic motivation of using the GNN mean aggregation operation is that it can compute the ***local* average w.r.t. the local topology of each node**, which serves as **a special case of the *global* average in conventional DCSC methods with a constant $\tau$**.
>
> We really hope that our explanations can help you better understand the motivation of using mean aggregation.

---

> > ### Comment · Reviewer_VrzY · 2024-11-27
> >
> > Thank you for your detailed responses. I appreciate the clarifications you have provided. However, I still have the following concerns.
> >
> > Firstly, while you mention that Theorem 8 presents a new upper bound for the conductance $\phi(C_r)$, it is hard to see how the new error bound benefits your node-wise correction algorithm. Therefore, the theoretical contribution of the paper may still be limited, and I am having difficulty seeing a substantial difference from the previous work.
> >
> > Secondly, the assumption in Theorem 8 seems rather strict, which might constrain the applicability of the method in practical settings. I wonder if it would be possible to explore more relaxed assumptions or provide sufficient conditions to help better justify or explain the necessity of these assumptions. This could potentially broaden the scope and applicability of the approach.

---

> > > ### Author Response · Authors · 2024-12-02
> > > **Respectfully Disagree with the Assessment of Reviewer VrzY**
> > >
> > > Dear AC,
> > >
> > >   During the rebuttal period, we have taken a great amount of efforts to give responses and revise our paper based on comments of all the reviewers. The corresponding revisions have been highlighted in **blue text** in the newly submitted paper.
> > >
> > >   Although we have highlighted again and again that **our contributions is three-fold** (i.e., (i) pure spectral analysis for DCSC, (ii) a unified framework involving a series of spectral clustering algorithm in Appendix F, and (iii) extension of DCSC with node-wise corrections), it seems that **Reviewer VrzY** simply ignore most of our efforts and contributions.
> > >
> > >   We would like to argue that the **first point** in **Reviewer VrzY**'s response to our rebuttal is **WRONG** (cf. https://openreview.net/forum?id=vxhzSm1D3J&noteId=nkmArKgOLL). Theorem 8 (as mentioned by **Reviewer VrzY**) is not about our node-wise correction algorithm. Instead, **Proposition 9** is the analysis of our method. We have also already demonstrated that **it is possible for our best method to outperform conventional DCSC algorithms (e.g., ISC)** in **Appendix F** but was simply ignored.
> > >
> > >   For the second point, we have also highlighted that **almost every theoretical work on DCSC includes similar conditions** including those well-known analysis accepted by top-tier venues. We really hope that you could understand that **it is extremely difficult or even impossible to avoid any conditions in a theoretical analysis**. We have also already highlighted that to **explore more relaxed assumptions is our significant future direction** as suggested by **Reviewer VrzY** in **Appendix J** but was simply ignored.
> > >
> > >   In summary, we respectfully **disagree with the assessment of Reviewer VrzY**, which lack a comprehensive evaluation about our three-fold contributions.

---

> ### Author Response · Authors · 2024-11-27
> **Response to Further Comments of Reviewer VrzY**
>
> Dear Reviewer VrzY,
>
> We sincerely appreciate your feedbacks regarding our responses and revisions.
>
> Please note that **Theorem 8 is the analysis on ISC (i.e., a conventional method with a constant $\tau$) but not on ASCENT (i.e., our extension with node-wise corrections)**. Therefore, we do not expect that Theorem 8 can interpret the benefits of ASCENT with node-wise corrections.
>
> Instead, we have provided **an improved analysis for ASCENT with node-wise corrections** in **Proposition 9** (of the revised paper), where we define $\hat \tau_r := {\max}\{ \tau_i | v_i \in \hat S_r \}$ for each cluster $\hat S_r$. Since different nodes $\{ v_i \}$ may have different $\{ \tau_i \}$, different clusters $\{ \hat S_r \}$ may further have different $\{ \hat \tau_i\}$. More importantly, based on **Proposition 9**, we further tried to answer the question: **when can ASCENT potentially outperform ISC?** in **Appendix F** and showed that **it is possible for ASCENT to outperform ISC**. In this sense, we strongly believe that **our analysis can help interpret the benefits of node-wise corrections**.
>
> Regarding the condition and assumption of Theorem 8, we would kindly highlight that **almost all the theoretical studies on DCSC** (including those based on random graph models like DCSBM) **have very similar conditions**. For instance, the well-known DCSBM-based analysis on RSC [1] should first satisfy $[K\ln (4N/\varepsilon )/(\delta  + \tau )] \le {\lambda _K}/(8\sqrt 3 )$ for a sufficiently large number of nodes $N$. **It may also be rather strict to satisfy this condition** as you mentioned, especially for a sufficiently large $N$. However, this paper was still accepted by a top-tier venue and has become a very famous work with high citations.
>
> In summary, we strongly believe that our paper is a good beginning for its spectral-based analysis on DSCC and extension with node-wise corrections. Also as summarized in **Appendix J**, **some new theoretical results on the combinatorial optimization problem of graph-cut minimization can help relax the assumption**.
>
> We really hope that you can re-evaluate our paper by comprehensively considering our (i) **unified theoretical analysis for a series of spectral clustering algorithms** (i.e., NJW, RSC, SCORE+, ISC, and ASCENT) in **Appendix F** and (ii) **novel extension of DCSC with node-wise corrections** (i.e., ASCETN), instead of just focusing on the condition on $\Psi$ in one theorem.
>
> [1] Qin et al. Regularized Spectral Clustering under the Degree-Corrected Stochastic Blockmodel.

---

### Official Review · Reviewer_g4ng · 2024-11-02

**Soundness:** 2
**Presentation:** 2
**Contribution:** 3
**Rating:** 5
**Confidence:** 2

**Summary:**

This paper examines Degree Corrected Spectral Clustering (DCSC) from a pure spectral view,  not requiring any model assumptions. In theory, the author establish a rigorous bound on the mis-clustered volume for the ISC approach (Qing and Wang) with respect to the optimal solution under conductance minimization. This bound depends on both high degree heterogeneity and weak clustering structure.  In algorithm, inspired by recent advances in GNNs, they propose ASCENT which iteratively updates the degree correction term $\tau$, which is used to construct the Laplacian. Experiment results demonstrate  that ASCENT is reduced to the DCSC algorithm, ISC, when the number of iteration is large, due to the over-smoothing issue. However, at early iterations, ASCENT can achieve  better accuracy compared to other DCSC algorithms. Results on real datasets further compare the performance of ASCENT with other DCSC algorithms.

**Strengths:**

The research question addressed is both intriguing and significant. Existing literature generally requires model assumptions, meaning that users must first confirm that the network data indeed fits the model assumptions before applying the corresponding results and error bounds. Understanding the robustness of spectral clustering under minimal assumptions is therefore a valuable contribution to the field, complementing existing work. This paper not only offers rigorous theoretical insights but also introduces an extension of the DCSC algorithm. The comprehensive analysis on various real datasets further strengthens the study.

**Weaknesses:**

The theoretical contributions of the paper could be presented more clearly. While several lemmas and theorems are introduced in Section 3, their purposes and connections are not immediately apparent, making it difficult to follow the logical flow. I suggest adding a high-level overview of the theorems and lemmas to clarify their roles in the analysis before diving into the details. Additionally, I am uncertain how useful the main result (Theorem 8) since the recovery accuracy by spectral clustering without a model assumption is still unknown. Theorem 8 appears to provide a bound comparing the spectral clustering to the optimal solution under conductance minimization, yet it does not provide insight into the accuracy of this optimal solution itself. Furthermore, I am concerned that the condition required for $\Psi$ in Theorem 8 might be too stringent and challenging to satisfy for high degree heterogeneity case. Another limitation is that conductance minimization may not be appropriate for disassortative networks.

Regarding  the ASCENT algorithm, the motivation behind iteratively updating the degree correction term $\tau$ for each node is unclear. While ASCENT demonstrates some advantages in the early stages, it remains unclear  what the number of iteration should be in practice, and the accuracy is only slightly improved as shown in Figure 2. I believe further justification for the optimal iteration number, such as if it is related to certain graph structures, would strengthen the paper. The authors might also consider some analysis to determine optimal iteration numbers for different types of graphs.

**Questions:**

1. Is it possible to consider alternative optimization objectives other than conductance minimization, which might yield an optimal solution that is closer to the output of ISC? Additionally, if we were to use SCORE+ instead of ISC, would this result in a very different upper bound?

2. Is it correct to select the first $K$ eigenpairs of the Laplacian matrix when the eigenvalues are ordered in a decreasing order? If we write $A =\mathbb E A + W$, where $\mathbb E A$ characterizes the low-rank signal of the network model, it is possible that $\mathbb E A$  may have a negative eigenvalue, $\lambda_k<0$, with relatively large magnitude. In this case, selecting the top eigenvalues by their value rather than their magnitude  could lead to the exclusion of the eigenvector associated with $\lambda_k$. This could potentially affect the performance of K-means clustering.  I would like to see the authors address this potential issue in their methodology section and discuss its implications for their results.


3. I don't understand why the clustering cost function include $d_i$ in front of $\| \tilde F_{i,:} - w_r\|^2_2$. It seems more intuitive to consider simply sum of $\| \tilde F_{i,:} - w_r\|^2_2$, which aligns with  the K-means objective. Additional explanation of this choice would be helpful.

4. The paper states that the upper bound $\Psi$ depends on a quantity that indicates the weak clustering structure. However, this is not convincing to me,  especially using a relatively loose upper bound for $\Psi$ to explain this. While this upper bound includes the term $1- \lambda_{K+1}/\lambda_K$ that reflects the weak signal level, I am not sure it is reasonable to say $\Psi$ shows the weak clustering structure. From the definition of $\Psi$, it only depends on $1 - \lambda_{K+2}$, and I don't think we can infer that $1 - \lambda_{K+2}$ is comparable to $1- \lambda_{K+1}/\lambda_K$.


5. In Lemma 4, why are  the eigenvectors compared to the columns of the orthogonal matrix $O$? shouldn't they be compared to the columns in $G$, as they are in Theorem 3?

6. In Theorem 8, the upper bound for $\mu(C_r\Delta \hat S_r)$ holds when
$
\Psi \leq \mu_{\min}/[132(1+ \lambda_1)^2 \alpha K \mu_{\max} ].
$
The RHS is a very small constant when $\alpha >0$ is fixed.  The definition of $\Psi $ says that
$
\Psi = \frac{1}{1 - \lambda_{K+2}} [ 1- \frac{d_{\min} }{d_{\max} + \tau} (1- \bar \phi_K(G)].
$
When there is very high degree heterogeneity such that $\frac{d_{\min} }{d_{\max} + \tau} (1- \bar \phi_K(G)= o(1)$, $\Psi \approx1/(1- \lambda_{K+2})$. Furthermore, $\lambda_{K+2}$ could  be quite small. For instance, for the laplacian of a DCBM, under mild conditions, $|\lambda_{K+2}|\leq C/ \sqrt{n\bar \theta^2} \approx 1/\sqrt{\bar d}$ where $\bar \theta$ is the average of the degree parameters. In this case, if $n\bar \theta^2 \to \infty$ or $\bar d\to \infty$, then $\lambda_{K+2} = o(1)$, and $\Psi \approx 1$.  Given these, I am concerned about whether the condition can be satisfied under high degree heterogeneity. I suggest providing a more thorough discussion of the practical implications of this condition, particularly in the context of high degree heterogeneity, along with specific examples where the condition can be met.

---

> ### Author Response · Authors · 2024-11-24
> **Response to Comments of Reviewer g4ng (1)**
>
> Dear Reviewer g4ng,
>
> We sincerely thank you for summarizing the strengths of our paper. In addition, we have also tried our best to make a major revision according to your comments and suggestions on the weaknesses.
>
> ***
> >**W1**. The theoretical contributions of the paper could be presented more clearly ... I suggest adding a high-level overview of the theorems and lemmas to clarify their roles in the analysis before diving into the details.
>
> **Response to W1**. Thank you for your suggestion regarding the presentation of our analysis. During the revision, we have adopted your suggestion to **add an overview about the logical flow** from **Theorem 3** to **Theorem 8** in **Fig. 1** and **briefly introduce the key idea of proof after introducing a theorem or lemma** in **Section 3**. For instance, the key idea to prove **Theorem 3** is to reformulate the matrix representation ${\bf{G}}$ of the optimal solution using the linear combination of orthogonal eigenvectors $\{ {\bf{u}}_i\}$. **Theorem 7** is proved by contradiction using the upper and lower bounds given by **Theorem 5** and **Theorem 7**, respectively. We really hope that our revisions can help you better understand the logical flow of our analysis.
>
> ***
> >**W2**. I am uncertain how useful the main result (Theorem 8) since the recovery accuracy by spectral clustering without a model assumption is still unknown.
>
> **Response to W2**. It is well known that spectral clustering adopts a relaxed version of the NP-hard combinatorial optimization objective of conductance minimization. To the best of our knowledge, **to provide bonds regarding the feasible solution given by an algorithm (e.g., $(C_1, \cdots, C_K)$ of ISC) w.r.t. the optimal solution (e.g., $(\hat S_1, \cdots, \hat S_K)$ of conductance minimization) is a widely-used strategy in the analysis of many combinatorial optimization problems on graphs**. For instance, we have $\rm{OBJ} \le \gamma {\rm{OPT}}$ (i.e., $\gamma$-approximation) for some well-known algorithms of traveling salesman problem (TSP) based on minimum spanning tree [1,2], with OBJ and OPT as objective values of (i) feasible solution given by the algorithm and (ii) optimal solution. Since **spectral clustering is an approximated algorithm for the NP-hard conductance minimization problem**, it is straightforward to adopt this strategy for our analysis of DCSC. Some early studies on vanilla spectral clustering (e.g., Peng et al. and Mizutani et al. in our paper) also follow this strategy. **Our analysis can be considered as a unified extension of these early studies**.
>
> Please also note that **the mis-clustered volume $\mu (C_r \Delta {\hat S}_r)$ in Theorem 8 is consistent with the accuracy metric of graph clustering**. Following the same renumbering of $(C_1, \cdots, C_K)$ in **Theorem 8**, the clustering accuracy is defined as $\sum\nolimits_{r = 1}^K {|{C_r} \cap {{\hat S}_r}|} /N$. In particular, we have ${C_r} \cap {\hat S_r} = {C_r} \cup {\hat S_r} - {C_r}\Delta {\hat S_r} \le V - {C_r}\Delta {\hat S_r}$ and thus lower mis-clustered volume $\mu (C_r \Delta {\hat S}_r)$ indicates higher clustering accuracy.
>
> During the revision, we have also reduced our analysis to NJW, RSC, AND SCORE+ in **Appendix F**. It extends our analysis to **a unified framework involving a series of spectral clustering**. Based on this framework, we further tried to answer the following questions. (i) *Why can ISC potentially outperform RSC*? (ii) *When can a DCSC algorithm (e.g., RSC) potentially outperform vanilla spectral clustering (i.e., NJW)*? (iii) *When can ASCENT potentially outperform ISC*? This extension can also be considered as a significant contribution of our analysis. In contrast, **most existing analysis only focuses on one specific algorithm (e.g., RSC)**.
>
> We really hope the aforementioned explanations can help you better understand that our analysis is useful, especially from the perspective of combinatorial optimization.
>
> [1] Laporte et al. The Traveling Salesman Problem: An Overview of Exact and Approximate Algorithms.
>
> [2] Pop et al. A Comprehensive Survey on the Generalized Traveling Salesman Problem.

---

> ### Author Response · Authors · 2024-11-24
> **Response to Comments of Reviewer g4ng (2)**
>
> >**W3**. I am concerned that the condition required for
>  in Theorem 8 might be too stringent and challenging to satisfy for high degree heterogeneity case...
>
> >**Q6**. In Theorem 8, the upper bound holds when $\Psi \le 1/[132(1+\lambda_1)^2\alpha K {\tilde\mu}]$ ... In this case, if $n{{\bar \theta }^2} \to \infty$ or $\bar d \to \infty$, then ${\lambda _{K + 2}} = O(1)$ and $\Psi \approx 1$. Given these, I am concerned about whether the condition can be satisfied under high degree heterogeneity.
>
> **Response to W3 & Q6**: To the best of our knowledge, **node degrees in most real-world graphs follow the typical power-law distributions**, where only a very few nodes have large degrees and most nodes have relatively small degrees. In this sense, **it is very extreme and rare that $\bar d \to \infty$ even for high degree heterogeneity** (i.e., large $d_{\max}$ and small $d_{\min}$). For instance, we have $(d_{\min}, d_{\max}, {\bar d}) = (1, 28754, 5.3)$ and $(1, 14815, 17.35)$ for *Youtube* and *LiveJournal*, which are famous large-scale graphs from SNAP. The statistics of datasets in our experiments as summarized in **Table 3** and some studies on large-scale complex systems (e.g., World Wide Web [1,2]) also verify this phenomenon. Therefore, **your concern about $\bar d \to \infty$ may be rare in real applications**.
>
> We are sorry for the insufficient explanations regarding the condition of $\Psi$ in Theorem 8. During the revision, we have adopted your suggestion to provide additional explanations at the end of **Section 3**. In fact, the condition of $\Psi$ indicates an implicit assumption of our analysis that (i) $G$ is well-clustered and (ii) the degree heterogeneity is not so high.
>
> In some early studies on vanilla spectral clustering (e.g., Peng et al. and Mizutani et al. in our paper), a graph is defined to be **well-clustered** if $\bar \phi_K(G) / (1 - \lambda_{K+1})$ is sufficiently small. The well-clustered assumption adopted by Peng et al. and Mizutani et al. indicates that the optimal solution $(\hat S_1, \cdots, \hat S_K)$ (i.e., clustering ground-truth in real applications) describes an explicit clustering structure of $G$. Our analysis can be considered as **a unified extension of these early studies**.
>
> **It is still possible that this condition may not hold when the degree heterogeneity is very high**. To the best of our knowledge, **most related analysis on spectral clustering cannot avoid including similar conditions that may not always hold**. For instance, the well-known DCSBM-based analysis on RSC [3] should first satisfy $[K\ln (4N/\varepsilon )/(\delta  + \tau )] \le {\lambda _K}/(8\sqrt 3 )$ for a sufficiently large number of nodes $N$. **It may also be stringent and challenging to satisfy this condition**, especially for a sufficiently large $N$.
>
> Please also note that **$\tau$ in DCSC can help resist the impact of high degree heterogeneity to the condition**:
>
> $\Psi \le 1/[132(1+\lambda_1)^2\alpha K {\tilde\mu}]$,
>
> $\Psi := {(1 - {\lambda _{K + 2}})^{ - 1}}[1 - \tilde d(1 - {{\bar \phi }_K}(G))]$,
>
> with $\tilde d: = {d_{\min }}/({d_{\max }} + \tau )$.
>
> A larger $\tau$ can result in smaller eigenvalues $\lambda_{K+2}$ and $\lambda_1$, which can further let the **left part of the inequality become smaller and the right part become larger**. In this case, this inequality is more likely to hold.
>
> Also as summarized in **Appendix J** of the revised paper, we plan to further improve this condition by extending some new theoretical results on the combinatorial optimization problem of graph-cut minimization (e.g., conductance minimization in this paper) to DCSC.
>
> [1] Kleinberg et al. The Web as a Graph: Measurements, Models, and Methods.
>
> [2] Barabási et al. Emergence of Scaling in Random Networks.
>
> [3] Qin et al. Regularized Spectral Clustering under the Degree-Corrected Stochastic Blockmodel.

---

> ### Author Response · Authors · 2024-11-24
> **Response to Comments of Reviewer g4ng (3)**
>
> >**W4**. Regarding the ASCENT algorithm, the motivation behind iteratively updating the degree correction term for each node is unclear ... it remains unclear what the number of iteration should be in practice, and the accuracy is only slightly improved as shown in Figure 2 ...
>
> **Response to W4**: Thank you for your comments regarding the motivation, parameter setting (e.g., $L$), and quality improvement of ASCENT.
>
> Please note that as described in the 1st paragraph of **Section 4**, most existing DCSC methods use the 'average/mean' of degrees as their constant correction $\tau$ (e.g., $\tau := \bar d$ and $\delta (d_{\min} + d_{\max})/2$ for RSC and ISC as summarized in **Table 1**). In this sense, **the constant $\tau$ can be considered as a *global* average of degrees**. Based on this interpretation of $\tau$, it is interesting to consider a ***local* average of degrees w.r.t. the local topology of each node, which can be computed by iteratively applying the GNN mean aggregation defined in (12)**. In summary, the motivation of iteratively updating degree corrections via the mean aggregation is that it can compute the ***local* average w.r.t. the local topology of each node**, which serves as **a special case of the *global* average in conventional DCSC methods with a constant $\tau$**.
>
> From the perspective of machine learning, $(\theta, L)$ are hyper-parameters for ASCENT, just like the number of GNN layers in some deep graph clustering baselines (e.g., GAP, SDCN, MinCutPool, DMoN, and DGCluster in our experiments). Moreover, the constant correction $\tau$ in RSC, SCORE+, and ISC can also be considered as a tunable hyper-parameter. Since graph clustering is a typical unsupervised task, **one can tune hyper-parameters of a method (e.g., ASCENT and other graph clustering baselines) using the unsupervised conductance metrics**. Concretely, we computed the conductance metrics w.r.t. the clustering results given by different hyper-parameter settings and then selected the setting with minimum conductance. During the revision, we have added the aforementioned details about hyper-parameters tuning in the last paragraph of **Section 5.1**.
>
> Please also note that **ASCENT can achieve significant improvement in some cases**, e.g., improvement of more than ~3% on **LFR-2** and ~5% on **ogbn-Protein** as highlighted in **Table 5** and **Table 8**. Our main purpose is to consider an extension of node-wise corrections (i.e., ASCENT) and **explore whether it can potentially achieve better clustering quality**, which has not been considered by most existing methods. To some extent, **our experiments validate such a possibility and provide some interesting phenomena**. Therefore, we believe that ASCENT is a good beginning to consider node-wise corrections $\{ \tau_i \}$ for DCSC. As summarized in **Appendix J**, we plan to consider an advanced extension with learnable $\{ \tau_i \}$, instead of manually setting them by adjusting $\{ \theta, L\}$ in our future work.

---

> ### Author Response · Authors · 2024-11-24
> **Response to Comments of Reviewer g4ng (4)**
>
> >**Q1**. Is it possible to consider alternative optimization objectives other than conductance minimization, which might yield an optimal solution that is closer to the output of ISC? Additionally, if we were to use SCORE+ instead of ISC, would this result in a very different upper bound?
>
> **Response to Q1**: To the best of our knowledge, spectral clustering is an algorithm for a relaxed version of several graph-cut objectives, e.g., conductance minimization (or equivalently normalized cut minimization) and ratio-cut minimization, as summarized in [1]. Different objectives have different (matrix) representations ${\bf{G}}$ for the optimal solution.
>
> For instance, we have ${\bf{G}}_{ir} = [d_i / \mu (\hat S_r)]^{0.5}$ if $v_i \in \hat S_r$
>
> and ${\bf{G}}_{ir} = 0$ otherwise for conductance minimization in our paper.
>
> In contrast, we have ${\bf{G}}_{ir} = 1/ |\hat S_r|$ if $v_i \in \hat S_r$
>
> and ${\bf{G}}_{ir} = 0$ otherwise for ratio-cut minimization.
>
> Despite such a difference, we believe that **our analysis can be easily extended to another graph-cut objective (e.g., ratio-cut) with a different definition of ${\bf{G}}$**, because the key idea of our analysis (e.g., **Theorem 3**) is to reformulate ${\bf{G}}$ using the linear combination of orthogonal eigenvalues $\{ {\bf{u}}_i \}$. As demonstrated in [2], there are also some relations between the graph-cut minimization and modularity maximization objectives. For instance, normalized cut minimization (or equivalently conductance minimization) is equivalent to a normalized form of modularity maximization. Therefore, **our analysis can also be easily extended to modularity maximization** by using this relation. We plan to consider these extensions in our future work as summarized in **Appendix J**.
>
> During the revision, we have also reduced our analysis on ISC to SCORE+ in **Appendix F**. As summarized in **Table 9** (of the revised paper), SCORE+ has bounds (w.r.t. $\mu (C_r \Delta \hat S_r)$ and $\phi (C_r)$) very similar to those of ISC. In this sense, **SCORE+ does not result in very different upper bounds**.
>
> [1] Luxburg et al. A Tutorial on Spectral Clustering.
>
> [2] Yu et al. Network Community Discovery: Solving Modularity Clustering via Normalized Cut.
>
> ***
> >**Q2**. Is it correct to select the first eigenpairs of the Laplacian matrix when the eigenvalues are ordered in a decreasing order? ...
>
> **Response to Q2**: Please not that our analysis follows the original standard procedure of a DCSC algorithm (e.g., ISC). Namely, **using the leading largest eigenvalues is the original design of the algorithm we analyze**. Thank you for pointing out such a potential limitation regarding eigenvalues, which may be **a common limitation for most existing spectral clustering approaches**.
>
> Please note that our extension of DCSC (i.e., ASCENT) focuses on the assignment of node-wise corrections and follows the standard steps of an existing DCSC algorithm (i.e., eigen-decomposition, spectral embedding arrangement, normalization, and KMeans clustering of ISC). In this sense, **any possible solution to this limitation of an existing DCSC algorithm** (e.g., considering the absolute values of eigenvalues) **can also be a solution to ASCENT**. We plan to address this potential limitation in our future work.
>
> ***
> >**Q3**. I don't understand why the clustering cost function include $d_i$ in front of $|{\bf{\tilde F}}_i - {\bf{w}}_r|$. It seems more intuitive to consider simply sum of $|{\bf{\tilde F}}_i - {\bf{w}}_r|$, which aligns with the K-means objective. Additional explanation of this choice would be helpful.
>
> **Response to Q3**: To help you better understand why we used $d_i |{\bf{\tilde F}}_{i,:} - {\bf{w}}_r|$, we have given the full proof of **Lemma 4** in **Appendix C** of the revised paper.
>
> The major reason is that for the matrix representation $\bf{G}$ of the optimal solution, we have ${\bf{G}}_{ir} = (d_i / \mu(\hat S_r))^{0.5}$ if $v_i \in {\hat S}_r$. In **the 3rd line from the bottom of Appendix C**, we can (i) **eliminate the $d_i$ in $\bf{G}$**
>
> and further (ii) **extract $\mu (\hat S_r)$ from ${\bf{G}}$** by **using the $d_i$ in $d_i |{\bf{\tilde F}}_{i,:} - {\bf{w}}_r|$**, in order to complete the proof of **Lemma 4** based on **Theorem 3** and Eq. (8).
>
> As highlighted in **Definition 2**, by assuming that **for each node $v_i \in V$, all the $d_i$ copies of ${\bf{\tilde F}}_{i,:}$ are contained in one of $\{ S_1, \cdots, S_K \}$**, **this cost function reduces to the standard KMeans cost**.
>
> Another advantage of this cost is that **it can be directly used to formulate overlapping graph clustering**, where each node may belong to multiple clusters. In general, **disjoint graph clustering (considered in this paper) is a special case of overlapping graph clustering**.
>
> We really hope that our revisions and explanations can help you better understand the reason of using $d_i |{\bf{\tilde F}}_{i,:} - {\bf{w}}_r|$.

---

> ### Author Response · Authors · 2024-11-24
> **Response to Comments of Reviewer g4ng (5)**
>
> >**Q4**. The paper states that the upper bound depends on a quantity that indicates the weak clustering structure. However, this is not convincing to me, especially using a relatively loose upper bound for $\Psi$ to explain this ... From the definition of $\Psi$, it only depends on $1 - \lambda_{K+2}$, and I don't think we can infer that $1 - \lambda_{K+2}$ is comparable to $1 - \lambda_{K+1} / \lambda_K$.
>
> **Response to Q4**: Thank you for your question regarding the physical meaning of $m_K$ in $\Psi$ (i.e., **how the impact of the weakness of clustering structures is considered in our analysis**). Please note that by using the properties of eigenvalues (i.e., $1 > {\lambda_K} \ge {\lambda_{K+1}} \ge {\lambda_{K+2}}$), we can **rigorously ensure** that $(1 - \lambda_{K+2}) \ge (1 - \lambda_{K+1} / \lambda_K)$. In this sense, it is **comparable**.
>
> In our main theoretical results (i.e., **Theorem 8**), we have $\Psi : = {(1 - {\lambda _{K + 2}})^{ - 1}}[1 - {d_{\min }}(1 - {{\bar \phi }_K}(g))/({d_{\max }} + \tau )]$.
>
> Let $\Psi ': = {(1 - {\lambda _{K + 1}}/{\lambda _K})^{ - 1}}[1 - {d_{\min }}(1 - {{\bar \phi }_K}(g))/({d_{\max }} + \tau )]$.
>
> The upper bound in **Theorem 8** can then be formulated as
>
> $\mu ({C_r}\Delta {\hat S_r}) \le {\rm{sth}} \cdot\Psi  \le {\rm{sth}}\cdot \Psi'$,
>
> where $\Psi'$ is directly related to the weakness of clustering structures. In particular, **weaker clustering structures** will result in **a smaller value of ${\rm{sth}}\cdot \Psi'$** and thus will **restrict ${\rm{sth}}\cdot \Psi$ to a smaller value** (i.e., **lower upper bound of $\mu (C_r \Delta \hat S_r)$**). In this sense, our analysis can **reveal the impact of the weakness of clustering structures to the clustering quality** measured by $\mu ({C_r}\Delta {\hat S_r})$.
>
> We really hope that the aforementioned interpretations can help you better understand how the impact of the weakness of clustering structures is considered in our analysis.
>
> ***
> >**Q5**. In Lemma 4, why are the eigenvectors compared to the columns of the orthogonal matrix ${\bf{O}}$? shouldn't they be compared to the columns in ${\bf{G}}$, as they are in Theorem 3?
>
> **Response to Q5**: Please note that **Lemma 4 is derived based on the clustering cost (2) and Eq. (8)**, as highlighted in **Fig. 1** of the revised paper. In particular, Eq. (8) is a rewritten form of the first term of (7) in **Theorem 3**, which **replaces each element ${\bf{G}}_{ir}$ of ${\bf{G}}$ with $[d_i / \mu (\hat S_r) ]^{0.5}$**. Therefore, we do not need to include ${\bf{G}}$ in **Lemma 4**. Instead, $({\bf{o}}_1, \cdots, {\bf{o}}_K)$ from the orthogonal matrix ${\bf{O}}$ can be considered as cluster centers in KMeans clustering. In contrast, ${\bf{G}}$ is used to encode the optimal solution of conductance minimization but not the cluster centers of KMeans.
>
> We really hope that our revisions and explanations can help better understand the key ideas of **Lemma 4**.

---

> > ### Comment · Reviewer_g4ng · 2024-12-02
> >
> > Thanks for the detailed explanations. I greatly appreciate  the inclusion of the diagram to provide an overview of the theoretical analysis. However, I am not satisfied with some responses provided: (1) the restriction of the condition in Theorem 8; (2) the impact of the weakness of clustering structures.
> >
> > For point (1), I disagree with the statement "it is very extreme and rare that $\bar d \to \infty$ even for high degree heterogeneity". In many theoretical works on networks, such as community detection for  SBM (Abbe et al. 2015) and  SCORE (Jin 2015) for DCBM, the condition $\bar d$, the average degree, is required  to be at least $\log(n)$, which indeed goes to $\infty$ when $n\to \infty$. The condition $\bar{d} \geq C \log(n)$ seems necessary for the graph to be connected and for recovery to be possible.  The additional explanations at the end of Section 3 do not seem convincing to me. Enlarging $\tau$ may decrease both $\lambda_1$ and $\lambda_{K+2}$. However, from the definition of $\Psi$, $1 / (1 - \lambda_{K+2})$ remains close to $1$, while the second factor in $\Psi$ increases. It is unclear how these factors are balanced to satisfy the condition in Theorem 8, especially given the large constant of 132.
> >
> > For  point (2), I fully understand that $1 - \lambda_{K+1}/\lambda_K \leq 1- \lambda_{K+2}$. However, if   $1 - \lambda_{K+1}/\lambda_K$ is very small, the upper bound obtained by replacing $m_K$ with $\tilde m_K$ might blow up and become useless. This does not provide a sharp upper bound. Thus,  weaker clustering structures don't say anything about $\Psi$ and further $\mu ({C_r}\Delta {\hat S_r})$.  Also, in your explanation with the re-defined $\Psi$, weaker clustering structures should give larger $\Psi$.
> >
> > Given these points, I still have concerns about the main results and the limitation of the contributions. As such, I will maintain my initial score.

---

> > > ### Author Response · Authors · 2024-12-02
> > > **Response to Further Comments of Reviewer g4ng**
> > >
> > > Dear Reviewer g4ng,
> > >
> > > We sincerely appreciate your feedbacks regarding our responses and revisions.
> > >
> > > We agree that the condition of $\Psi$ is related to the degree heterogeneity. We also highlighted that **it is still possible that this condition may not hold when the degree heterogeneity is very high** in our rebuttal and revised paper.
> > >
> > > We would like to also highlight that **most related analysis on spectral clustering cannot avoid including similar conditions that may not always hold**. For instance, the well-known DCSBM-based analysis on RSC [1] should first satisfy $[K\ln (4N/\varepsilon )/(\delta  + \tau )] \le {\lambda _K}/(8\sqrt 3 )$ for a sufficiently large number of nodes $N$. It may also be challenging to satisfy this condition (e.g., for a sufficiently large $N$) as you mentioned. However, this paper was still accepted by a top-tier venue and has become a very famous work with high citations.
> > >
> > > We really hope that you could understand that **it is extremely difficult or even impossible to avoid any conditions in a theoretical analysis**.
> > >
> > > During the rebuttal, we have check statistics of many real-world datasets (e.g., from SNAP and OGB). As mentioned in our rebuttal, most of them, even for those with very high maximum degrees (e.g., 14,815), are still with low average degree (e.g., $17.35$). Even though $\bar d \to \infty$ may occurs as you mentioned. **We still believe our observations on real-world graphs**. Since you also mentioned 'community detection for SBM/DCSBM', it indicates that **some real-world graphs may NOT strictly follow SBM/DCSBM** as you mentioned.
> > >
> > > From our perspective, the value of $1 - {\lambda _{K + 1}}/{\lambda _K}$ must affect the value of $1 - {\lambda _{K + 2}}$. Therefore, **it must have affect to the upper bound of $\mu (C_r \Delta \hat S_r)$**, even though such an affect is not direct. In this sense, our analysis can, to some extent, still reflect the impact of weak clustering structures.
> > >
> > > Please also note that in addition to the pure spectral analysis on DCSC you focused on, our contributions also include the (i) **unified framework** involving a series of spectral clustering algorithms (cf. **Appendix F**) and (ii) **extension of DCSC with node-wise corrections**. We really hope that you can comprehensively consider these three contributions.
> > >
> > >
> > > [1] Qin et al. Regularized Spectral Clustering under the Degree-Corrected Stochastic Blockmodel.

---

### Official Review · Reviewer_YKQp · 2024-11-03

**Soundness:** 2
**Presentation:** 2
**Contribution:** 2
**Rating:** 3
**Confidence:** 3

**Summary:**

This paper provides novel analyses of the performance of degree-corrected spectral clustering.
Compared to the existing analyses, the advantage of this analysis is that it does not assume a specific random graph model.
To build on the analysis, this paper also proposes ASCENT, inspired by the recent over smoothing discussion of GNNs.
This paper also empirically demonstrates the effectiveness of the proposed ASCENT.

**Strengths:**

-- The theoretical bound on the mis-clustering rate without assuming the random model graph.

-- The node-wise correction model of the DCSC and also theoretical results for the model

**Weaknesses:**

-- It is not clear how the bound is useful/non-trivial. How large the coefficient is? For instance, from the bound of Qin & Rohe, the spectral clustering on DCSBM is shown to be consistent, i.e., the bound is effectively smaller with respect to the size of the graph. Due to this, the bound of DCSBM is non-trivial. Can you say your bound is not trivial?

-- Since the $\Phi$ needs to be lower than \mu_{\min}/(132 (1+\lmabda_{1})^2 \alpha K \mu_{\max}), when this is satisfied? For a heterophilous graph, $\lambda_{1}$ tends to be smaller since the topological information for a heterophilous graph spans from smaller to larger frequencies. Thus, the analysis of heterogeneity carries this assumption. Yes, in this sense, while this paper does not assume the random graph, you still have some assumptions for the graph. Thus, you need to clarify what this assumption is.

In any case, since the analysis is still weak for the bound, I vote for rejection at this point.

**Questions:**

Please see the weakness.

---

> ### Author Response · Authors · 2024-11-24
> **Response to Comments of Reviewer YKQp (1)**
>
> Dear Reviewer YKQp,
>
> We sincerely thank you for summarizing the strengths of our paper. Besides, we have also tried our best to make a major revision according to your comments on the weaknesses.
>
> ***
> >**W1**. It is not clear how the bound is useful/non-trivial. How large the coefficient is? ... Can you say your bound is not trivial?
>
> **Response to W1**: As highlighted in **Theorem 8**, **the clustering quality measured by the mis-clustered volume $\mu (C_r \Delta \hat S_r)$ and conductance $\phi (C_r)$ is related to (i) degree heterogeneity and (ii) weakness of clustering structures**. In this sense, our analysis can quantitatively reveal impacts of these two properties to the clustering quality of DCSC. It is **useful**, since it is **consistent with the fact that the quality of a graph clustering algorithm should be affected by the several inherent properties of the graph to be partitioned** (e.g., degree heterogeneity and weakness of clustering structures).
>
> It is well known that spectral clustering adopts a relaxed version of the NP-hard combinatorial optimization objective of conductance minimization. To the best of our knowledge, **to provide bonds regarding the feasible solution given by an algorithm (e.g., $(C_1, \cdots, C_K)$ of ISC) w.r.t. the optimal solution (e.g., $(\hat S_1, \cdots, \hat S_k)$ of conductance minimization) is a widely-used strategy in the analysis of many combinatorial optimization problems on graphs**. For instance, we have $\rm{OBJ} \le \gamma {\rm{OPT}}$ (i.e., $\gamma$-approximation) for some well-known algorithms of traveling salesman problem (TSP) based on minimum spanning tree [1,2], with OBJ and OPT as objective values of (i) feasible solution given by the algorithm and (ii) optimal solution. Since **spectral clustering is an approximated algorithm for the NP-hard conductance minimization problem**, it is straightforward to adopt this strategy for our analysis of DCSC. Some early studies on vanilla spectral clustering (e.g., Peng et al. and Mizutani et al. in our paper) also follow this strategy. **Our analysis can be considered as a unified extension of these early studies**.
>
> During the revision, we have also reduced our analysis to NJW, RSC, AND SCORE+ in **Appendix F**. It extends our analysis to **a unified framework involving a series of spectral clustering**. Based on this framework, we further tried to answer the following questions. (i) *Why can ISC potentially outperform RSC*? (ii) *When can a DCSC algorithm (e.g., RSC) potentially outperform vanilla spectral clustering (i.e., NJW)*? (iii) *When can ASCENT potentially outperform ISC*? This extension can also be considered as a significant contribution of our analysis. **In contrast, most existing analysis (including the work you mentioned) only focuses on one specific algorithm (e.g., RSC)**.
>
> We really hope that the aforementioned explanations can help you better understand that our analysis is useful, especially from the perspective of combinatorial optimization.
>
> [1] Laporte et al. The Traveling Salesman Problem: An Overview of Exact and Approximate Algorithms.
>
> [2] Pop et al. A Comprehensive Survey on the Generalized Traveling Salesman Problem.

---

> ### Author Response · Authors · 2024-11-24
> **Response to Comments of Reviewer YKQp (2)**
>
> >**W2**. Since $\Psi$
>  needs to be lower than $\mu_{\min}/(132 (1+\lambda_{1})^2 \alpha K \mu_{\max})$, when this is satisfied? ... the analysis of heterogeneity carries this assumption ... you still have some assumptions for the graph. Thus, you need to clarify what this assumption is.
>
> **Response to W2**: We are sorry for the insufficient explanations regarding the condition of $\Psi$ in Theorem 8. During the revision, we have adopted your suggestion to give additional explanations regarding the assumption it implies at the end of **Section 3**.
>
> Concretely, to ensure that the condition $\Psi \le 1/[132(1+\lambda_1)^2\alpha {\tilde \mu} K]$ holds, we need small $\bar \phi_K (G) / (1 - \lambda_{K+2})$ and large $d_{\min} / [(d_{\max + \tau}) (1 - \lambda_{K+2})]$. In some early studies on vanilla spectral clustering (e.g., Peng et al. and Mizutani et al. in our paper), a graph is defined to be **well-clustered** if $\bar \phi_K(G) / (1 - \lambda_{K+1})$ is sufficiently small. It is consistent with that $\bar \phi_K (G) / (1 - \lambda_{K+2})$ is small. The well-clustered assumption adopted by Peng et al. and Mizutani et al. indicates that the optimal solution $(\hat S_1, \cdots, \hat S_K)$ (i.e., clustering ground-truth in real applications) describes an explicit clustering structure of $G$. Our analysis can be considered as **a unified extension of these early work**. Moreover, large $d_{\min} / [(d_{\max + \tau}) (1 - \lambda_{K+2})]$ indicates that the degree heterogeneity should not be very high. In summary, the condition in **Theorem 8** implies an implicit assumption that (i) $G$ is well-clustered and (ii) the degree heterogeneity is not so high.
>
> Also as summarized in **Appendix J** of the revised paper, we plan to further improve this condition by extending some new theoretical results on the combinatorial optimization problem of graph-cut minimization (e.g., conductance minimization in this paper) to DCSC.
>
> We really hope that the aforementioned explanations and our revisions can help you better understand the implicit assumption of our main theoretical results (i.e., Theorem 8).
>
> ***
> **Additional Rebuttal**: Please note that in addition to our **unified analysis on DCSC**, we also proposed ASCENT, an extension of DCSC with node-wise corrections $\{ \tau_i \}$, and explored its potential to achieve better clustering quality. To the best of our knowledge, **we are the first to consider DCSC with node-wise corrections $\{ \tau_i \}$ determined by the mean aggregation mechanism of GNNs**. In particular, ASCENT reduces to existing DCSC methods (e.g., ISC) when encountering the over-smoothing of mean aggregation.
>
> We really hope that you can re-evaluate our paper by comprehensively considering our contributions, including the **unified analysis** mentioned in our **response to W1** and the **extension of DCSC** (i.e., ASCENT).

---

> > ### Comment · Reviewer_YKQp · 2024-11-27
> >
> > Thank you for the rebuttal. In the second response, I thought the authors might want to provide the easiest non-trivial example if the punchline of this paper is that you do not assume the specific random model. Since this is the punchline, I still think that the authors want to state what is truly novel in this paper, which I believe has not been explored yet. Thus, I still maintain my score.

---

> > > ### Author Response · Authors · 2024-11-27
> > > **Response to Further Comments of Reviewer YKQp**
> > >
> > > Dear Reviewer YKQp,
> > >
> > > We sincerely appreciate your feedbacks regarding our responses and revisions.
> > >
> > > Please note that although our analysis still indicates some assumptions due to the condition on $\Psi$, it is **an *alternative analysis* from the perspective of spectral theory and combinatorial optimization, without using random graph models**. In contrast, most existing theoretical studies on DCSC rely on random graph models (e.g., DCSBM). Please note that we **did NOT claim that our analysis does not rely on any assumptions** or **our analysis is better than existing studies with random graph models**. We only highlight that our analysis is conducted from a different perspective. Such a **difference** indicates the potential **novelty**.
> > >
> > > Please also note that we have extended to our analysis to **a unified framework involving a series of spectral clustering algorithms** (e.g., NJW, RSC, SCORE+, ISC, and ASCENT) as highlighted in **Appendix F** and **Table 9**. Based on this unified framework, we also tried to answer three questions. (i) *Why can ISC potentially outperform RSC*? (ii) *When can a DCSC algorithm (e.g., RSC) potentially outperform vanilla spectral clustering (i.e., NJW)*? (iii) *When can ASCENT potentially outperform ISC*? In contrast, **most related theoretical studies** (including the one you mentioned, i.e., Qin & Rohe et al.) **only consider one single method and thus cannot even answer the three questions**.
> > >
> > > We would further kindly highlight that we also proposed **ASCENT**, **an extension of DCSC with node-wise corrections** and explored its potential to achieve better clustering quality. To the best of our knowledge, **we are the first to consider DCSC with node-wise corrections  determined by the mean aggregation mechanism of GNNs**. **Reviewer g4ng**, **Reviewer VrzY**, **Reviewer 2rdr**, and **Reviewer hGxZ** all gave positive comments to such a new extension.
> > >
> > > We really hope that you can re-evaluate our paper by comprehensively considering these **three contributions** instead of just focusing on whether our analysis needs an assumption.

---

> > > ### Author Response · Authors · 2024-12-02
> > > **Respectfully Disagree with the Assessment of Reviewer YKQp**
> > >
> > > Dear AC
> > >
> > >   During the rebuttal period, we have taken a great amount of efforts to give responses and revise our paper based on comments of all the reviewers. The corresponding revisions have been highlighted in **blue text** in the newly submitted paper.
> > >
> > >   Although we have highlighted again and again that **our contributions is three-fold** (i.e., (i) pure spectral analysis for DCSC, (ii) a unified framework involving a series of spectral clustering algorithm in Appendix F, and (iii) extension of DCSC with node-wise corrections), **Reviewer YKQp** only focuses on the basic strategy to conduct our theoretical analysis (e.g., whether it is non-trivial) while **simply ignoring our all other efforts and contributions**.
> > >
> > >   In the rebuttal, we have also tried our best to highlight that **this strategy is widely used in the analysis of combinatorial optimization problems on graphs**. From this perspective, it is 'non-trivial'. However, **Reviewer YKQp** did not give direct response to this point.
> > >
> > >   In summary, we respectfully **disagree with the assessment of Reviewer YKQp**, which lack a comprehensive evaluation about our three-fold contributions.

---

### Author Response · Authors · 2024-11-24
**Overall Response & Summary of Revisions**

Dear Reviewers,

We sincerely appreciate your insightful comments and constructive suggestions that helped us achieve a better presentation of our paper. We have comprehensively adopted the suggestions of all the reviewers and tried our best to make a major revision according to your concerns. All the major revisions have been highlighted with **blue text** in the newly submitted version of our paper. We really hope that our revisions and responses can address your concerns. For convenience, we summarize our major revisions as follows.

- To accurately describe the advantages of our analysis, we changed the statement 'indicate the ability of DCSC to handle (i) high degree heterogeneity and (ii) weak clustering structures' to '**indicate impacts of (i) high degree heterogeneity and (ii) weak clustering structures to the clustering quality of DCSC**'.
- We have added a **high-level overview** to demonstrate the logical flows of our theoretical analysis in **Fig. 1**.
- To ensure that our analysis is self-contained, we have given the full proof of **Lemma 4** in **Appendix C**.
- [*New Theoretical Results*] In addition to the upper bound of mis-clustered volume $\mu (C_r \Delta \hat S_r)$, we have also derived the **upper bound of conductance** $\phi (C_r)$ w.r.t. the optimal solution $\phi (\hat S_r)$ in **Theorem 8** and provided the corresponding proof in **Appendix D**.
- We have provided some more explanations regarding the physical meaning and conditions of our main theoretical results just after **Theorem 8** at the end of **Section 3**.
- [*New Theoretical Results*] We have replaced Proposition 9 and Theorem 10 in the previous version with an improved analysis for ASCENT in **Proposition 9** of the revised paper and provided the corresponding proof in **Appendix E**.
- [*New Theoretical Results*] In **Appendix F**, we have provided **further theoretical analysis** on DCSC, which extends our analysis to **a unified framework** involving a series of spectral clustering algorithms. Concretely, we reduced our theoretical analysis on ISC to NJW, RSC, and SCORE+, and tried to answer three questions. (i) *Why can ISC potentially outperform RSC?* (ii) *When can a DCSC algorithm potentially outperform vanilla spectral clustering?* (iii) *When can ASCENT potentially outperform ISC?*
- In **Appendix G**, we have added further **complexity analysis** for ASCENT.
- [*New Experiment Results*] In **Appendix I**, we have conducted additional **efficiency analysis** in terms of runtime for all the methods, with corresponding results reported in **Tables 27-30**.
- In **Appendix J**, we have summarized some new future research directions according to the concerns of reviewers.
- We have tried our best to find some potential grammatical mistakes as well as typos and revised them.

---

### Author Response · Authors · 2024-12-03
**Summary of our Contributions and Revisions (1/2)**

Dear AC,

  Since the rebuttal period will end soon, we would highlight that our major **contributions** of this paper are **three-fold**.

  - (1) **Pure Spectral Analysis on DCSC**. Different from most existing analysis on DCSC that use random graph models (e.g., DCSBM), to the best of our knowledge, we are the first to consider an alternative analysis from the perspectives of spectral theory and combinatorial optimization, while providing upper bounds for both mis-cluster volume $\mu (C_r \Delta \hat S_r)$ and conductance $\phi (C_r)$ w.r.t. the optimal solution to conductance minimization.

  - (2) **A Unified Theoretical Framework Including a Series of Spectral Clustering Algorithms**. We have also extended our analysis to a unified framework involving a series of spectral clustering algorithms (e.g., NJW, RSC, SCORE+, ISC, and ASCENT) in **Appendix F**. This framework can provide the upper bounds of $\mu (C_r \Delta \hat S_r)$ and $\phi (C_r)$ for each method. Based on this framework, we can further answer three questions. (i) Why can ISC potentially outperform RSC? (ii) When can a DCSC algorithm potentially outperform vanilla spectral clustering? (iii) When can ASCENT potentially outperform ISC? However, most related theoretical studies only focus on one single algorithm (e.g., RSC) and cannot answer these questions.

  - (3) **Extension of DCSC with Node-Wise Corrections**. To the best of our knowledge, we are the first to consider an extension of DCSC (i.e., ASCENT), where different nodes $\{ v_i \}$ may have different corrections $\{ \tau_i \}$ determined by a GNN mean aggregation, and explore its potential to achieve better clustering quality, especially in some early aggregation stages before over-smoothing. In contrast, most existing DCSC methods only use a constant correction $\tau$ for all the nodes.

---

> ### Author Response · Authors · 2024-12-03
> **Summary of our Contributions and Revisions (2/2)**
>
> During the rebuttal period, we have also taken a great amount of efforts to give responses and revise our papers according to suggestions and concerns of all the reviewers. The major revisions to address reviewers' concerns are summarized as follows.
>
>   - **New Theoretical Results** to Demonstrate the Advantages of Node-Wise Corrections (in ASCENT). We have replaced Proposition 9 and Theorem 10 in the previous version with an improved analysis (cf. **Proposition 9** in the revised paper) using cluster-wise corrections $\{ \hat \tau_i \}$. Since different nodes $\{ v_i \}$ in ASCENT may have different corrections $\{ \tau_i \}$, different clusters $\{ \hat S_r \}$ have also have different $\{\hat \tau_i\}$. We have also answered the question: *when can ASCENT potentially outperform ISC* in Appendix F and showed that *it is possible for ASCENT (with node-wise corrections) to outperform ISC (with a constant correction)*. Most reviewers did not raise further concerns about this revision. However, it seems that **Reviewer VrzY ignored this revision** and simply reached a conclusion that 'there is no substantial difference from the previous work'.
>
>   - **Further Complexity and Efficiency Analysis** to Show the Impact of Node-Wise Corrections Derivation. We have provided additional complexity analysis in **Appendix G** and efficiency analysis in **Appendix I**, which demonstrate that *the derivation of node-wise corrections will not increase the overall complexity and is also not the bottleneck of ASCENT*. Reviewers did not raise further concerns about this revision.
>
>   - **Further Explanations about the Condition of $\Psi$** in Main Theoretical Results. We have given further explanations about the condition of $\Psi$ and corresponding assumptions in **Section 3** (after **Theorem 8**). Some reviewers have raised further concerns that this condition of $\Psi$ is strict. We would kindly highlight that **almost all the theoretical studies on DCSC have very similar conditions**. Some of these papers were still accepted by top-tier venues and have achieved high citations. We really hope that you can understand that **it is extremely difficult or even impossible to avoid any conditions in a theoretical analysis**. We have also adopted the suggestion of some reviewers to discuss some future directions to improve this condition in **Appendix J**.
>
>   - **Compromise about the Presentation**. We have also comprehensively adopted suggestions from all the reviewers to adjust the presentation of our paper. We really hope that **people with different backgrounds may have different opinions about the presentation of a paper with rigorous theoretical analysis**. We have tried our best to make a compromise among all the reviewers' comments regarding the presentation.
>
>
>   Our responses to clarify the misunderstanding of reviewers are summarized as follows.
>
>   - **Motivations of Using GNN Mean Aggregation for Node-Wise Corrections**. We have highlighted that existing DCSC algorithms use the *global average node degree* to set the constant correction $\tau&. In contrast, ASCENT adopts *local average degrees w.r.t. local topology of each node* via the GNN mean aggregation to set node-wise corrections $\tau_i$. As the number of iterations increases, *local average degrees* will finally reduce to the *global average degree* due to over-smoothing. Reviewers did not raise further concerns about this revision.
>
>   - **Quality Improvement of ASCENT**. We have highlighted that ASCENT can achieve significant quality improvement (e.g., more than ~3% on **LFR-2** and ~5% on **ogbn-Protein** as in **Table 5** and **Table 8**) compared with conventional DCSC baselines in some cases. Reviewers did not raise further concerns about this revision.
>
>   ***
>
>   In summary, we believe that our revisions and responses have addressed most of the concerns. However, it seems that **some reviewers just simply ignore most of our contributions, responses, and revisions, especially our efforts in the rebuttal peroid, lacking a comprehensive evaluation of our paper**.

---

### Meta-Review · Area_Chair_A2qR · 2024-12-20

**Metareview:**

The paper gives an analysis of the performance of "degree correlated spectral clustering" DCSC, which is a form of spectral clustering where the graph adjacency matrix is normalized by a shifted degree matrix (i.e. D+taul I for some parameter tau). The main strength of the paper is that this analysis is general, and not just under a random graph model like a stochastic block model variant. In particular, the analysis bounds the quality of the solution in terms of an optimal solution and several parameters of the input graph.

The main weaknesses of the paper were 1) its presentation is difficult to follow and 2) the upshot of the theoretical results is not clear. Do they fundamentally change our understanding of DCSC or apply in a natural setting where we previously did not have theoretical bounds for the method? Several reviewers struggled to interpret the bounds intuitively, understand why assumed conditions should be met or parameters should be bounded, and understand when the bounds would be non-vacuous or how they could be verified in practice. Ultimately this gap was the main reason for rejecting the paper.

**Additional Comments On Reviewer Discussion:**

The excessive length of the rebuttal and repeated summaries of discussions in the authors' own words hampered efficient evaluation of the paper. A more concise and to-the-point rebuttal would be more effective in the future.

During the rebuttal period the authors made some changes which could significantly strengthen the paper -- notably extending their analysis to apply to several related spectral clustering variants and using their approach to understand differences in performance between these variants. We encourage the authors to continue to improve the presentation of the paper and include these stronger results in a resubmission. We encourage the authors to think about how their main theoretical results could be made more understandable so that a reader can really take away a high-level idea from the theorems, beyond a heavily parameterized and difficult to verify error bound.

---

### Decision · Program_Chairs · 2025-01-22

Reject